# Linking global terrestrial and ocean biogeochemistry with process-based, coupled freshwater algae-nutrient-solid dynamics in LM3-FANSY v1.0

Minjin Lee[1], Charles A. Stock[2], John P. Dunne[2], Elena Shevliakova[2]

[1]Program in Atmospheric and Oceanic Sciences, Princeton University; Princeton, NJ 08540, USA
[2]NOAA/Geophysical Fluid Dynamics Laboratory; Princeton, NJ 08540, USA

*Correspondence to*: Minjin Lee (minjinl@princeton.edu)

**Abstract.** Estimating global river solids, nitrogen (N), and phosphorus (P), in both quantity and composition, is necessary for understanding the development and persistence of many harmful algal blooms, hypoxic events, and other water quality issues in inland and coastal waters. This requires a comprehensive freshwater model that can resolve intertwined algae, solid, and nutrient dynamics, yet previous global watershed models have limited mechanistic resolution of instream biogeochemical processes. Here we develop a global, spatially explicit, process-based, Freshwater Algae, Nutrient, and Solid cycling and Yields (FANSY) model and incorporate it within the Land Model LM3. The resulting model, LM3-FANSY v1.0, is intended as a baseline for eventual linking of global terrestrial and ocean biogeochemistry in next generation Earth System Models to project global changes that may challenge empirical approaches. LM3-FANSY explicitly resolves interactions between algae, N, P, and solid dynamics in rivers and lakes at 1 degree spatial and 30 minute temporal resolution. Simulated suspended solids (SS), N, and P in multiple forms (particulate/dissolved, organic/inorganic) agree well with measurement-based yield (kg km$^{-2}$ yr$^{-1}$), load (kt yr$^{-1}$), and concentration (mg l$^{-1}$) estimates across a globally distributed set of large rivers, with an accuracy comparable to other global nutrient and SS models. Furthermore, simulated global river loads of SS and N and P in different forms to the coastal ocean are consistent with published ranges, though regional biases are apparent. River N loads are estimated to contain approximately equal contributions by dissolved inorganic N (41%) and dissolved organic N (39%), with a lesser contribution by particulate organic N (20%). For river P load estimates, particulate P, which includes both organic and sorbed inorganic forms, is the most abundant form (64%), followed by dissolved inorganic and organic P (25% and 11%). Time series analysis of river solid and nutrient loads in large U. S. rivers for the period ~1963-2000 demonstrates that simulated SS and N loads in different N forms covary with variations of measurement-based loads. LM3-FANSY, however, has less capability of capturing interannual variability of P loads, likely due to the lack of terrestrial P dynamics in LM3. Analyses of the model results and sensitivity to components, parameters, and inputs suggest that fluxes from terrestrial litter and soils, wastewater, and weathering are the most critical inputs to the fidelity of simulated river nutrient loads to observation-based estimates. Sensitivity analyses further demonstrate a critical role of algal dynamics in controlling the ratios of inorganic and organic nutrient forms in freshwaters. While the simulations are able to capture significant cross-watershed contrasts at a global scale, disagreement for individual rivers can be substantial. This

limitation is shared by other global river models and could be ameliorated through further refinements in nutrient sources, freshwater model dynamics, and observations. Current targets for future LM3-FANSY development include the additions of terrestrial P dynamics, freshwater carbon, alkalinity, and enhanced sediment dynamics, and anthropogenic hydraulic controls.

## 1 Introduction

Dramatic increases in fossil fuel combustion, deforestation, agriculture, fertilizer use, and sewage outflows have increased loadings of terrestrial sediments and nutrients (e.g., nitrogen (N), phosphorus (P)) to rivers and coastal waters and changed N:P ratios (Cordell et al., 2009; Fowler et al., 2013; Lee et al., 2019; Sytvitski et al., 2005). These changes in sedimentary and nutrient loadings have altered turbidity and biogeochemistry in many freshwater and coastal ecosystems. The consequences include 1) changes in ecosystem productivity and carbon (C) exports (Liu et al., 2021), 2) increases in frequency, duration, and severity of harmful algal blooms (HABs) (Anderson et al., 2002; 2021; Heisler et al., 2008; Paerl et al., 2018) and hypoxic dead zones (Diaz and Rosenberg, 2008), and 3) perturbations of aquatic plant, seagrass, and coral reef ecosystems, incurring substantial socioeconomic costs (Lacoul and Freedman, 2006; McLaughlin et al., 2003; Restrepo et al., 2006).

Resolving prominent drivers of the aforementioned aquatic ecosystem consequences requires a freshwater biogeochemistry model that captures intertwined algae, nutrient, and solid dynamics. In general, strong positive relationships have been observed between P and phytoplankton production in freshwaters, while N increases have been linked with the development of large algal blooms and hypoxic events in estuarine and coastal waters (Howarth and Marino, 2006; Smith, 2003). In particular, excessive inorganic nutrients, which are generally more bioavailable than organic forms (Sipler and Bronk, 2004), have been recognized as critical drivers of algal blooms (including non-HABs) and hypoxic events (Kemp et al., 2005). Shifts in community composition towards more toxic or harmful algal species furthermore have been attributed to changes in nutrient supply ratios, including N:P (Anderson et al., 2002; Heisler et al., 2008) and relative abundance of different N and P forms (e.g., nitrate ($NO_3^-$, Parsons et al., 2002), ammonium ($NH_4^+$, Trainer et al., 2007; Leong et al., 2004), urea (Glibert et al., 2001; Glibert and Terlizzi, 1999), and dissolved inorganic N and P (DIN and DIP, Glibert et al., 2008)). Such shifts can be explained by differences in algal species-specific nutrient acquisition pathways that are controlled by nutritional status and preferences, uptake capability, and physiological status (Anderson et al., 2002). Furthermore, nutrient and algae dynamics are strongly linked with solid dynamics, for example, through phosphate ($PO_4^{3-}$) sorption/desorption interactions with solid particles (McGechan and Lewis, 2002) and algae growth reduction due to light shading by suspended solids (SS) (Dio Toro, 1978). Estimating river solids, N, and P in both quantity and composition resulting from intertwined algae, nutrient, and solid dynamics is thus necessary for understanding the development and persistence of many HABs and hypoxic events.

Building confidence in projected global freshwater biogeochemistry changes rests in part on the development of process-based models that are robust under unprecedented conditions expected in the next century. Process-based freshwater biogeochemistry models of the N cycle alone (LM3-TAN, Lee et al., 2014; DLEM, Tian et al., 2020; IMAGE-DGNM, Vilmin et al., 2020; INCA, Wade et al., 2002) or the P cycle alone (DLEM, Bian et al., 2022; IMAGE-DGNM, Vilmin et al., 2022) have been widely applied across scales. However, prior applications of models that capture coupled algae, multi-

nutrient, and/or solid cycles, such as RIVE (Billen et al., 1994) and QUAL2K (Pelletier et al., 2006), have generally been limited to relatively small watersheds. Modeling river nutrient yields/loads on a global scale, in both magnitude and form, has been challenged by the difficulty of balancing desired details of instream biogeochemical processes with limitations imposed by available knowledge, input and validation datasets.

Global NEWS (Mayorga et al., 2010) and IMAGE-GNM (Beusen et al., 2015; 2016) are global watershed models that have been widely used for pioneering simulations of river nutrient loads on global scales. Global NEWS estimates have been shown to be consistent with measurement-based estimates across a globally distributed set of major rivers, and provided important nutrient inputs for global and regional ocean biogeochemistry model simulations. Global NEWS and IMAGE-GNM, however, do not resolve coupled algae, nutrient, and solid dynamics in freshwaters despite their intertwined

dynamics. Global NEWS, representing a hybrid of empirical, statistical, and mechanistic components, formulates and implements different elements and their chemical forms independently based on basin-averaged properties. IMAGE-GNM applied at a global scale does not differentiate dissolved, particulate, inorganic, and organic nutrient forms, though such differentiation has been considered at regional scales. Global applications of both models do not mechanistically resolve instream biogeochemical processes.

Prior global watershed models are also limited in their capacity to represent process-based nutrient storage in terrestrial plants and soils. Global NEWS assumes that nutrients are in steady state and do not accumulate on land. IMAGE-GNM does not explicitly simulate terrestrial nutrient dynamics, such as vegetation growth, leaf fall, natural and fire-induced mortality, and soil microbial processes, but takes a mass balance approach to calculate dynamic soil nutrient budgets. Organic nutrient

delivery to rivers, however, depends on changes in vegetation and soil organic nutrient storage in response to the aforementioned terrestrial dynamics under long-term (multiple decades to centuries) historical climate and land use changes. Terrestrial storage changes, for example, have been shown to significantly alter multi-decadal river nutrient trends (Van Meter et al., 2018; Lee et al., 2019) and seasonal to multi-year river nutrient extremes (Kaushal et al., 2008; Lee et al., 2016; Lee et al., 2021).

Here we develop a global, spatially explicit, process-based, Freshwater Algae, Nutrient, and Solid cycling and Yields (FANSY) model, and incorporate it within the National Oceanic and Atmospheric Administration (NOAA)/Geophysical

Fluid Dynamics Laboratory (GFDL) Land Model LM3 which is capable of resolving coupled water, C, and N dynamics and storage changes in a vegetation-soil system (Lee et al., 2014; Lee et al., 2019). The resulting coupled terrestrial-freshwater ecosystem model LM3-FANSY constitutes a significant step toward a more process-based representation of the coupled, freshwater algae, nutrient, and solid dynamics. LM3-FANSY v1.0 is aimed at linking global terrestrial and ocean biogeochemistry towards next generation Earth System Models. Here we provide a detailed model description, performance assessment against measurement-based estimates of solids and nutrients across world major rivers, and sensitivity evaluation to a range of components, parameters, and inputs.

## 2 Model description

### 2.1 LM3-FANSY framework

LM3-FANSY is an expansion of NOAA/GFDL LM3-Terrestrial and Aquatic Nitrogen (TAN) (Lee et al., 2014; Lee et al., 2019) to include a terrestrial soil erosion process and comprehensive freshwater sediment and biogeochemical dynamics (Sect. 2.2). The terrestrial component LM3, which has been described in detail elsewhere (Gerber et al., 2010; Milly et al., 2014; Shevliakova et al., 2009), captures coupled water, C, and N dynamics within a vegetation-soil system. LM3 simulates transfers and transformations of three N species (i.e., organic, $NH_4^+$, and $NO_3^-$) for vegetation and soil systems, considering the effects of anthropogenic N inputs, land use, atmospheric $CO_2$, and climate over timescales of hours to centuries. LM3 simulates the distribution of five vegetation functional types (C3 and C4 grasses, temperate deciduous, tropical, and cold evergreen trees) based on prevailing climate conditions and C-N storage in vegetation including leaves, fine roots, sapwood, heartwood, and labile storage. There are 4 soil organic pools (fast/slow litter and slow/passive soil) and 2 soil inorganic pools ($NH_4^+$ and $NO_3^-$). Scenarios of land use states and transitions are used to simulate four land use types (primary lands – lands effectively undisturbed by human activities, secondary lands – abandoned agricultural land or regrowing forest after logging, croplands, and pastures). LM3 captures key terrestrial dynamics that affect the state of vegetation and soil C-N storage, such as vegetation growth, leaf fall, natural and fire induced mortality, deforestation for agriculture, wood harvesting, reforestation after harvesting, and various soil microbial processes. LM3 extended to include a global river routing and lake model (Milly et al., 2014) is thus well suited to simulate the delivery of terrestrial N to rivers and coastal waters.

The terrestrial component LM3, including the newly introduced terrestrial soil erosion process (Sect. 2.2.1), receives N inputs of fertilizer applications and atmospheric deposition, simulates biological N fixation, and estimates N outputs including net harvest (N in harvested wood, crops, and grasses after subtracting out internally recycled inputs, e.g., manure applied to croplands and sewage), emissions to the atmosphere, and eroded sediment and N fluxes from terrestrial to river systems. In addition to terrestrial runoff of three N species (dissolved organic N (DON), $NH_4^+$, and $NO_3^-$) introduced in our

previous study (Lee et al., 2014), here we have added particulate organic N (PON) fluxes from terrestrial litter and soils
(Sect. 2.2.1). Lee et al (2014; 2019) provide further details on the terrestrial model.

The freshwater component FANSY receives N, P, and solids in multiple forms either from LM3 or from prescribed inputs (Sect. 3.1). It then simulates biogeochemical transformations and transport of each form of the nutrients and solids within streams, rivers, and lakes (Sect. 2.2).

**2.2 Freshwater component FANSY**

FANSY constituents of algae, nutrients, and solids in rivers and lakes are listed in Table 1 and described in Fig. 1. FANSY has 13 prognostic state variables and 5 diagnostic state variables. Inorganic suspended solid (ISS) is delivered from the terrestrial soil erosion process (Sect. 2.2.1). ISS dynamically interacts with benthic sediment inorganic solid (Sed) through deposition and resuspension processes. Primary interactions between SS and other model components are through the
140 shading effect of turbidity on algae growth (Sect. 2.2.2) and the sorption of $PO_4^{3-}$ to ISS as particulate inorganic P (PIP) (Sect. 2.2.4). Algae take up N and P, which is subsequently partitioned between organic and inorganic N and P pools via algae mortality (Sect. 2.2.2). Algae C ($C_{al}$) and algae P ($P_{al}$) are diagnosed from algal N ($N_{al}$) assuming the Redfield C:N:P ratio (Chapra, 1997; Redfield et al., 1963). Chlorophyll a (CHL) is derived using the photoacclimation model of Geider et al. (1997) to predict a CHL-to-C ratio ($r_{CHLC}$), and CHL is calculated from $r_{CHLC}$ and $C_{al}$. The 5 prognostic N variables contain
an oxidized and reduced dissolved inorganic forms ($NH_4^+$ and $NO_3^-$), as well as a dissolved and two particulate (suspended and sedimentary) organic forms (DON, PON, and benthic sediment N (SedN), Sect. 2.2.3). The 5 prognostic P variables include the same organic forms as for N variables (dissolved organic P (DOP), particulate organic P (POP), and benthic sediment P (SedP), Sec. 2.2.4). The other two prognostic P variables are dissolved and particulate inorganic forms ($PO_4^{3-}$ and PIP). FANSY does not distinguish between $PO_4^{3-}$, DIP, and soluble reactive phosphorus (SRP). The subsections that follow
(Sect. 2.2.1-2.2.4) provide a detailed description of each variable and associated processes.

| Variable | Symbol |
|---|---|
| Prognostic variable | |
| Inorganic suspended solids | ISS |
| Benthic sediment inorganic solids | Sed |
| Algae nitrogen | $N_{al}$ |
| Ammonium nitrogen | $NH_4^+$ |
| Nitrate nitrogen | $NO_3^-$ |
| Phosphate phosphorus (dissolved inorganic phosphorus or soluble reactive phosphorus) | $PO_4^{3-}$ (DIP or SRP) |
| Particulate organic nitrogen | PON |

| Benthic sediment nitrogen | SedN |
|---|---|
| Dissolved organic nitrogen | DON |
| Particulate organic phosphorus | POP |
| Benthic sediment phosphorus | SedP |
| Dissolved organic phosphorus | DOP |
| Particulate inorganic phosphorus | PIP |
| Diagnostic variable | |
| Particulate organic matter (detritus or nonliving organic suspended solids) | POM |
| Suspended solids | SS |
| Algae phosphorus | $P_{al}$ |
| Algae carbon | $C_{al}$ |
| Chlorophyll a | CHL |

**Table 1: FANSY prognostic and diagnostic state variables.**

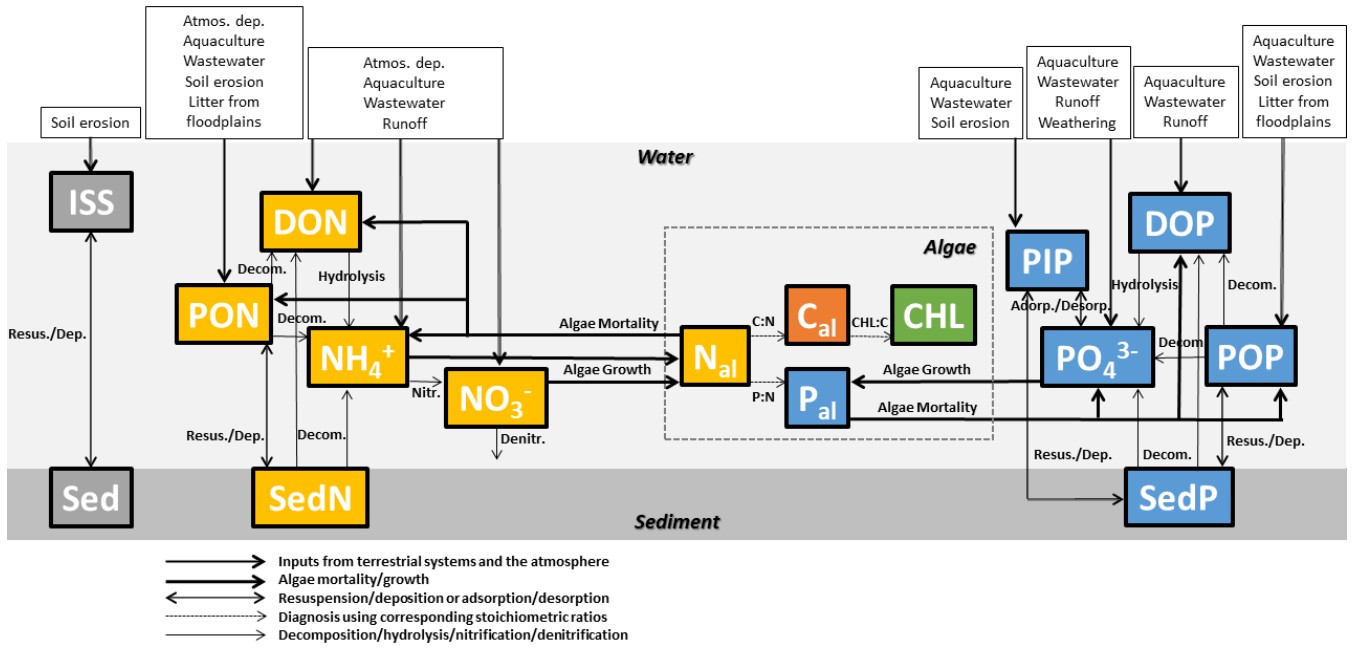

**Figure 1: FANSY structure with arrows depicting fluxes of constituents of algae, nutrients, and solids in rivers and lakes. The constituents are listed in Table 1.**

Added solids and nutrients to streams and rivers are subject to retention within rivers and lakes or transformed during transport to the coastal ocean. Each model grid cell contains one river reach and/or one lake. Water containing solids and

nutrients in each river reach or lake flows to another river reach in the downstream grid cell following a network that ultimately discharges to the ocean (Milly et al., 2014). Freshwater physics, hydrology, and hydrography are described in detail elsewhere (Milly et al., 2014). In each river reach or lake, for each prognostic variable i, Settling/resuspension dynamics and/or Biogeochemical reactions ($SB_i$) are calculated according to the process-based formulations described in the following subsections 2.2.1-2.2.4. For example, if i is PON, $SB_i$ is Eq. (28). If i is SedN, $SB_i$ is Eq. (29). A general mass balance for a variable i in a river reach or lake at each computation time step (30 minutes in this study) is written as:

$$\frac{dX_i}{dt} = F_i^{in} - F_i^{out} + I_i + SB_i \text{ , if i is a river/lake water column variable, i.e., i = all variables except Sed, SedN, or SedP} \tag{1}$$

$$\frac{dX_i}{dt} = SB_i \text{ , } \qquad\qquad \text{if i is a river/lake benthic sediment variable, i.e., i = Sed, SedN, or SedP} \tag{2}$$

where i is a prognostic variable listed in Table 1, $X_i$ is the amount of variable i (kg), $F_i^{in}$ and $F_i^{out}$ are inflow and outflow of variable i (kg s$^{-1}$) through the river/lake network, $I_i$ is inputs of variable i from terrestrial systems and/or the atmosphere (kg s$^{-1}$), and $SB_i$ is settling/resuspension dynamics and/or biogeochemical reactions of variable i (kg s$^{-1}$).

### 2.2.1 Solid dynamics

In LM3-FANSY, terrestrial soil erosion is controlled by land surface slope, rainfall, and leaf area index (LAI) based on Pelletier (2012), as described in Eq. (3). N fluxes from terrestrial litter and soils, in the form of PON, in Eq. (4) are based on the simulated soil erosion fluxes and litter/soil N concentrations. This approach is consistent with that employed by several previous modeling studies (Tian et al., 2015; Zhang et al., 2022). The litter/soil concentrations for this purpose are estimated by using litter/soil contents and effective soil depths simulated by LM3 (Gerber et al., 2010).

Inorganic soil inputs to rivers are derived from the simulated soil erosion fluxes by subtracting the PON contribution, as described in Eq. (5). This requires an assumed ratio of POM:PON in eroded soils (i.e., 13.9, Table 2). Previous studies have shown a wide range of C content in tree biomass (~42-61%, Thomas and Martin, 2012) and of C:N ratios in litter and soils (~5-500, Gerber et al., 2010 and references in Gerber et al., 2010's Table S1). This implies that POM:PON ratios in soil erosion fluxes can also vary significantly. We have found, however, that predicted river SS loads are insensitive to an order of magnitude variation in the ratio (i.e., 1.39 vs. 139, see Sect. 4.4), because organic contents in eroded soils are generally small. We thus used the POM:PON ratio of 13.9. The same ratio has been used to estimate the contribution of PON to SS in freshwaters, again noting that it is generally a small fraction of SS.

$$E = C_1 \cdot \frac{\rho_b}{\rho_w} \cdot S^{5/4} \cdot r \cdot e^{-L} \text{ ,} \tag{3}$$

$$E^{PON} = E \cdot \left(\frac{N_{FL}+N_{SL}+N_{SS}}{h_s \cdot \rho_b}\right), \tag{4}$$

$$E^{ISS} = E - r_{DN,Ero} \cdot E^{PON} \tag{5}$$

where E, $E^{PON}$, and $E^{ISS}$ is terrestrial soil erosion flux (kgDry matter (D) $m^{-2}$ $s^{-1}$), terrestrial PON flux (kgN $m^{-2}$ $s^{-1}$), and terrestrial inorganic soil erosion flux (kgD $m^{-2}$ $s^{-1}$) respectively, $C_1$ is a free parameter of terrestrial soil erosion (unitless), $\rho_b$ is soil bulk density (kgD $m^{-3}$), $\rho_w$ is water density (kg $m^{-3}$), S is slope $\tan\theta$, with $\theta$ as hillslope angle (unitless), r is rainfall (kg $m^{-2}$ $s^{-1}$), L is LAI (unitless), $N_{FL}$, $N_{SL}$, and $N_{SS}$ is N content in fast litter, slow litter, and slow soil pool respectively (kgN $m^{-2}$), $h_s$ is effective soil depth (m), and $r_{DN,Ero}$ is a POM-to-PON ratio in eroded fluxes (kgD kgN$^{-1}$).

Terrestrial soil erosion is known to be scale dependent, because it could be dominated by different spatial scale processes (e.g., interrill, rill, and gully erosion, landsliding; Poesen et al., 1996; Renschler and Harbor, 2002). We thus adapt from Pelletier (2012) a degree of freedom via the coefficient of $C_1$ to account for spatial resolution of input data (e.g., slope at 1 degree resolution). $C_1$ is a single global value scale parameter and coarsely calibrated to reduce prediction errors of SS loads across world major rivers (see Sect. 2.2.5 for details in model calibration). Sensitivity of the model to $C_1$ is addressed in Sect. 4.4. It has been suggested to model soil erosion at event scales (daily or subdaily time steps) to account for episodic, substantial mass transport (Tan et al., 2017). We calculate soil erosion rates at the model time step (30 minutes).

| Parameter | Description | Value | Unit | Reference/Rationale |
|---|---|---|---|---|
| Reported parameters | | | | |
| $T_{ref}$ | Reference temperature | 20 | °C | Chapra (1997) |
| $\kappa$ | Karman constant | 0.4 | unitless | Pelletier (2012) |
| R | Submerged specific gravity | 1.65 | unitless | Ferguson & Church (2004) |
| $\upsilon$ | Kinematic viscosity | $1\ 10^{-6}$ at 20 °C | $m^2\,s^{-1}$ | |
| $C_2$, $C_3$ | Reported constants in the settling velocity | 18, 1.0 | unitless | |
| $r_{CN}$ | Algae C-to-N ratio | 5.56 | kgC kgN$^{-1}$ | Chapra (1997), Redfield et al. (1963) |
| $r_{PN}$ | Algae P-to-N ratio | 0.14 | kgP kgN$^{-1}$ | |
| $\zeta$ | Cost of biosynthesis | 0.05 | $\zeta$ | Stock et al. (2014) |
| $c_{CHLC,max}$ | Maximum algae CHL-to-C ratio | 0.03 | kgCHL kgC$^{-1}$ | |
| $c_{CHLC,min}$ | Minimum algae CHL-to-C ratio | 0.002 | kgCHL kgC$^{-1}$ | |
| $s_{CHL1}$ | Algae self-shading factor | 0.0088 | L µgCHL$^{-1}$ m$^{-1}$ | Chapra (1997), Riley (1956) |
| $s_{CHL2}$ | Algae self-shading factor | 0.054 | L µgCHL$^{-2/3}$ m$^{-1}$ | |

| | | | | |
|---|---|---|---|---|
| $s_{ISS}$ | ISS light shading factor | 0.052 | L mgD$^{-1}$ m$^{-1}$ | Chapra (1997), Di Toro (1978) |
| $s_{POM}$ | POM light shading factor | 0.174 | L mgD$^{-1}$ m$^{-1}$ | |
| a | Reported fitted kinetic parameter in the $PO_4^{3-}$ sorption/desorption | 0.8 | mgP gSS$^{-1}$ | Garnier et al. (2005) |
| b | Reported fitted kinetic parameter in the $PO_4^{3-}$ sorption/desorption | 0.2 | unitless | |
| Calibrated parameters | | | | |
| Calibrated within broad reported ranges | | | | |
| $r_{DN,Ero}$ | POM-to-PON ratio in terrestrial erosion fluxes | 13.9 | kgD kgN$^{-1}$ | Gerber et al. (2010) and references in there, Thomas and Martin (2012) |
| $r_{DN}$ | POM-to-PON ratio in freshwaters | 13.9 | kgD kgN$^{-1}$ | Chapra (1997), Redfield et al. (1963) |
| d | Grain diameter | 0.01 | m | Ferguson & Church (2004) |
| $P_{max}^C$ | Maximum photosynthesis rate | 6.0 10$^{-5}$ | s$^{-1}$ | Geider et al. (1997) |
| $\alpha^{CHL}$ | CHL-specific initial slope of the photosynthesis-light curve | 1.0 10$^{-5}$ | gC m$^2$ gCHL$^{-1}$ μmolPhotons$^{-1}$ | |
| θ | Temperature correction factor for all processes except those for algae and sediment dynamics | 1.066 | unitless | Bowie et al. (1985), Chapra (1997), Eppley (1972) |
| $\theta_{Al}$ | Temperature correction factor for algae dynamics | 1.08 | unitless | |
| $\theta_{Sed}$ | Temperature correction factor for benthic sediment dynamics | 1.08 | unitless | |
| $k_{NO_3^-}$ | $NO_3^-$ half-saturation constant for algae growth | 0.1 | mgN L$^{-1}$ | |
| $k_{NH_4^+}$ | $NH_4^+$ half-saturation constant for algae growth | 0.02 | mgN L$^{-1}$ | |
| $k_{PO_4^{3-}}$ | $PO_4^{3-}$ half-saturation constant for algae growth | 0.002 | mgP L$^{-1}$ | |
| $k_{ew}$ | Light extinction due to particle- | 0.05 | m$^{-1}$ | |

| | | | | |
|---|---|---|---|---|
| | free water and color | | | |
| $k_{SedN,d}$ | SedN decomposition rate coefficient | 0.001/sperd | $s^{-1}$ | |
| $k_{SedP,d}$ | SedP decomposition rate coefficient | 0.001/sperd | $s^{-1}$ | |
| $k_{PON,d}$ | PON decomposition rate coefficient | 0.001/sperd | $s^{-1}$ | |
| $k_{POP,d}$ | POP decomposition rate coefficient | 0.001/sperd | $s^{-1}$ | |
| $k_{DON,d}$ | DON hydrolysis rate coefficient | 0.2/sperd | $s^{-1}$ | |
| $k_{DOP,d}$ | DOP hydrolysis rate coefficient | 0.01/sperd | $s^{-1}$ | |
| $k_{nitr}$ | Nitrification rate coefficient | 0.4/sperd | $s^{-1}$ | |
| $k_{denitr}$ | Denitrification rate coefficient | 0.15/sperd | $s^{-1}$ | |
| Calibrated to recreate measurement-based estimates | | | | |
| $C_1$ | Free parameter of terrestrial soil erosion | 0.012 | unitless | Pelletier (2012) |
| $k_m$ | Algae mortality rate | 0.8 $10^{-5}$ | $kgN^{-1/3}\,s^{-1}$ | Dunne et al. (2005) |
| $f_{m,DON}$ | Fraction of algae mortality which is deposited to the DON pool | 0.3 | unitless | |
| $f_{m,PON}$ | Fraction of algae mortality which is deposited to the PON pool | 0.3 | unitless | |
| $f_{SedP,POP}$ | Fraction of SedP resuspension which is deposited to the POP pool | 0.4 | unitless | Bowie et al. (1985), Chapra (2008) |
| $f_{PON,DON}$ | Fraction of PON decomposition which is deposited to DON pool | 0.8 | unitless | |
| $f_{POP,DOP}$ | Fraction of POP decomposition which is deposited to DOP pool | 0.8 | unitless | |
| $f_{SedN,DON}$ | Fraction of SedN decomposition which is deposited to DON pool | 0.8 | unitless | |

| $f_{SedP,DOP}$ | Fraction of SedP decomposition which is deposited to DOP pool | 0.8 | unitless | |

**Table 2: Model parameters, their descriptions, values, units, and references/rationale. "sperd" is seconds per days (86400).**

Once introduced from the land model to the river and lake systems, particulate solids and nutrients (i.e., ISS, PON, and POP) are subject to either deposition or suspension based on a Rouse number-dependent criterion, defined as settling velocity divided by the von Karman constant and shear velocity (Pelletier, 2012).

$$R_\# = \frac{w_S}{\kappa \cdot u_*} = \frac{w_S}{\kappa \sqrt{g \cdot z \cdot S}} , \tag{6}$$

where $R_\#$ is Rouse number (unitless), $w_S$ is settling velocity (m s$^{-1}$), $\kappa$ is Karman constant (unitless), $u_*$ is shear velocity (m s$^{-1}$), g is acceleration due to gravity (m s$^{-2}$), and z is river or lake depth (m). If Rouse number is less than 1.2 (a reported value in Pelletier, 2012), any newly introduced particulate matter, as well as those already in benthic sediments (i.e., SedN, SedP, and Sed), are suspended into the water column and subject to transport through the river network. Otherwise, the particulate matter is deposited to the benthic sediments. While organic material deposited to the sediments is assumed to be

remineralized over time (i.e., no net long-term burial), PIP is subject to a long-term burial in areas where resuspension does not occur.

Settling velocity ($w_s$) is estimated as a function of grain diameter, fluid viscosity, and fluid and solid density (Ferguson and Church, 2004).

$$w_S = \frac{R \cdot g \cdot d^2}{C_2 \cdot \upsilon + (0.75 \cdot C_3 \cdot R \cdot g \cdot d^3)^{0.5}} , \tag{7}$$

where R is submerged specific gravity (unitless), d is particle diameter (m), $\upsilon$ is kinematic viscosity of the fluid (m$^2$ s$^{-1}$), and $C_2$ and $C_3$ are reported constants (unitless). For this initial FANSY implementation, a characteristic grain diameter (d) is assumed for all particulate material sinks.

For a batch river and lake system, settling and resuspension dynamics for ISS and Sed are written as:

$$SB_{ISS} = \begin{cases} \frac{Sed}{dt} & R_\# < 1.2 \\ -\frac{w_S}{z}\left(\frac{1}{dt} + \frac{w_S}{z}\right)^{-1}\frac{ISS}{dt} & R_\# \geq 1.2 \end{cases} , \tag{8}$$

$$SB_{Sed} = \begin{cases} -\frac{Sed}{dt} & R_\# < 1.2 \\ \frac{w_S}{z}\left(\frac{1}{dt} + \frac{w_S}{z}\right)^{-1}\frac{ISS}{dt} & R_\# \geq 1.2 \end{cases} , \tag{9}$$

where ISS is inorganic suspended solid (kgD) and Sed is benthic sediment inorganic solid (kgD). ISS is gained by Sed resuspension and lost by deposition. The opposite holds for Sed. ISS deposition is modeled by implicitly solving for the ISS mass flux to Sed via $w_S$ divided by a river or lake depth and multiplied by an ISS mass in the water column. This implicit scheme reduces the numerical burden and improves stability.

The conversion of PON to POM in freshwaters for the purpose of calculating SS to compare with observations is as:

$$POM = r_{DN} \cdot PON , \tag{10}$$

$$SS = ISS + POM , \tag{11}$$

where POM, PON, and SS is particulate organic matter (kgD), particulate organic N (kgN), and suspended solid (kgD) respectively, and $r_{DN}$ is a POM-to-PON ratio in freshwaters (kgD kgN$^{-1}$).

### 2.2.2 Algae dynamics

Algae dynamics are governed by the balance of net growth (i.e., gross growth – respiration) and generalized mortality (i.e., non-predatory mortality + grazing + settling + excretion). The net growth is the difference between gross photosynthesis and respiration. The generalized mortality may include contributions from grazing, viruses, cell death, and excretion, though these diverse contributions are ultimately parameterized as a simple density-dependent loss term (Dunne et al., 2005). For a batch river and lake system, biogeochemical reactions for algae are written as:

$$SB_{N_{al}} = \mu(I_{av}, T, N, P) \cdot N_{al} - m(T) , \tag{12}$$

where $N_{al}$ is algae N (kgN), $\mu(I_{av}, T, N, P)$ is algae net growth rate (s$^{-1}$) as a function of euphotic zone averaged irradiance $I_{av}$, temperature (T), N, and P, and m(T) is generalized algae mortality (kgN s$^{-1}$) as a function of T.

A dynamic regulatory model is adapted to predict a CHL-to-C ratio ($r_{CHLC}$) and net growth rate ($\mu$) as a function of euphotic zone averaged irradiance, temperature, and nutrients (Geider et al., 1997). The $\mu$ is the difference between photosynthesis and respiration rates, as represented in Eq. (13). The $r_{CHLC}$ is up- and down-regulated in accordance with light and nutrient conditions according to Eq. (14).

$$\mu(I_{av}, T, N, P) = \frac{P_m^C}{1+\zeta} \cdot \left[ 1 - \exp\left( \frac{-\alpha^{CHL} \cdot I_{av} \cdot r_{CHLC}}{P_m^C} \right) \right] , \tag{13}$$

$$r_{CHLC} = \max\left[ r_{CHLC,min}, \frac{r_{CHLC,max}}{1 + \left( \frac{r_{CHLC,max} \cdot \alpha^{CHL} \cdot I_{av}}{2 \cdot P_m^C} \right)} \right] , \tag{14}$$

where $P_m^C$ is C-specific, light-saturated photosynthesis rate (s$^{-1}$), $\zeta$ is cost of biosynthesis, $\alpha^{CHL}$ is CHL-specific initial slope of the photosynthesis-light curve (gC m$^2$ gCHL$^{-1}$ µmolPhotons$^{-1}$), $I_{av}$ is euphotic zone averaged irradiance (µmolPhotons m$^{-2}$

$s^{-1}$), $r_{CHLC}$ is algae CHL-to-C ratio (kgCHL kgC$^{-1}$), and $r_{CHLC,min}$ and $r_{CHLC,max}$ are minimum and maximum algae CHL-to-C ratios (kgCHL kgC$^{-1}$).

The C-specific, light-saturated photosynthesis rate ($P_m^C$) is calculated as a function of temperature and nutrient limitation, also following the approach of Geider et al. (1997).

$$P_m^C(T, N) = P_{max}^C(T) \cdot min\left[\left(lim_{NO_3^-} + lim_{NH_4^+}\right), lim_{PO_4^{3-}}\right], \tag{15}$$

where $P_{max}^C(T)$ is temperature-dependent maximum photosynthesis rate ($s^{-1}$), $lim_{NO_3^-}$, $lim_{NH_4^+}$, and $lim_{PO_4^{3-}}$ are $NO_3^-$, $NH_4^+$, and $PO_4^{3-}$ limitations (unitless).

In LM3-FANSY, freshwater biogeochemical reaction rates approximately double for a temperature increase of 10°C based on the Arrhenius equation, with a scaling factor θ (Chapra, 1997; Eppley, 1972). The simulated maximum and minimum water temperatures are limited to 30°C and -3°C respectively.

$$P_{max}^C(T) = P_{max}^C \cdot \theta_{Al}{}^{T-T_{ref}}, \tag{16}$$

where T is temperature (°C), $T_{ref}$ is reference temperature (°C), $P_{max}^C$ is maximum photosynthesis rate at $T_{ref}$ ($s^{-1}$), and $\theta_{Al}$ is temperature correction factor for algae dynamics.

To combine the limiting effects of nutrients N and P, Liebig's law of the minimum is used. A $NH_4^+$ preference factor is used to account for inhibition of $NO_3^-$ uptake when $NH_4^+$ concentrations are high compared to a $NH_4^+$ half-saturation constant (Frost and Franzen, 1992). A saturating Monod relationship is used for handling the $NH_4^+$ and $PO_4^{3-}$ limiting effects. The maximum $NO_3^-$, $NH_4^+$, and $PO_4^{3-}$ concentrations are limited to 10 moles L$^{-1}$ to avoid numerical issues that can arise under extremely dry conditions.

$$lim_{NO_3^-} = \frac{[NO_3^-]}{\left(k_{NO_3^-} + [NO_3^-]\right) \cdot \left(1 + \frac{[NH_4^+]}{k_{NH_4^+}}\right)}, \tag{17}$$

$$lim_{NH_4^+} = \frac{[NH_4^+]}{k_{NH_4^+} + [NH_4^+]}, \tag{18}$$

$$lim_{PO_4^{3-}} = \frac{[PO_4^{3-}]}{k_{PO_4^{3-}} + [PO_4^{3-}]}, \tag{19}$$

where $[NO_3^-]$, $[NH_4^+]$, and $[PO_4^{3-}]$ are $NO_3^-$, $NH_4^+$, and $PO_4^{3-}$ concentrations (mgN L$^{-1}$ and mgP L$^{-1}$) and $k_{NO_3^-}$, $k_{NH_4^+}$, and $k_{PO_4^{3-}}$ are $NO_3^-$, $NH_4^+$, and $PO_4^{3-}$ half-saturation constants for algae growth (mgN L$^{-1}$ and mgP L$^{-1}$).

Photosynthetically available, visible irradiance at the surface is used for algae growth dynamics. Light attenuation with depth is modeled by the Beer-Lambert law using an extinction coefficient ($k_e$, Chapra, 1997). The euphotic zone (depth where light intensity falls to one percent of that at the surface) averaged light level ($I_{av}$) is used. The extinction coefficient is estimated dynamically to account for temporal and spatial variations in turbidity due to algae shading (Chapra, 1997; Riley, 1956), light extinction due to particle-free water and color, and variations in ISS and POM (Chapra, 1997; Dio Toro, 1978).

$$I_z = I_s \cdot e^{-k_e z} , \tag{20}$$

$$z_{0.01} = -\frac{\ln(0.01)}{k_e} , \tag{21}$$

$$I_{av} = \begin{cases} \frac{I_s}{k_e \cdot z}\left(1 - e^{-k_e \cdot z}\right) & z \leq z_{0.01} \\ \frac{I_s}{k_e \cdot z_{0.01}}\left(1 - e^{-k_e \cdot z_{0.01}}\right) & z > z_{0.01} \end{cases} , \tag{22}$$

$$k_e = k_{ew} + s_{ISS} \cdot [ISS] + s_{POM} \cdot [POM] + s_{CHL1} \cdot [CHL] + s_{CHL2} \cdot [CHL]^{2/3} , \tag{23}$$

where $I_z$ and $I_s$ are irradiance at z and at the surface ($\mu molPhotons\ m^{-2}\ s^{-1}$), $z_{0.01}$ is river or lake depth where light intensity falls to one percent of that at the surface (m), $k_e$ is light extinction coefficient ($m^{-1}$), $k_{ew}$ is light extinction due to particle-free water and color ($m^{-1}$), $s_{CHL1}$ and $s_{CHL2}$ are algae self-shading factors (L $\mu gCHL^{-1}\ m^{-1}$ and L $\mu gCHL^{-2/3}\ m^{-1}$), $s_{ISS}$ and $s_{POM}$ are constants accounting for the shading impacts of ISS and POM (L $mgD^{-1}\ m^{-1}$), and [ISS], [POM], and [CHL] are ISS, POM, and CHL concentrations ($mgD\ L^{-1}$ and $\mu gCHL\ L^{-1}$).

Biomass-specific algal mortality is assumed to increase non-linearly with algae concentration, reflecting a presumed increase in predators with algal prey (e.g., Steele and Henderson, 1992).

$$m(T) = k_m(T) \cdot Na^{4/3} , \tag{24}$$

where $k_m$ is temperature-dependent algae mortality rate ($kgN^{-1/3}\ s^{-1}$) reflecting the combined impacts of zooplankton grazing and other phytoplankton loss terms (e.g., viral-induced losses, cell death). Exponents between 4/3 and 2 have been commonly applied in this relationship, with higher values corresponding to more tightly coupled top-down control (Dunne et al., 2005). We have adopted a value of 4/3 to enable high biomass in nutrient rich environments. The Arrhenius relationship with the same scaling as algae growth is applied to account for the effect of temperature on algae mortality (Eq. 16). The division of algal mortality between inorganic/organic and dissolved/particulate nutrient pools is described in the following sections.

Algae P, C, and CHL are diagnosed from algae N using the Redfield C:N:P ratio (Chapra, 1997; Redfield et al., 1963) and $r_{CHLC}$ estimated above based on Geider et al. (1997).

$$P_{al} = N_{al} \cdot r_{PN} , \tag{25}$$

$$C_{al} = N_{al} \cdot r_{CN}, \tag{26}$$

$$CHL = C_{al} \cdot r_{CHLC}, \tag{27}$$

where $P_{al}$, $C_{al}$, and CHL are algae P (kgP), C (kgC), and CHL (kgCHL) and $r_{PN}$ and $r_{CN}$, are algae P-to-N (kgP kgN$^{-1}$) and C-to-N (kgC kgN$^{-1}$) ratios.

### 2.2.3 N dynamics

For a batch river and lake system, settling/resuspension dynamics and biogeochemical reactions are written for PON and SedN as:

$$SB_{PON} = \begin{cases} f_{m,PON} \cdot m(T) - k_{PON,d}(T) \cdot PON + \frac{SedN}{dt} & R_{\#} < 1.2 \\ f_{m,PON} \cdot m(T) - k_{PON,d}(T) \cdot PON - \frac{w_S}{z}\left(\frac{1}{dt} + \frac{w_S}{z}\right)^{-1} \frac{PON}{dt} & R_{\#} \geq 1.2 \end{cases}, \tag{28}$$

$$SB_{SedN} = \begin{cases} -k_{SedN,d}(T) \cdot SedN - \frac{SedN}{dt} & R_{\#} < 1.2 \\ -k_{SedN,d}(T) \cdot SedN + \frac{w_S}{z}\left(\frac{1}{dt} + \frac{w_S}{z}\right)^{-1} \frac{PON}{dt} & R_{\#} \geq 1.2 \end{cases}, \tag{29}$$

where PON is particulate organic N (kgN), SedN is benthic sediment N (kgN), $f_{m,PON}$ is fraction of algae mortality which is deposited to the PON pool (unitless), and $k_{PON,d}(T)$ and $k_{SedN,d}(T)$ are temperature-dependent PON and SedN decomposition rates (s$^{-1}$).

In FANSY, PON is gained by algae mortality and benthic sediment N (i.e., SedN) resuspension and lost by deposition and decomposition. The same holds for SedN, except that it does not receive inputs from algae mortality. A fraction ($f_{m,PON}$) is adapted from Chapra (2008) to represent the portion of algae mortality released as PON. First-order kinetics are used to describe various decay processes and transformations. PON and SedN are lost by decay processes that breakdown complex organic compounds into simpler organic N (i.e., DON) or into NH$_4^+$. Rate coefficients for these decay processes are thus much smaller than those for release of NH$_4^+$ due to DON decay (i.e., hydrolysis), oxidation of NH$_4^+$ to NO$_3^-$ (i.e., nitrification), and reduction of NO$_3^-$ to N$_2$ (i.e., denitrification) (Chapra, 2008). The Arrhenius-based relationship (Eq. 16) is used to adjust the rate coefficients for temperature effects with different temperature correction factors of $\theta_{Sed}$ for sediment dynamics and $\theta$ for all dynamics except algae and sediment dynamics.

In FANSY, DON is gained by algae mortality and decomposition of PON and SedN and lost by hydrolysis. A fraction ($f_{m,DON}$) is adapted from Chapra (2008) to represent the portion of algae mortality released as DON. Decomposition of PON and SedN releases both dissolved organic and inorganic N (i.e., DON and NH$_4^+$). Fractions ($f_{PON,DON}$ and $f_{SedN,DON}$) are adapted from Bowie et al. (1985) to partition the fluxes to DON and NH$_4^+$.

$$SB_{DON} = f_{m,DON} \cdot m(T) + f_{PON,DON} \cdot k_{PON,d}(T) \cdot PON + f_{SedN,DON} \cdot k_{SedN,d}(T) \cdot SedN - k_{DON,d}(T) \cdot DON, \qquad (30)$$

where DON is dissolved organic N (kgN), $f_{m,DON}$ is fraction of algae mortality which is deposited to the DON pool (unitless), $f_{PON,DON}$ and $f_{SedN,DON}$ are fractions of PON and SedN decomposition fluxes which are deposited to the DON pool (unitless), and $k_{DON,d}(T)$ is temperature-dependent DON hydrolysis rate (s$^{-1}$).

In FANSY, NH$_4^+$ and NO$_3^-$ are removed by algae uptake during photosynthesis. NH$_4^+$ is returned to the water column
through soluble excretions of algae (which is included in the generalized algae mortality term) and decomposition/hydrolysis of SedN, PON, and DON. Removal of NH$_4^+$ by nitrification generates NO$_3^-$, which is in turn lost by denitrification.

$$SB_{NH_4^+} = \left(1 - f_{m,PON} - f_{m,DON}\right) \cdot m(T) + \left(1 - f_{PON,DON}\right) \cdot k_{PON,d}(T) \cdot PON + \left(1 - f_{SedN,DON}\right) \cdot k_{SedN,d}(T) \cdot SedN +$$
$$k_{DON,d}(T) \cdot DON - k_{nitr}(T) \cdot NH_4^+ - f_{NH_4^+,up}\mu(I_{av}, T, N, P) \cdot N_{al}, \qquad (31)$$

$$SB_{NO_3^-} = k_{nitr}(T) \cdot NH_4^+ - k_{denitr}(T) \cdot NO_3^- - \left(1 - f_{NH_4^+,up}\right) \cdot \mu(I_{av}, T, N, P) \cdot N_{al}, \qquad (32)$$

$$f_{NH_4^+,up} = \left(\frac{lim_{NH_4^+}}{lim_{NO_3^-} + lim_{NH_4^+}}\right), \qquad (33)$$

where NH$_4^+$ and NO$_3^-$ are ammonium and nitrate N (kgN), $k_{nit}(T)$ and $k_{denit}(T)$ are temperature-dependent nitrification and denitrification rates (s$^{-1}$), and $f_{NH_4^+,up}$ is fraction of NH$_4^+$ uptake for algae growth (unitless).

### 2.2.4 P dynamics

Overall, P dynamics are similar to those of N, but with several differences. Because there are two suspended particulate
forms of POP and PIP, SedP resuspension is divided into POP and PIP pools with a fraction of $f_{SedP,POP}$. Unlike N, P does not exist in a gaseous form, and FANSY includes no loss term for P to the atmosphere. Dissolved inorganic P sorbs strongly to solid particles. The exchange of PO$_4^{3-}$ between the dissolved and particulate forms are modeled based on Freundlich kinetics (Garnier et al., 2005; Nemery, 2003), with the flux estimated as the disequilibrium between the two phases.

$$SB_{POP} = \begin{cases} r_{PN} \cdot f_{m,PON} \cdot m(T) - k_{POP,d}(T) \cdot POP + f_{SedP,POP} \cdot \frac{SedP}{dt} & R_\# < 1.2 \\ r_{PN} \cdot f_{m,PON} \cdot m(T) - k_{POP,d}(T) \cdot POP - \frac{w_S}{z}\left(\frac{1}{dt} + \frac{w_S}{z}\right)^{-1}\frac{POP}{dt} & R_\# \geq 1.2 \end{cases}, \qquad (34)$$

$$SB_{SedP} = \begin{cases} -k_{SedP,d}(T) \cdot SedP - \frac{SedP}{dt} & R_\# < 1.2 \\ -k_{SedP,d}(T) \cdot SedP + \frac{w_S}{z}\left(\frac{1}{dt} + \frac{w_S}{z}\right)^{-1}\left(\frac{POP}{dt} + \frac{PIP}{dt}\right) & R_\# \geq 1.2 \end{cases}, \qquad (35)$$

$$SB_{PIP} = \begin{cases} F_{PO_4^{3-}\_to\_PIP} + \left(1 - f_{SedP,POP}\right)\frac{SedP}{dt} & R_\# < 1.2 \\ F_{PO_4^{3-}\_to\_PIP} - \frac{w_S}{z}\left(\frac{1}{dt} + \frac{w_S}{z}\right)^{-1}\frac{PIP}{dt} & R_\# \geq 1.2 \end{cases}, \qquad (36)$$

$$SB_{DOP} = r_{PN} \cdot f_{m,DON} \cdot m(T) + f_{POP,DOP} \cdot k_{POP,d}(T) \cdot POP + f_{SedP,DOP} \cdot k_{SedP,d}(T) \cdot SedP - k_{DOP,d}(T) \cdot DOP, \quad (37)$$

$$SB_{PO_4^{3-}} = r_{PN} \cdot (1 - f_{m,PON} - f_{m,DON}) \cdot m(T) + (1 - f_{POP,DOP}) \cdot k_{POP,d}(T) \cdot POP + (1 - f_{SedP,DOP}) \cdot k_{SedP,d}(T) \cdot SedP +$$

$$k_{DOP,d}(T) \cdot DOP - r_{PN} \cdot \mu(I_{av}, T, N, P) \cdot Na - F_{PO_4^{3-}\_to\_PIP}, \quad (38)$$

$$[PIP_{eq}] = a \cdot [PO_4^{3-}]^b \cdot [ISS]_2, \quad (39)$$

$$F_{PO_4^{3-}\_to\_PIP} = ([PIP_{eq}] - [PIP]) \cdot H_2O \cdot 10^{-3}, \quad (40)$$

where POP is particulate organic P (kgP), SedP is benthic sediment P (kgP), PIP is particulate inorganic P (kgP), DOP is dissolved organic P (kgP), $PO_4^{3-}$ is phosphate (kgP), $H_2O$ is water volume (m$^3$), $k_{POP,d}(T)$ and $k_{SedP,d}(T)$ are temperature-dependent POP and SedP decomposition rates (s$^{-1}$), $k_{DOP,d}(T)$ is temperature-dependent DOP hydrolysis rate (s$^{-1}$), $f_{SedP,POP}$

is fraction of SedP resuspension which is deposited to the POP pool (unitless), $f_{POP,DOP}$ and $f_{SedP,DOP}$ are fractions of POP and SedP decomposition fluxes which are deposited to the DOP pool (unitless), [PIP] and $[PO_4^{3-}]$ are PIP and $PO_4^{3-}$ concentrations (mgP L$^{-1}$), $[ISS]_2$ is ISS concentration (gD L$^{-1}$) (Notice the concentration unit difference from [ISS] in Eq. 23), $[PIP_{eq}]$ is PIP equilibrium concentration (mgP L$^{-1}$), $F_{PO_4^{3-}\_to\_PIP}$ is fluxes from $PO_4^{3-}$ to PIP (kgP), and a (mgP gSS$^{-1}$) and b (unitless) are reported empirical kinetic parameters.

## 2.2.5 Model calibration

Because many of the reported parameters required to simulate the coupled freshwater algae, nutrient, and solid dynamics within LM3-FANSY vary widely, it is difficult to assign a single global value for each parameter. Informed by parameter sensitivity analysis herein (Sect. 4.4), our approach was to coarsely calibrate a limited set of uncertain yet highly influential parameters within their broad observed ranges to reduce errors in simulated SS, N, and P loads in different forms across a

globally-distributed subset of large rivers.

First, as described above, the terrestrial soil erosion parameter $C_1$ was calibrated to reduce prediction errors of SS loads. We emphasize that this parameter is expected to be resolution dependent.

Second, the generalized algal mortality constant $k_m$ was tuned to produce reasonable chlorophyll a concentrations in globally distributed lakes, acknowledging the limitation of the present lake biogeochemistry in LM3-FANSY (see Sect. 4.5 for further discussion). We adapted an approach of Chapra (2008) to partition nutrient fluxes from algae mortality to different pools of nutrients in different forms (i.e., particulate organic, dissolved organic, and inorganic), based on fixed fractions. Uncertainties in these fixed factions due to the lack of theoretical and empirical evidence are investigated in the sensitivity

analysis (Sect. 4.4).

Third, we find that the rate coefficients for hydrolysis, nitrification, and denitrification are highly influential parameters for determining river nutrient loads in different forms, relative to those for decay processes that breakdown complex organic compounds into simpler organic and inorganic compounds. These highly influential parameters have been calibrated to reduce prediction errors of nutrient loads in different forms. We adapted an approach of Bowie et al (1985) to partition fluxes from complex organic nutrient decomposition to simpler organic vs. inorganic nutrient pools based on a uniform fraction. Uncertainties in the faction are investigated in the sensitivity analysis (Sect. 4.4).

## 3 Model forcing and simulations

### 3.1 Baseline simulations

LM3-FANSY was implemented globally at 1 degree spatial and 30 minute temporal resolution with all inputs regridded to 1 degree resolution. Following ~11,000 years of spin-up from Lee et al. (2019), the terrestrial component LM3 was run for the period 1700-1899 by recycling 30 years (1948-1977) of observation-based, historical climate forcing (Sheffield et al., 2006) and Coupled Model Intercomparison Project (CMIP6) datasets for atmospheric $CO_2$ (Meinshausen et al., 2017), atmospheric N deposition (CMIP6 Forcing Datasets Summary, 2023), and land-use states and transitions (Hurtt et al., 2020). Since the freshwater component requires a shorter time for equilibrium than vegetation and soil, the merged terrestrial and freshwater components LM3-FANSY were run for only the 1900-2000 period using additional CMIP6 datasets for fertilizer N applications (Hurtt et al., 2020) and reported N and P inputs to rivers (Beusen et al., 2015).

The observation-based, historical climate forcing data available for the period 1948-2010 (Sheffield et al., 2006) includes precipitation, specific humidity, air temperature, surface pressure, wind speed, and short- and long-wave downward radiation at 1 degree and 3 hour resolution. This forcing was cycled over a period of 30 years (1948–1977) to perform long-term simulations from 1700 to 1947, and the 1948–2000 forcing data were used for the simulations from 1948 to 2000. Annual atmospheric $CO_2$ estimates (Meinshausen et al., 2017) available for the period 1-2500 are used for the corresponding period simulation from 1700 to 2000. The atmospheric N deposition data (CMIP6 Forcing Datasets Summary, 2023) includes oxidized and reduced N ($NO_y$ and $NH_x$) at 2.5 longitude by 1.9 latitude degree and 1 month resolution for the period 1850-2099. The $NO_y$ and $NH_x$ deposition for the year 1850 was applied to soil $NO_3^-$ and $NH_4^+$ pools respectively for the 1700-1849 simulation, and then the 1850-2000 deposition was applied for the 1850-2000 simulation.

The dataset of land-use states and transitions and fertilizer N applications at 0.25 degree and 1 year resolution (Hurtt et al., 2020) is available for the period 850-2100. The 1700-2000 land-use state and transition data were used for the simulations from 1700 to 2000. Since the amount of fertilizer applications in the dataset is zero until 1915, the 1916-2010 fertilizer N was applied for the simulations from 1916 to 2000. For land use and fertilizer applications, 12 land-use types reported in the Land Use Harmonization (LUH2) (Hurtt et al., 2020) were grouped into 4 types in LM3-FANSY: 1) primary land in LM3-

FANSY is the sum of forested primary land and non-forested primary land in LUH2, 2) secondary land in LM3-FANSY is the sum of potentially forested secondary land, potentially non-forested secondary land, and urban land in LUH2, 3) cropland in LM3-FANSY is the sum of C3 annual cropland, C3 perennial cropland, C4 annual cropland, C4 perennial cropland, and C3 N-fixing cropland in LUH2, and 4) pasture in LM3-FANSY is the sum of managed pasture and rangeland in LUH2. The sum of fertilizers allocated to the 5 croplands in LUH2 was applied to the cropland in LM3-FANSY.

The terrestrial soil erosion component requires slope, rainfall, and LAI as inputs. Rainfall was simulated by using 3 hourly precipitation and temperature from Sheffield et al. (2006) and assuming that all of the precipitation falls as snow with the temperature less than 0°C, otherwise it is assumed to be rain. The slope input was derived from Danielson and Gesch (2011). The LAI input was from an observationally derived, monthly average global vegetation LAI dataset from Global Inventory Modeling and Mapping Studies (GIMMS) Normalized Difference Vegetation Index (NDVI3g) for the period 1982-2010 (Zhu et al., 2013) to avoid potential errors that might be caused by using modeled LAI. The 1982 LAI was used to perform long-term simulation from 1900 to 1981 and the 1982-2000 LAI was used for the simulations from 1982 to 2000.

Solid and nutrient inputs from terrestrial systems and from the atmosphere to rivers are either simulated by LM3-FANSY or provided by Beusen et al. (2015) (Table 3). For solids, all inputs were simulated by LM3-FANSY. For N, all inputs were simulated by LM3-FANSY except aquaculture, wastewater, and atmospheric deposition, which were provided by Beusen et al. (2015). For P, which is not currently included in LM3, all inputs were provided by Beusen et al. (2015).

Beusen et al. (2015) provided five-year interval data for the period 1900-2000 at 0.5 degree resolution. The data were regridded to our 1 degree resolution by summing up the values given in kg yr$^{-1}$ and linearly interpolated across the five-year intervals. Beusen et al. (2015)'s wastewater N and P inputs were from Morée et al. (2013)'s urban waste N and P discharge estimates to surface waters. Beusen et al. (2015) calculated aquaculture N and P inputs using Bouwman et al. (2013b)'s finfish and Bouwman et al. (2011)'s shellfish data. For atmospheric N deposition inputs, Beusen et al. (2015)'s input for the year 2000 was from Dentener et al. (2006)'s ensemble of reactive-transport models and those for the years before 2000 were made by scaling the deposition with Bouwman et al. (2013a)'s ammonia emissions. Beusen et al. (2015)'s surface runoff P inputs included those leached from soil P budgets (i.e., the sum of fertilizer and animal manure minus crop and grass withdrawal) and those driven by soil erosion estimates based on Cerdan et al. (2010). Beusen et al. (2015)'s P inputs of litter from floodplains were estimated as 50% of total NPP with a C:P ratio of 1200. Beusen et al. (2015)'s weathering P inputs were computed based on Hartmann et al. (2014)'s chemical weathering P release estimates.

We divided yearly total N (TN) and total P (TP) inputs from Beusen et al. (2015) into different N and P forms (listed in Table 3) based on Vilmin et al. (2018). Vilmin et al (2018) suggested fractions to divide TN and TP inputs into three N and P species according to the source. For sewage, fractions were given for three types (i.e., untreated, primary treated, and

secondary/tertiary treated). The sewage inputs from Beusen et al. (2015), however, were aggregated. Thus, to divide Beusen et al. (2015)'s sewage TN and TP inputs into three N and P species, we have taken a middle ground and used the fractions for primary treated sewage. Although we acknowledge that this is a simplification, we find that our results are relatively insensitive to alternatives, assuming that all sewage is untreated, all sewage has secondary treatment, and two options are based on that over 80% of wastewater is discharged without "adequate treatment" (Environment and Natural Resources Department, 2022, Table 3). Specifically, the fractions are driven by assuming that 1) 80%, 10%, and 10% of Beusen et al. (2015)'s sewage are untreated, primary treated, and secondary/tertiary treated respectively and 2) 40%, 40%, and 20% of Beusen et al. (2015)'s sewage are untreated, primary treated, and secondary/tertiary treated respectively. In all cases, the simulated river loads of each species change by $\leq 9\%$, and the simulated total loads does not change (see Sect. 4.4).

For TP fluxes from agricultural lands to rivers, distinct species fractions were given for two sources (i.e., surficial runoff and soil loss), while surface runoff TP inputs from Beusen et al. (2015) were aggregated. To divide Beusen et al. (2015)'s agricultural surface runoff TP inputs into three P species, we assume nearly equal fractions. As is the case for sewage, perturbation experiments show that our results are relatively insensitive to a wide range of the fractions (see Table 3 and the sensitivity analysis in Sect. 4.4). The 6 uncertainty simulations use 1) the fractions for surficial runoff for all agricultural fluxes, 2) the fractions for soil loss for all agricultural fluxes, fractions driven by assuming that 3) 40% and 60% of Beusen et al. (2015)'s agricultural surface runoff TP inputs are from surficial runoff and soil loss respectively, 4) 60% and 40% of those are from surficial runoff and soil loss respectively, 5) 20% and 80% of those are from surficial runoff and soil loss respectively, and 6) 80% and 20% of those are from surficial runoff and soil loss respectively. In all cases, the simulated river loads of each species change by $\leq 13\%$, and the simulated total load change by $\leq 1\%$.

For aquaculture, two groups of fractions are given for particulate and dissolved sources. The fractions in Table 3 are driven by assuming that about 12% and 31% of aquaculture TN and TP inputs from Beusen et al. (2015) are particulate, approximating Figure 3 of Bouwman et al (2013a). We did not conduct sensitivity simulations for aquaculture, because aquaculture nutrient inputs are very small compared to the other sources (< 1% of total inputs for both N and P), and thus uncertainties associated with the fractions should be negligible for the global application herein.

| Input source | N | | | P | | | Solid | |
|---|---|---|---|---|---|---|---|---|
| | $NH_4^+$ | $NO_3^-$ | ON | $PO_4^{3-}$ | PIP | OP | POM | ISS |
| Subsurface runoff | | | | | | | | |
| Litter from floodplains | LM3-FANSY | LM3-FANSY | LM3-FANSY | 0.00 | 0.00 | 1.00 | LM3-FANSY | LM3-FANSY |
| Natural surface runoff | | | | 0.00 | 0.25 | 0.75 | | |
| Agricultural surface runoff | | | | 0.34 | 0.33 | 0.33 | | |

| | | | | | | | | |
|---|---|---|---|---|---|---|---|---|
| Weathering | | | | 1.00 | 0.00 | 0.00 | | |
| Atmospheric deposition | 0.35 | 0.35 | 0.30 | | | | | |
| Aquaculture | 0.26 | 0.62 | 0.12 | 0.69 | 0.06 | 0.25 | | |
| Wastewater | 0.90 | 0.00 | 0.10 | 0.80 | 0.10 | 0.10 | | |
| Uncertainty tests associated with fractions that divide TP inputs into three P species (see Sect. 3.1 for details), assuming that agricultural surface runoff from Beusen et al (2015) contains the followings: | | | | | | | | |
| Surficial runoff 100% | | | | 0.60 | 0.00 | 0.40 | | |
| Soil loss 100% | | | | 0.00 | 0.75 | 0.25 | | |
| Surficial runoff 40% soil loss 60% | | | | 0.24 | 0.45 | 0.31 | | |
| Surficial runoff 60% soil loss 40% | | | | 0.36 | 0.30 | 0.34 | | |
| Surficial runoff 20% soil loss 80% | | | | 0.12 | 0.60 | 0.28 | | |
| Surficial runoff 80% soil loss 20% | | | | 0.48 | 0.15 | 0.37 | | |
| Uncertainty tests associated with fractions that divide TN and TP inputs into three N and P species (see Sect. 3.1 for details), assuming that wastewater from Beusen et al (2015) contains the followings: | | | | | | | | |
| Untreated | 0.65 | 0.00 | 0.35 | 0.15 | 0.30 | 0.55 | | |
| Secondary/tertiary treated 100% | 0.60 | 0.30 | 0.10 | 0.80 | 0.10 | 0.10 | | |
| Untreated 80% Primary treated 10% Secondary/tertiary treated 10% | 0.67 | 0.03 | 0.30 | 0.28 | 0.26 | 0.46 | | |
| Untreated 40% Primary treated 40% Secondary/tertiary treated 20% | 0.74 | 0.06 | 0.20 | 0.54 | 0.18 | 0.28 | | |

**Table 3: Description of solid and nutrient inputs from terrestrial systems and the atmosphere to rivers, which were either simulated by LM3-FANSY or provided by Beusen et al. (2015). The numbers are the fractions of dividing TN and TP inputs provided by Beusen et al. (2015) into different N and P species. Organic N (ON) and organic P (OP) from Beusen et al. (2015) are considered to be mainly (70%) particulate.**

### 3.2 Sensitivity simulations

Model sensitivities to the inputs, components, and parameters are examined by analyzing the responses of river solid and nutrient loads to their changes for the period 1948-2000. Model sensitivities to the nutrient inputs to rivers are examined by increasing each input source by 15% or removing it. When the inputs were from LM3, the 15% increase was applied to each

input variable (i.e., $NO_3^-$, $NH_4^+$, DON, and PON). When the inputs were externally prescribed, the 15% increase was applied to each input source listed in Table 3. In both cases, the increases were applied uniformly over the grid and time.


One of the distinct features of LM3-FANSY is the capability of modeling interactions between algae, nutrient, and solid dynamics. Light shading by SS and algae themselves modulates the strength of algal productivity and, in turn, river solids and nutrients. In LM3-FANSY, the light extinction coefficient is dynamically simulated as a function of ISS, POM, and CHL (Eq. 23), instead of using a prescribed parameter. To evaluate how critical the dynamic light extinction component is

for model predictive capacity, the component was replaced with a prescribed parameter value ($k_e$ = 0.15 or 0.45) and the river load responses to more active algal populations are examined.

Model sensitivities to the parameters are examined by decreasing each of all calibrated parameters listed in Table 2 by half or increasing it by twice. For the parameters which have much smaller or much larger impacts compared to the other

parameters, additional sensitivity tests have been performed to show responses of river loads to the parameter changes in their broad observed ranges.

### 3.3 Comparisons of measurement-based and modeled estimates

For cross-watershed evaluations, we compare LM3-FANSY results of river SS, $NO_3^-$, $NH_4^+$, DIN (the sum of $NO_3^-$ and $NH_4^+$), DON, total Kjeldahl N (TKN, the sum of $NH_4^+$, DON, and PON), $PO_4^{3-}$, DOP, and TP (the sum of $PO_4^{3-}$, DOP, PIP, and POP) yields (kg km$^{-2}$ yr$^{-1}$), loads (kt yr$^{-1}$), and concentrations (mg L$^{-1}$) with measurement-based estimates from 70 of the

world's major rivers (Table SI1-9, the GEMS-GLORI world river discharge database, Meybeck and Ragu, 2012). The 70 river basins cover 55% of global land area (excluding the Antarctic) and are distributed globally across various climates and land use (Fig. SI1). The LM3-FANSY performance is also compared with that of Global NEWS (Mayorga et al., 2010) by using the same measurement-based estimates, yet excluding a few unavailable rivers in Global NEWS (Table SI1, SI4, SI6-

9).

The 70 rivers were chosen by the following procedure. First, river basins with areas < 100,000 km$^2$, about 10 grid cells in our 1 degree resolution, were excluded from the comparisons. Second, river locations were identified by matching latitudes, longitudes, and basin areas of the modeled and reported rivers, which are located either at the river mouths or near the river

mouths. Third, rivers that were not properly represented within the LM3-FANSY river network were excluded. For example, the GEMS-GLORI database provides a Sanaga River SS concentration at 3.8°N and 10.1°E with its basin area 119,300 km$^2$. LM3-FANSY, however, does not capture the Sanaga River at a comparable location (3.5° N and 10.5° E), where the river has a much smaller basin area (5,607 km$^2$). The Sanaga River was thus excluded. In total, 9 rivers (i.e., Anabar, Brantas, Burdekin, Don, Hayes, Huai, Pyasina, Sanaga, and Sepik) were excluded. When a river in the LM3-FANSY river network

captures two merged small rivers in GEMS-GLORI, the water discharge weighted mean concentration of the two rivers was

used for analysis. When more than one data were given for a river, the data selected for the 1st line was used, since it was considered as "the most reliable and generally were obtained first hand by local engineers or scientists (Meybeck and Ragu, 1997)". When data in the 1st line was outdated (1970s-early 1980s) or reported as zero, the latest data was used if available in the next lines. When more than one data were given for a river with different basin areas, the data monitored at the location with the largest basin area (i.e., nearest to the river mouth) was used.

Since hydraulic controls like damming, irrigation, and diversion affect many rivers, natural river water discharges are distinguished from actual, modified ones in the GEMS-GLORI database. LM3-FANSY does not resolve such hydraulic controls and thus, if available, the natural discharges of GEMS-GLORI are used, when calculating loads and yields from the GEMS-GLORI's concentrations. Comparisons of the model results with loads and yields calculated by using the actual discharges are also presented in Supplementary Information. Since Global NEWS accounts for anthropogenic hydraulic controls, Global NEWS results are compared with the loads and yields calculated by using the actual discharges.

We report the Pearson correlation coefficient (r) and Nash–Sutcliffe model efficiency coefficient (NSE) between the log-transformed modeled and measurement-based estimates across the 70 rivers. We also report the prediction error computed as the difference between the modeled and measurement-based estimates of loads expressed as a percentage of the measurement-based load. For global evaluations, LM3-FANSY results are compared with reported global estimates from various references. For cross-watershed and global evaluations, annual results for the year 1990 are analyzed and presented. In parenthesis, ranges of using annual results for the years 1990-2000 are also provided throughout the manuscript.

For time series evaluations, we used reported annual solid and nutrient loads across 8 stations in large U. S. rivers from the USGS National Water Quality Network (NWQN, Lee, 2022). We note that the robustness of evaluating simulated interannual variability against simple flow-weighted annual observations depends on the frequency and timing of chemical samplings (Lee et al., 2021). The reported annual loads from NWQN were estimated with the USGS's latest load estimation method, WRTDS-K, for all analyzed rivers, except for the Columbia River for which only the REG method estimates are available (Lee et al., 2017). The WRTDS-K method was proved to be the most accurate annual load estimation method among methods studied by Lee et al. (2019).

The 8 stations are located in the 3 largest river basins in NWQN (6 stations in the Mississippi River Basin, 1 station in the St. Lawrence River Basin, and 1 station in the Columbia River Basin). The basin area of Yukon River station is larger than those of St. Lawrence and Columbia River stations, but the Yukon River station was excluded for this analysis, because data is only available beyond 2000. The corresponding station locations in the LM3-FANSY river network were identified by matching latitudes, longitudes, and basin areas of the reported and modeled rivers. The yearly data provided by NWQN is based on "a water year" defined as "the 12-month period from October 1 for any given year through September 30 of the

following year". The corresponding water yearly results of LM3-FANSY for the periods ~1963-2000 were used for this analysis.

## 4 Results and discussion

### 4.1 Model performance analysis

Measurement-based and simulated annual SS estimates across 65 rivers are significantly correlated, with Pearson correlation
coefficient, r values equal to 0.65 (0.57-0.65) for yields, 0.76 (0.71-76) for loads, and 0.67 (0.67-0.69) for concentrations for the year 1990 (range for the years 1990-2000) (Fig. 2, Table 4). This result, which allows for a coarsely calibrated value of the one free terrestrial soil erosion parameter ($C_1$=0.012), demonstrates that LM3-FANSY reproduces the measurement-based SS estimates fairly well, especially given that the model contains only one calibrated parameter for SS. This model performance is competitive with other global model estimates using larger numbers of free parameters (Hatono and
Yoshimura, 2020; Mayorga et al., 2010; Tan et al., 2017). For example, model performance of Global NEWS 2, when analyzed on the same dataset used for our model performance evaluation, yet excluding a few unavailable rivers (Table SI1), is slightly better than LM3-FANSY for yields and loads and slightly worse for concentrations based on correlations (Fig SI2). The total amount of global river SS loads to the coastal ocean is 10 (10-11) Pg yr$^{-1}$ for the year 1990 (range for the years 1990-2010) for LM3-FANSY, at the lower bond of previous estimates (Table 5, Global NEWS estimate of 11-27 Pg
yr$^{-1}$, Beusen et al., 2005; Discharge Relief Temperature sediment delivery model (QRT) estimate of 12.6 Pg yr$^{-1}$, Syvitski et al., 2005).

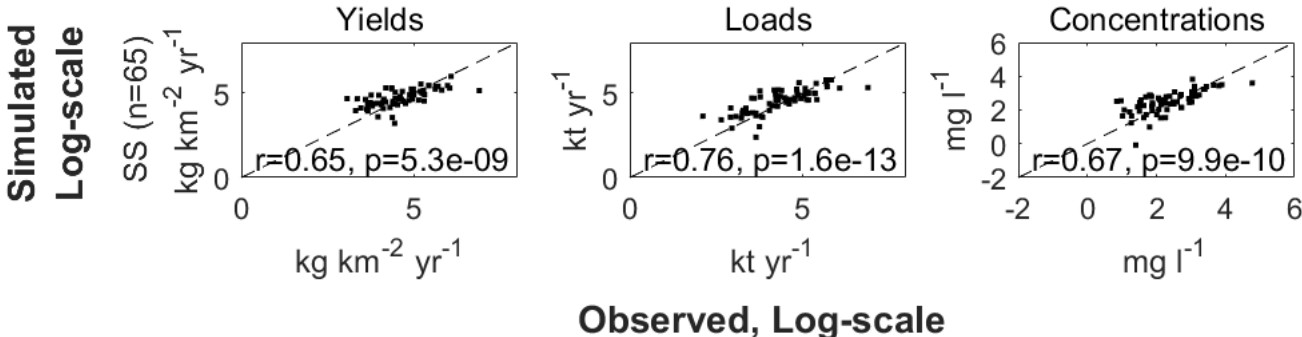

**Figure 2: Pearson correlation coefficients (r) and p values (p) between the log-transformed measurement-based vs.**
**simulated SS yields, loads, and concentrations across 65 rivers for the year 1990.**

|  | SS | NO$_3^-$ | NH$_4^+$ | DON | TKN | PO$_4^{3-}$ | DOP | TP |
|---|---|---|---|---|---|---|---|---|
| Model predictive capacity of spatial variation, r | .76 (.71, .76) | .79 (.75, .81) | .64 (.63, .72) | .85 (.85, .93) | .66 (.59, .72) | .70 (.67, .74) | .93 (.93, .95) | .98 (.95, .98) |

| | | | | | | | | | |
|---|---|---|---|---|---|---|---|---|---|
| Model efficiency, NSE | | .54 (.48, .55) | .43 (.33, .49) | .31 (.29, .45) | .66 (.62, .77) | .00 (.00, .21) | .49 (.42, .54) | .85 (.85, .88) | .82 (.81, .86) |
| Prediction errors | Min | -97 (-98, -97) | -86 (-92, -79) | -76 (-90, -76) | -75 (-76, -68) | -56 (-60, -24) | -98 (-99, -86) | -51 (-55, -47) | -64 (-66, -61) |
| | 25th | -35 (-53, -35) | 0 (-14, 3) | -44 (-49, -32) | -43 (-48, -7) | 14 (-18, 19) | -59 (-62, -51) | -39 (-50, -38) | -62 (-65, -60) |
| | Med | 40 (24, 53) | 128 (73, 202) | 47 (9, 72) | 35 (2, 75) | 54 (27, 55) | -21 (-25, -7) | -3 (-17, 18) | -10 (-16, 0) |
| | 75th | 254 (227, 334) | 462 (395, 655) | 526 (332, 526) | 243 (140, 293) | 256 (236, 358) | 167 (131, 211) | 155 (114, 155) | 78 (26, 78) |
| | Max | 3219 (2324, 4238) | 1796 (1346, 3260) | 2946 (1790, 4158) | 610 (358, 733) | 1040 (911, 1374) | 1956 (1956, 2955) | 365 (274, 449) | 182 (103, 208) |
| | IQR | 289 (272, 377) | 462 (403, 669) | 570 (381, 570) | 286 (176, 317) | 242 (240, 375) | 226 (191, 271) | 194 (156, 194) | 141 (88, 141) |

**Table 4: Pearson correlation coefficients (r) and Nash–Sutcliffe model efficiency coefficients (NSE) between the log-transformed measurement-based vs. simulated loads across world major rivers for the year 1990 (range for the years 1990-2000 in parenthesis). The prediction error is computed as the difference between the simulated and measurement-based estimates of loads expressed as a percentage of the measurement-based load.**

| | SS | TN | DIN | DON | PON | TP | PO$_4^{3-}$ (DIP) | DOP | PP |
|---|---|---|---|---|---|---|---|---|---|
| LM3-FANSY | 10 (10, 11) | 35 (34, 38) | 14 (14, 16) | 13 (13, 14) | 7 (7, 8) | 7 (7, 8) | 2 | 1 | 5 |
| Global NEWS 1 (year 1995) Global NEWS 2 (year 2000) | 19 (11, 27) Beusen et al. (2005) | 44.9 NEWS 2, Mayorga et al. (2010) | 18.9 NEWS 2, Mayorga et al. (2010) | 10 NEWS-DON, Harrison et al. (2005) | 13.5 NEWS 2, Mayorga et al. (2010) | 9.04 NEWS 2, Mayorga et al. (2010) | 1.45 NEWS-DIP-HD, Harrison et al. (2010) | 0.6 NEWS-DOP, Harrison et al. (2005) | 6.56 NEWS 2, Mayorga et al. (2010) |
| QRT; (years 1960-1995) | 12.6 Syvitski et al. (2005) | | | | | | | | |
| IMAGE-GNM (year 2000) | | 36.5 Beusen et al. (2016) | | | | 4 Beusen et al. (2016) | | | |
| Boyer et al. (2006) (mid-1990s) | | 48 | | | | | | | |
| Galloway et al. (2004) (early-1990s) | | 47.8 | | | | | | | |
| Green et al. (2004) (mid-1990s) | | 40 | 14.5 | | | | | | |
| Smith et al. (2003) (1990s) | | | 18.9 | | | | 2.3 | | |

**Table 5: LM3-FANSY and published estimates of global river loads to the coastal ocean in Pg yr$^{-1}$ for SS and Tg yr$^{-1}$ for nutrients. LM3-FANSY results are for the year 1990 (range for the years 1990-2000).**

Correlations between measurement-based vs. simulated $NO_3^-$, $NH_4^+$, DON, and TKN yields, loads, and concentrations across 51, 37, 18, and 12 rivers respectively (Fig. 3, Table 4) indicate that LM3-FANSY can also explain the observed spatial variations in river N in multiple forms and units to a reasonable extent. The fidelity of LM3-FANSY in terms of N is comparable to that of Global NEWS 2 (which does not estimate $NO_3^-$ and $NH_4^+$ separately, but only estimate their sum as DIN). Spatial DIN patterns evaluated by r values are better represented by Global NEWS 2, while LM3-FANSY estimates

better spatial DON patterns (Tables SI4, SI9, Fig. SI2). Measurement-based estimates of particulate nutrient compounds to evaluate our model implemented at 1 degree resolution herein are limited. The evaluation of modeled PON is limited to a few measurement-based TKN estimates that include PON, but these aggregate values are matched reasonably well with the model estimates.

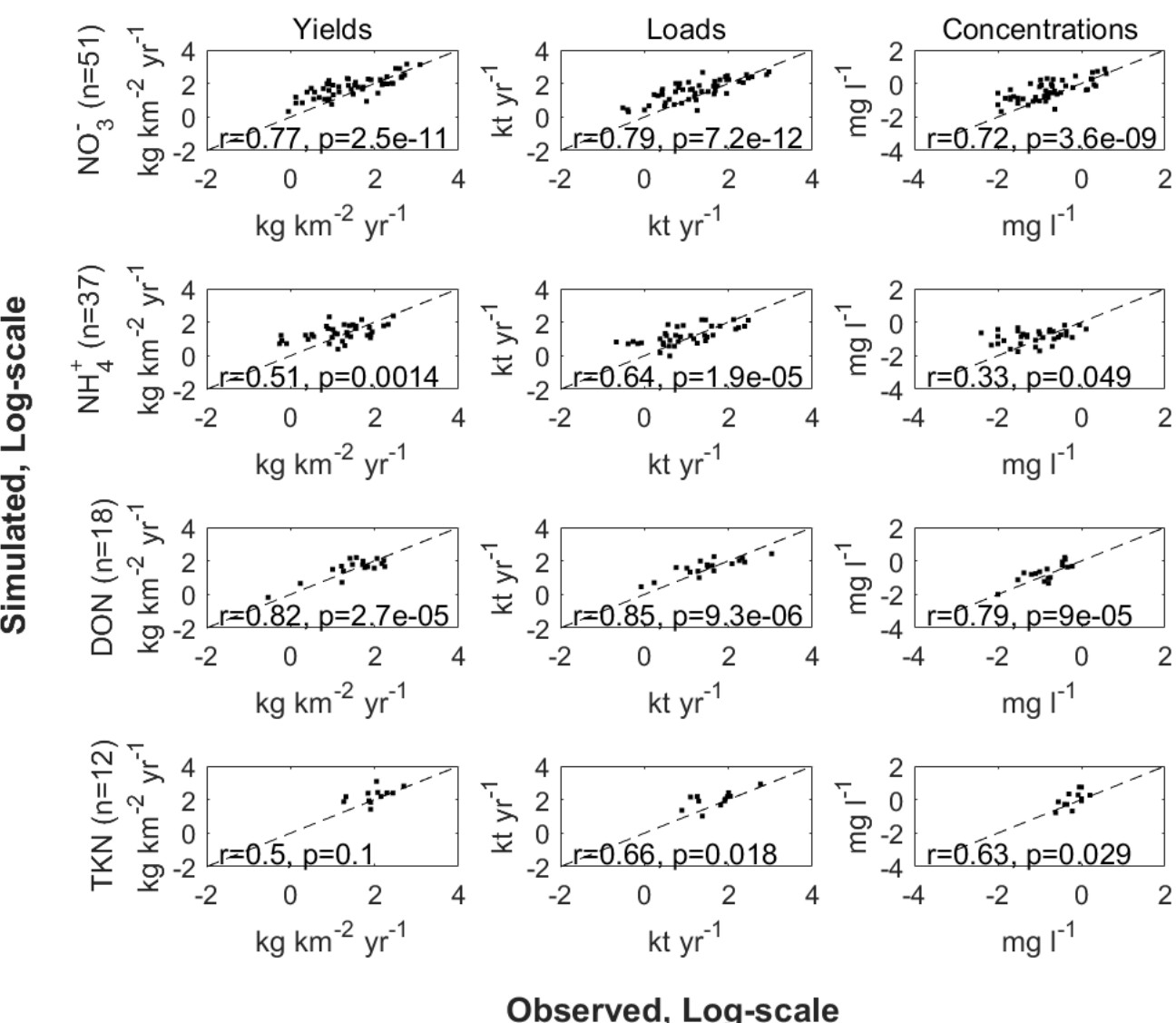

**Figure 3: Pearson correlation coefficients (r) and p values (p) between the log-transformed measurement-based vs. simulated NO$_3^-$, NH$_4^+$, DON, and TKN yields, loads, and concentrations across 51, 37, 18, and 12 rivers for the year 1990.**

Globally, TN inputs to rivers (N inputs hereafter) in LM3-FANSY are 85 (85-91) TgN yr$^{-1}$, of which about 59 (56-60)% are lost to the atmosphere or retained within freshwaters (N loss/retention hereafter) and the other 41 (40-43)% are exported to the coastal ocean (N exports hereafter) (Table 6). LM3-FANSY does not include net long-term N burial to bottom sediments, as all organic N delivered to the sediments is ultimately remineralized. While year to year sediment N inventories may vary, effectively all long-term N losses are to the atmosphere via freshwater denitrification. LM3-FANSY estimates that N

loss/retention is 144 (130-148)% of the N exports, consistent with the 147% and 143% of Galloway et al. (2004) and Seitzinger et al. (2006), yet larger than the 73% estimated by IMAGE-GNM (Beusen et al., 2016). LM3-FANSY estimates of 59 (56-60)% of the N inputs are lost or retained in freshwaters, consistent with the 60% of Galloway et al. (2004), yet larger than the 42% of IMAGE-GNM (Beusen et al., 2015). Recent estimates of global river TN loads to the coastal ocean vary widely, ranging from about 36.5 to 48 TgN yr$^{-1}$ (Table 5, Beusen et al., 2016; Boyer et al., 2006; Galloway et al., 2004;

Green et al., 2004; Mayorga et al. 2010). Our global estimate 35 (34-38) TgN yr$^{-1}$ for the year 1990 (range for the years 1990-2000) is thus at the lower bound of the published range. The simulated global river TN loads contain approximately equal contributions by DIN (the sum of $NO_3^-$ and $NH_4^+$, 14 (14-16) TgN yr$^{-1}$, 41% of TN) and DON (13 (13-14) TgN yr$^{-1}$, 39% of TN), with a lesser contribution by PON (7 (7-8) TgN yr$^{-1}$, 20% of TN). The estimates of global river DIN loads are at the lower end of recent estimates, which range from 14.5 to 18.9 TgN yr$^{-1}$ (Mayorga et al. 2010; Green et al., 2004; Smith et

al., 2003). This may be partly due to an overestimation of freshwater denitrification and/or algae-mediated transformations to organic forms. In contrast, our global river DON load estimate is slightly higher than a previous estimate 10 TgN yr$^{-1}$ (Harrison et al., 2005). Our global river PON load estimate is considerably lower than a previous estimate, 13.5 TgN yr$^{-1}$ (Mayorga et al. 2010). See Sect. 4.5 for further discussion.

| | Tg yr$^{-1}$ in the 1st line % of N inputs into freshwaters in the 2nd line % of N exports to the coastal ocean in the 3rd line | | | | | |
|---|---|---|---|---|---|---|
| | N inputs into freshwaters | N loss by freshwater denitr. & N retention within freshwaters | N exports to the coastal ocean | P inputs into freshwaters | P retention within freshwaters | P exports to the coastal ocean |
| LM3-FANSY Year 1990 (range for 1990-2000) | 85 (85-91) | 50 (48-53) 59 (56-60)% 144 (130-148)% | 35 (34-38) 41 (40-43)% | 8 (8-9) | 1 9 (6-10)% 10 (7-11%) | 7 (7-8) 91 (90-94)% |
| Galloway et al. (2004) Early-1990s | 118.1 | 70.3[A] 60% 147% | 47.8 40% | | | |
| IMAGE-GNM, Beusen et al. (2016) Year 2000 | 64 | 27 42% 73% | 37 58% | 9 | 5 56% 125% | 4 44% |
| Seitzinger et al. (2006) Mid-1990s | | 66[B] - | 46 | | | |

| | | | | | |
|---|---|---|---|---|---|
| | 143% | | | | |
| Maavara et al. (2015) Year 2000 | | | 10.8[C] | 1.3[D] 12% | |
| Global NEWS, Seitzinger et al. (2010) Year 2000 | | 43.2 | | | 8.6 |

[A]"Nr input to rivers" minus "Nr export to coastal areas" in Table 1 of Galloway et al. (2004).

[B]The sum of denitrification in "rivers" and "lakes and reservoirs" in Table 1 of Seitzinger et al. (2006).

[C]Inputs of P to dam reservoirs in Table 1 of Maavara et al. (2015).

[D]Retention of P by dam reservoirs in Table 1 of Maavara et al. (2015).

**Table 6: LM3-FANSY and published estimates of global freshwater N and P budgets.**

Simulated river $PO_4^{3-}$, DOP, and TP yields, loads, and concentrations are in reasonable agreement with more limited measurement-based estimates across 47, 9, and 5 rivers respectively (Fig. 4, Table 4). Global NEWS 2 has generally higher correlations for yields/loads and lower correlations for concentrations for the three species, compared to LM3-FANSY (Tables SI6-SI8, Fig. SI2).

Globally, TP inputs to rivers in LM3-FANSY are 8 (8-9) TgP yr$^{-1}$, of which about 9 (6-10)% are stored within freshwaters and 91 (90-94)% are exported to the coastal ocean (Table 6). IMAGE-GNM estimates that ~56% (5 of 9 TgP yr$^{-1}$) of the P inputs are stored within freshwaters (Beusen et al., 2016). This is a large difference from our estimate of 9 (6-10)% retention, but the difference is around a very uncertain number as the storage has not been directly measured. LM3-FANSY, which does not account for dams and reservoirs, likely underestimates global freshwater P retention by at least ~12% (Table 6, Maavara et al., 2015, see Sect. 4.5 for further discussion). The overall consistency of our SS, N, and P estimates with the observed cross-watershed constraints (Figs. 2-4, SI6, Table 4), however, suggests that the bias introduced by the lack of dams and reservoirs may not be large. In contrast, underestimates of P exports to the coastal ocean from high-exporting basins such as the Amazon, Ganges and Yangtze Rivers shown in Figure 2 of Harrison et al. (2019) imply that IMAGE-GNM likely overestimates global freshwater P retention. Even though our freshwater P retention estimates are near the lower bound, our P retention estimates are far higher than those for N, mainly reflecting the sorption of $PO_4^{3-}$ onto solids and its deposition to bottom sediments. Our estimate of global river TP loads to the coastal ocean (7, 7-8 TgP yr$^{-1}$) falls within the range of other estimates (9.04 TgP yr$^{-1}$ from Global NEWS 2, Mayorga e al., 2010 and 4 TgP yr$^{-1}$ from IMAGE-GNM, Beusen et al., 2016, Table 5). LM3-FANSY estimates that globally, rivers export 5 TgP yr$^{-1}$ as PP (64% of TP), 2 TgP yr$^{-1}$ as $PO_4^{3-}$ (25%), and 1 TgP yr$^{-1}$ as DOP (11%). The global river $PO_4^{3-}$, DOP, and PP load estimates are consistent well with previous estimates of 1.45-2.3 TgP yr$^{-1}$ for $PO_4^{3-}$ (Harrison et al., 2010; Smith et al., 2003), 0.6 TgP yr$^{-1}$ for DOP (Harrison et al., 2005), and 6.6 TgP yr$^{-1}$ for PP (Mayorga et al., 2010).

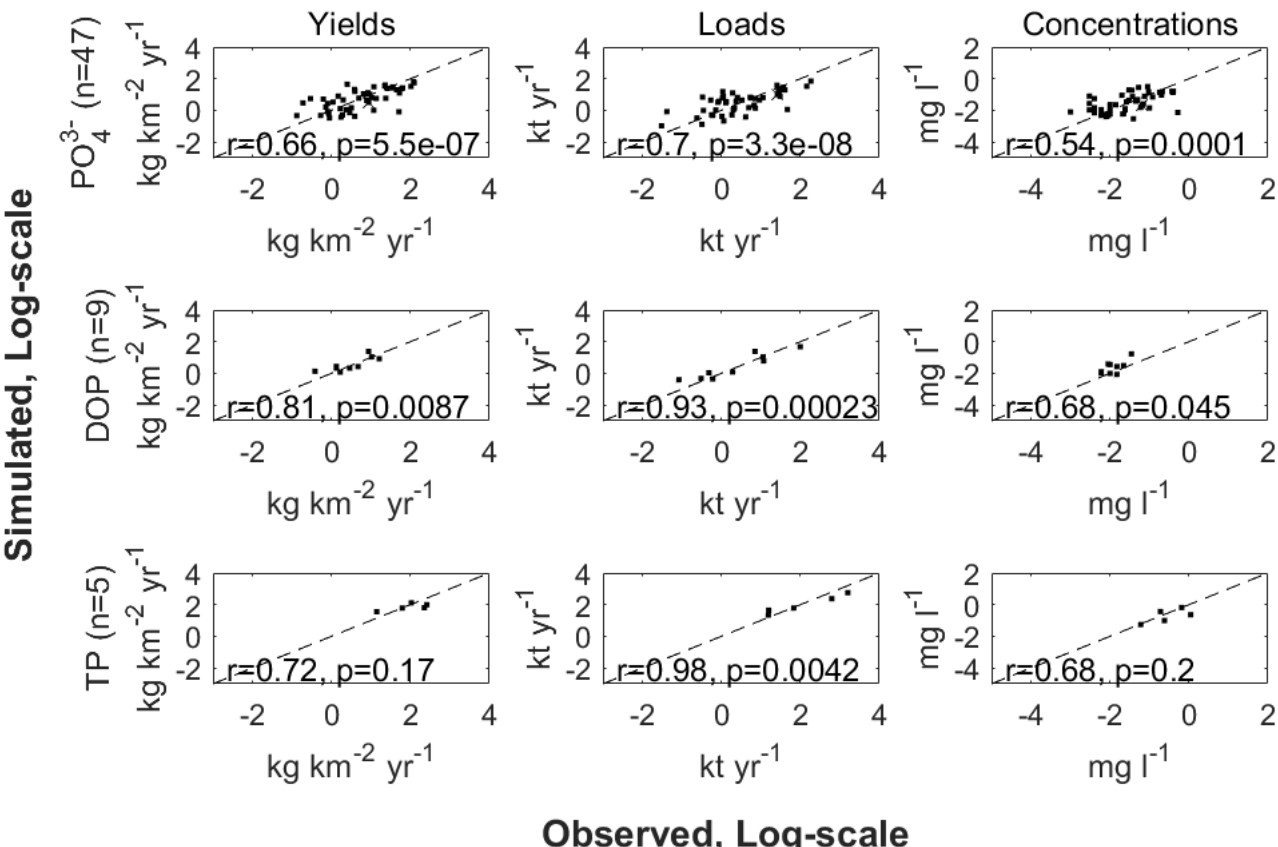

**Figure 4: Pearson correlation coefficients (r) and p values (p) between the log-transformed measurement-based vs. simulated PO4[3-], DOP, and TP yields, loads, and concentrations across 47, 9, and 5 rivers for the year 1990.**

Global watershed model performance of simulating N:P ratios has not been reported in prior publications. While our simulations are reasonably successful in capturing cross ecosystem differences in both N and P species, variations in their

ratios proved more challenging. The model captures the mean ratio of DIN and DIP, but little of the variation (Fig. SI3). One factor that likely contributes to this is the inconsistency between the N inputs to rivers, the majority of which (~92%) were simulated within LM3, and the P inputs, all of which were drawn from another model (IMAGE-GNM). Continued refinement is thus needed to reliably capture N:P ratio variations in rivers and their subsequent water quality implications (see Sect. 4.5).

Despite the relatively simple nature of lake biogeochemistry in LM3-FANSY (i.e., vertically unresolved mixed reactors, Lee et al., 2019), the model creates a reasonable range of chlorophyll a concentrations (Fig. SI4) that generally fall within a range of in-situ estimates from globally distributed lakes (Sayers et al., 2015). The in-situ estimates in the compilation of Sayers et

al. (2015) range from ~0 to ~100 mg m$^{-3}$, mostly falling between 5 and 50 mg m$^{-3}$ (Fig. 5 of Sayers et al. (2015), available at https://www.tandfonline.com/doi/full/10.1080/01431161.2015.1029099).

Despite the significant correlation between the measurement-based and modeled estimates for each solid and nutrient form across various rivers, errors on a basin-by-basin scale are substantial, with high-load, large basins tending to have large absolute errors (Figs. 2-4). However, the ranges of prediction errors in our model simulation, as demonstrated in the interquartile range (IQR) and distribution of prediction errors (Table 4), are comparable to those of other models (Dumont et al., 2005; Harrison et al., 2005; Harrison et al., 2010). These suggest that our model has a competitive correlation (r value), precision (IQR), and bias (median error) for each species compared to previous efforts even while including fewer observational constraints on the river sources and more explicitly parameterized freshwater biogeochemical processes.

## 4.2 Spatial pattern analysis

Spatial maps of river solid and nutrient yields/loads help identify global hotspots of water pollution and provide insight into which processes modulate the magnitudes and form of inputs. A global map of simulated terrestrial soil erosion rate from Eq. (3) is more strongly related to the basin slope map than to the rainfall or LAI maps, reflecting the prominent role of topographic steepness in controlling soil erosion (Fig. 5). This is consistent with previous studies (Pelletier, 2012; Syvitski et al., 2003). The eroded soil is transported as suspended load to rivers, some of which is stored within rivers and lakes, and the rest makes its way to large river outlets to the coastal ocean. Simulated river SS yields are high in mountains like the Andes, Rockies, and Himalayas and low in most gently sloping areas. The yields (i.e., loads per area) decrease with distance from mountains, as some of the soil is stored in lowland rivers and lakes and as basin areas (the denominator in yields) increase downstream. In contrast, simulated river SS loads tend to increase downstream, because larger rivers carry more soils from many small streams and tributaries. The Ganges, Changjiang, Indus, and Huang He Rivers in Asia, the Parana and Amazon Rivers in South America, and Mississippi and Columbia in North America are among the largest river SS exporters (i.e., highest loads) in LM3-FANSY.

The Mississippi, Chang Jiang, Ganges, Ob, Amazon, Parana, and Orinoco Rivers are among the top exporters of all three N forms (DIN, DON, and PON) to the coastal ocean in LM3-FANSY (Fig. 6). These basins are characterized by tropical humid climates with high terrestrial productivity, high population/agricultural pressures, and/or high river water discharge. The highest river DIN yields/loads occurred in European and Asian rivers (e.g., Rine, Elbe, Danube, and Zhujiang), despite their relatively low river water discharge and small basin areas, are largely due to substantial anthropogenic N inputs (Dumont et al., 2005; Mayorga et al, 2010). In contrast, the lowest river DIN yields/loads are estimated for arid regions and most high latitude basins with low population densities and less intensive agriculture. The Amazon, Parana, Orinoco, Zaire, Ganges, Zhujiang, Hong, Chang jiang, Mississippi, Yukon, Ob, and Yenisey Rivers are estimated to produce the largest river DON yields/loads. The largest river DON yields/loads are from humid tropical regions, despite lower human pressures,

indicating a critical role of non-anthropogenic sources (i.e., terrestrial soil and litter fluxes from N-enriched natural forests) in exporting the dissolved organic form (Harrison et al., 2005). Low river DON yields/loads tend to occur in relatively dry regions with low anthropogenic pressures.


The Mississippi, Chang Jiang, Ganges, Amazon, and Danube Rivers are among the highest exporters of all three P forms (DIP, DOP, and PP (the sum of POP and PIP)) to the coastal ocean (Fig. 7). Hot spots for river $PO_4^{3-}$ yields/loads tend to occur in river basins including densely populated large urban centers, such as Chang Jiang, Huang He, Mekong, Shatt el Arab, Ganges, Godavari, Narmada, and Danube. The critical role of urban areas with sewage effluents in producing high

river $PO_4^{3-}$ yields is consistent with previous studies (Harrison et al., 2010; Mayorga et al., 2010). High river $PO_4^{3-}$ yields also occur in humid river basins characterized by high P weathering rates, such as the Amazon, Parana, Zaire, Niger, Ganges, Chang jiang, and Mekong or in river basins including intensively farmed areas like the Mississippi (Harrison et al., 2010). Highest DOP and PP yields/loads tend to follow a pattern similar to that of $PO_4^{3-}$, but there are also differences in patterns of $PO_4^{3-}$ yields from patterns of PP yields. The differences, in part, result from deforestation and agricultural

expansion in river basins like Columbia and Amur demonstrating elevated PP yields in comparison to $PO_4^{3}$ yields (Harrison et al., 2019).

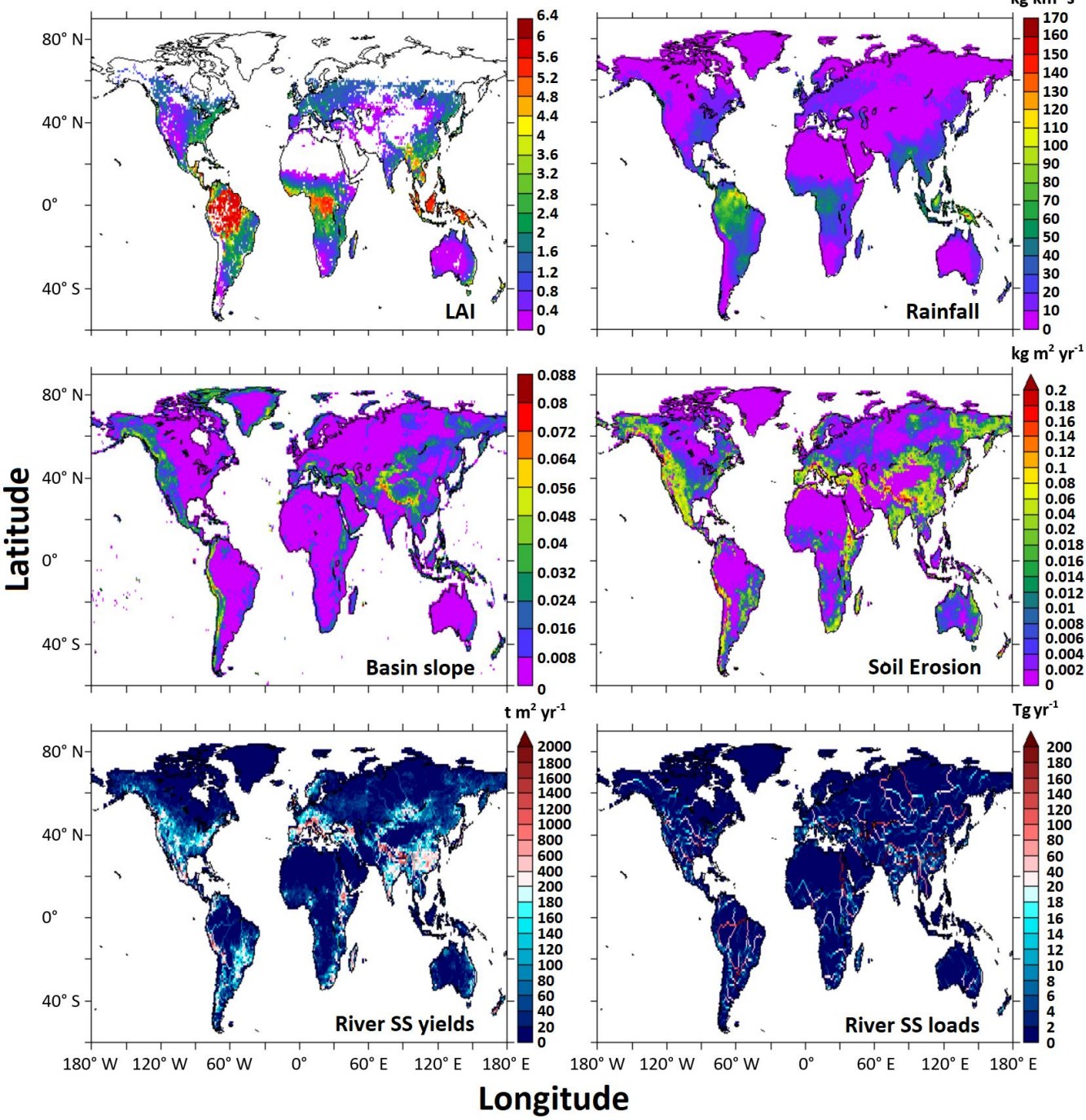

Figure 5: Global maps of the model inputs of LAI, rainfall, and basin slope and of the simulated soil erosion rate, river SS yields and loads for the year 1990.

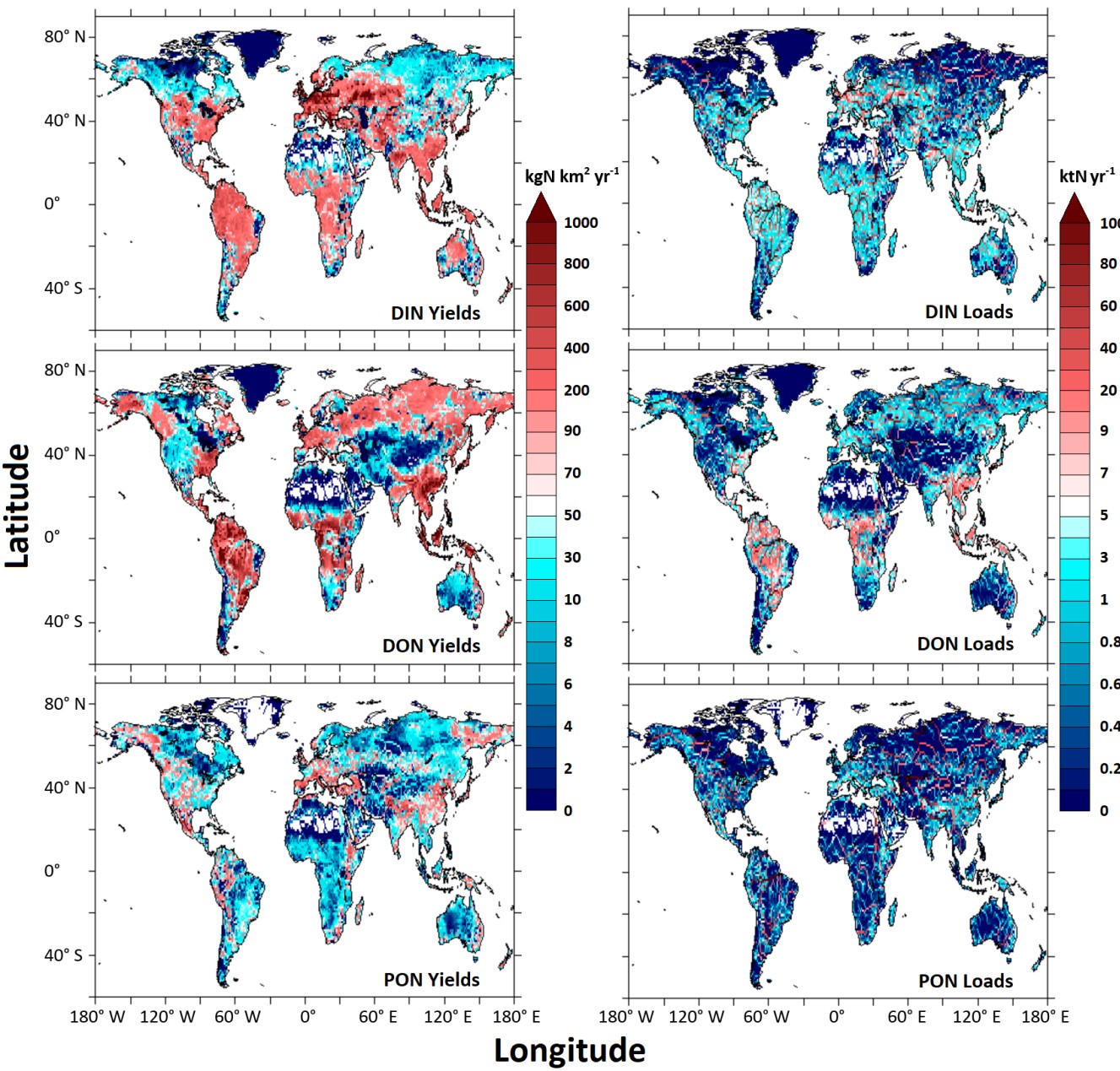

**Figure 6: Global maps of the simulated river DIN, DON, and PON yields and loads for the year 1990.**


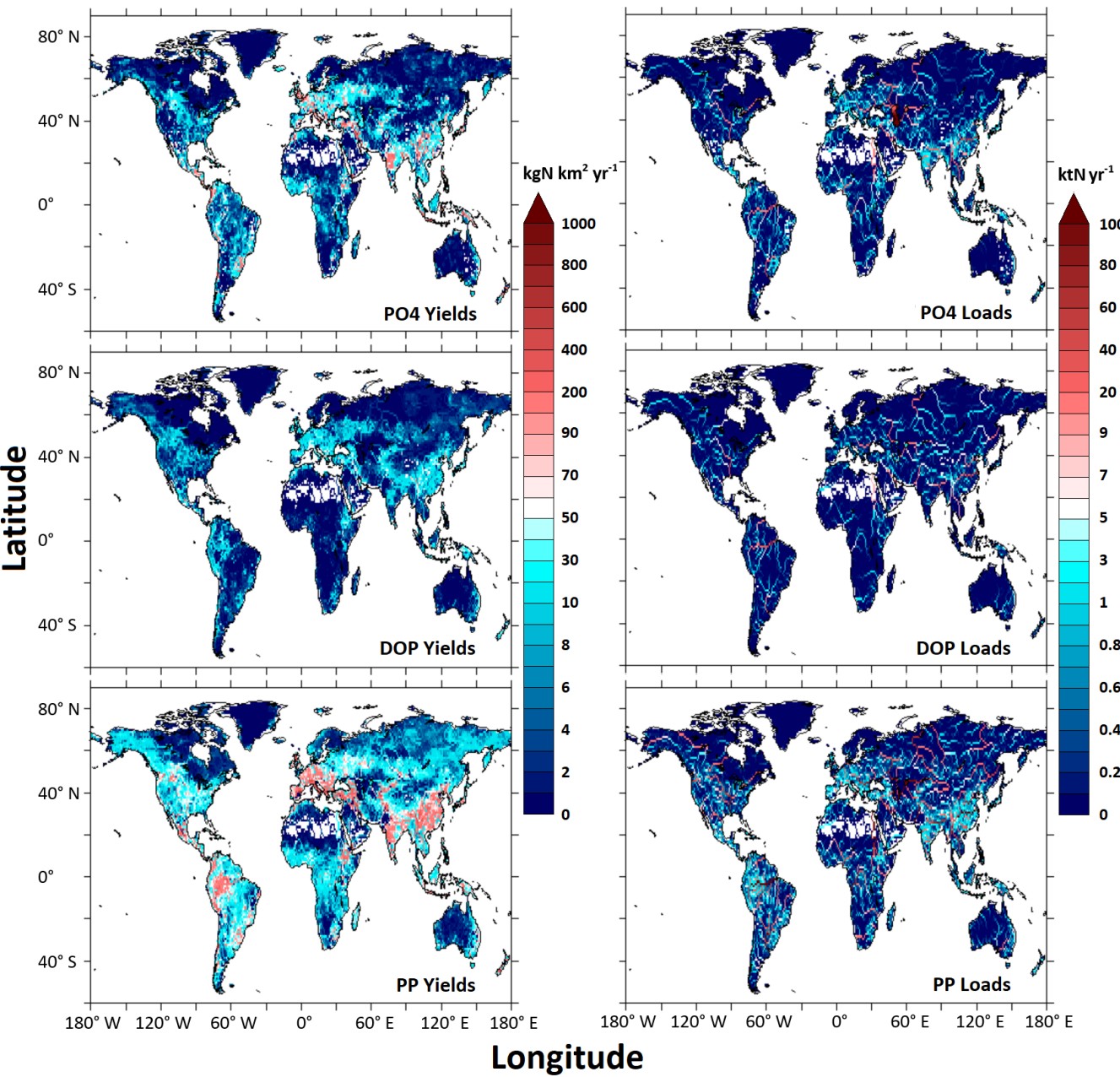

**Figure 7: Global maps of the simulated river PO₄³⁻, DOP, and PP yields and loads for the year 1990.**


## 4.3 Time series analysis

Correlations between the measurement-based vs. simulated annual river solid and nutrient load time series across 8 stations in large U. S. rivers for the period ~1963-2000 demonstrate that, while model results for individual rivers may be biased, the simulated solid and N loads in different N species covary with variations of the measurement-based loads (Table 7, Fig. 8, Fig. SI5). LM3-FANSY, however, has less capability of capturing interannual variability of the P loads. The prominent difference between the solid and N dynamics vs. the P dynamics in LM3-FANSY is that the large terrestrial litter and soil

sources for N are simulated by LM3, while the corresponding P inputs are externally prescribed, because LM3 does not yet include terrestrial P dynamics. The lack of terrestrial P dynamics in LM3 is one of the most plausible reasons for the less capability of capturing the P load interannual variability (see Sect. 4.5 for further discussion).

Prediction errors of the average river solid and nutrient loads for the periods ~1963-2000 across the 8 stations (Table 7) are

comparable to the errors shown in the cross-watershed analysis (Table 4), as well as to the errors of cross-watershed analyses in previous global watershed modeling studies (Harrison et al., 2010; He et al., 2011; Mayorga et al., 2010; Pelletier, 2012). The cross-watershed emphasis herein reflects the intended global application of the model, but significant biases for individual rivers are a natural consequence of the prioritization. For example, the Mississippi River N loads in the LM3-FANSY simulation herein are significantly underestimated despite the model's modest global load bias. Focused

investigations of larger misfits will be pursued in future work to increase model robustness through targeted enhancements to better reflect regional variations. Alternatively, for regional applications, tuning a single parameter for N dynamics or solid dynamics allows LM3-FANSY to be calibrated for a specific river. For example, for the Mississippi River, reducing the freshwater denitrification rate coefficient from 0.15 day$^{-1}$ to 0.05 day$^{-1}$ can reduce the errors from -60% to -15% for $NO_3^-$ and from -23% to 8% for TN, while increasing the correlation coefficients from 0.7 to 0.75 for $NO_3^-$ and from 0.65 to 0.67 for

TN (Fig. 8). Reducing the free parameter of terrestrial soil erosion by half can reduce the error of SS from 150% to 27%. Lastly, the t-statistic shows 30 of the 37 measurement-based time series loads have no significant linear trends over the time periods at the 0.05 level (Table 7). LM3-FANSY captures 30 of the 37 measurement-based trends in terms of whether the trends are significant, and when significant, all of the slope signs demonstrating downward vs. upward trends.

| 8 NWQN stations in large U. S. rivers | $NO_3^-$ | TN | $PO_4^{3-}$ | TP | SS |
|---|---|---|---|---|---|
| Mississippi River near St. Francisville, LA | 1968 | 1975 | 1970 | 1974 | 1993 |
| | .70, <.01 | .65, <.01 | .34, .06 | .14, .49 | .76, .03 |
| | -60 % | -23 % | -43 % | 91 % | 150 % |
| | +/+ | none/none | -/- | none/- | -/none |
| Mississippi River at Thebes, IL | 1973 | 1973 | 1977 | 1973 | 1973 |
| | .49, <.01 | .60, <.01 | .07, .74 | .40, .03 | .45, .02 |
| | -71 % | -53 % | -21 % | 51 % | -4 % |
| | none/none | none/none | none/none | none/none | none/none |

| Station | | | | | |
|---|---|---|---|---|---|
| Missouri River at Hermann, MO | 1967<br>.43, .01<br>34 %<br>none/+ | 1970<br>.48, <.01<br>41 %<br>none/none | 1979<br><.01, .97<br>116 %<br>none/none | 1967<br>.35, .04<br>208 %<br>none/none | 1975<br>.66, <.01<br>8 %<br>none/none |
| Ohio River at Olmsted, IL | 1963<br>.56, <.01<br>-75 %<br>none/+ | 1973<br>.76, <.01<br>-46 %<br>none/none | 1964<br>.36, .03<br>-71 %<br>-/- | 1968<br>.05, .80<br>-31 %<br>none/- | 1973<br>.48, .01<br>178 %<br>-/- |
| Mississippi River Below Grafton, IL | 1975<br>.72, <.01<br>-95 %<br>none/none | 1975<br>.73, <.01<br>-86 %<br>none/none | 1979<br>.10, .67<br>-93 %<br>none/none | 1975<br>.23, .26<br>-65 %<br>none/- | 1993<br>.68, .06<br>-35 %<br>none/none |
| Arkansas River at David D Terry Lock and Dam below Little Rock, AR | 1969<br>.62, <.01<br>-29 %<br>none/none | 1970<br>.64, <.01<br>92 %<br>none/none | 1981<br>-0.35, .13<br>20 %<br>none/none | 1969<br>.40, .02<br>515 %<br>none/none | No Data |
| Columbia River near Beaver Army Terminal, OR | 1993<br>.74, .04<br>140 %<br>none/none | 1993<br>.75, .03<br>150 %<br>none/none | 1993<br>.21, .62<br>-2 %<br>none/none | 1993<br>.07, .87<br>318 %<br>none/none | 1993<br>.61, .11<br>1588 %<br>none/none |
| St. Lawrence River at Cornwall, Ontario, near Massena, NY | 1974<br>.50, <.01<br>-20 %<br>+/none | 1974<br>-.07, .75<br>-44 %<br>none/none | No data | 1974<br>.68, <.01<br>530 %<br>-/- | No data |

Table 7: Summary of the LM3-FANSY's capacity to simulate observation-based estimates of time varying solid and nutrient loads from 8 stations in large U.S. rivers with extended time series. Rows correspond to stations and columns to different nutrients or suspended solids. Each table entry contains multiple skill metrics. The first entry indicates when the time series starts, with all time series continuing to 2000. The second entry provides the Pearson correlation coefficient quantifying the correspondence between observed and simulated annual loads. A p value indicating the significance level of this correlation

is also provided. The third entry gives the prediction error computed as the difference between the simulated and measurement-based load averages expressed as a percentage of the measurement-based load average over the time series. The fourth line indicates whether the observation-based and LM3-FANSY estimates have significantly positive (+), negative (-) trends, or no trend (none) over the time series period. Thus, "+/+" indicates that both the observation-based and LM3-FANSY estimates have a significantly increasing trend, whereas "+/-" indicates that the observation-based estimates have a significantly increasing trend while the LM3-FANSY estimates have a significantly decreasing trend. Trend significance was

tested at the 5% level using the p-value of the t-statistic for a linear trend.

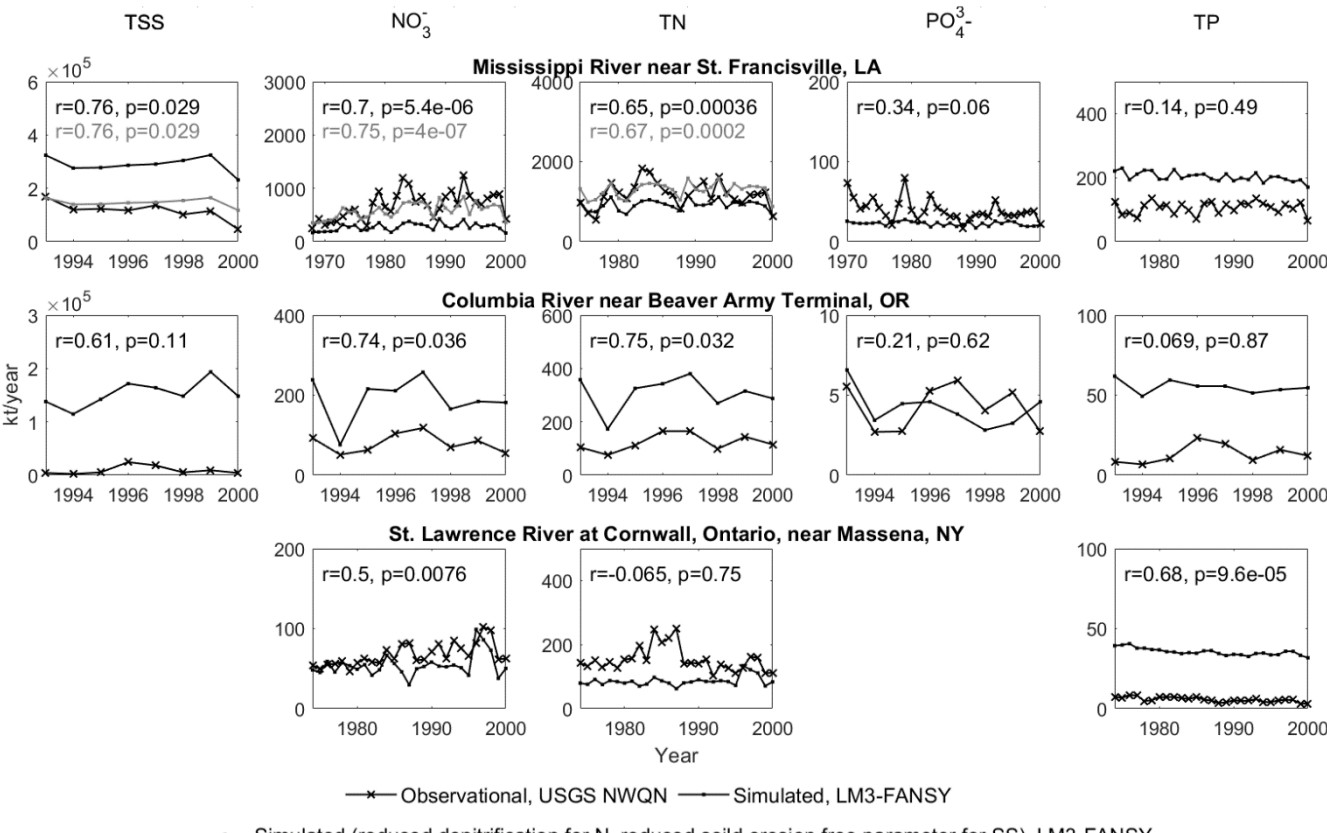

**Figure 8: Pearson correlation coefficients (r) and p values (p) between the measurement-based vs. simulated annual loads from large U. S. rivers for the period ~1968-2000. The grey shows the load responses to the changes in the freshwater denitrification rate coefficient from 0.15 day⁻¹ to 0.05 day⁻¹ for N dynamics and the free parameter of terrestrial soil erosion by half for solid dynamics (Table 8).**

**4.4 Model sensitivity with changes in parameter settings and nutrient inputs**

For solid dynamics, the free scale parameter of terrestrial soil erosion ($C_1$) plays a significant role in determining the overall amount of river SS loads, with its decrease (increase) by half (twice) leading to a 50% decrease (99% increase) in global river SS loads (Table 8). The decrease (increase) in $C_1$ also reduces (enhances) the erosion associated fluxes from terrestrial litter and soils and, in turn, river PON loads. In addition, the decrease (increase) in $C_1$ reduces (enhances) sorption of $PO_4^{3-}$ to solid particles, as reflected by DIP vs. PIP load changes. These results associated with terrestrial soil erosion are, however, found to be insensitive to one order of magnitude changes in POM-to-PON ratios in eroded terrestrial soils and/or freshwaters ($r_{DN,Ero}$ and/or $r_{DN}$, Sect. 2.2.1).

|  |  | SS | TN | DIN | DON | PON | TP | DIP | PIP | DOP | POP |
|---|---|---|---|---|---|---|---|---|---|---|---|
| Runoff and erosion | Removal | -1 | -90 | -84 | -99 | -84 | -74 | -57 | -61 | -91 | -94 |
|  | +15% | 0 | 13 | 12 | 15 | 13 | 11 | 11 | 7 | 13 | 14 |

| Parameter | Scenario | | | | | | | | | | |
|---|---|---|---|---|---|---|---|---|---|---|
| Wastewater | Removal | 0 | -10 | -14 | -1 | -19 | -10 | -16 | -7 | -14 | -7 |
| | +15% | 0 | 2 | 2 | 0 | 3 | 2 | 3 | 1 | 2 | 1 |
| Aquaculture | Removal | 0 | -1 | -1 | 0 | -1 | 0 | -1 | 0 | -1 | 0 |
| | +15% | 0 | 0 | 0 | 0 | 0 | 0 | 0 | 0 | 0 | 0 |
| Atmospheric deposition | Removal | 0 | -1 | -1 | 0 | -2 | 0 | 0 | 0 | 0 | 0 |
| | +15% | 0 | 0 | 0 | 0 | 0 | 0 | 0 | 0 | 0 | 0 |
| Weathering | Removal | 0 | -2 | 1 | -1 | -11 | -15 | -32 | -13 | -11 | -4 |
| | +15% | 0 | 0 | 0 | 0 | 2 | 2 | 5 | 2 | 2 | 1 |
| $r_{DN,Ero}$ | ×0.1 (1.39) | 0 | 0 | 0 | 0 | 0 | 0 | 0 | 0 | 0 | 0 |
| | ×10 (139) | -4 | 0 | 0 | 0 | 0 | 0 | 1 | -2 | 0 | 0 |
| $r_{DN,Ero}$, $r_{DN}$ | ×0.1 (1.39) | 0 | 0 | 0 | 0 | 2 | 0 | -2 | -1 | 2 | 1 |
| | ×10 (139) | -4 | -1 | 0 | 0 | -5 | 0 | 6 | -1 | -5 | -2 |
| d | Half (0.005, small silt) | 0 | 0 | 0 | 0 | 0 | 0 | 0 | 0 | 0 | 0 |
| | Twice (0.02, large silt) | 0 | 0 | 0 | 0 | 0 | 0 | 0 | 0 | 0 | 0 |
| | (0.084, large phytoplankton) | -1 | 0 | 0 | 0 | -2 | 0 | 0 | 0 | 0 | -1 |
| $\theta_{Al}$, $\theta$, $\theta_{Sed}$ | (1.066, 1.024, 1.024) | 0 | 2 | 4 | 1 | -2 | 0 | 0 | 1 | -1 | 0 |
| | (1.066, 1.047, 1.047) | 0 | 1 | 2 | 0 | 0 | 0 | 0 | 0 | 0 | 0 |
| | (1.066, 1.066, 1.066) | 0 | 0 | 1 | 0 | 0 | 0 | 0 | 0 | 0 | 0 |
| | (1.066, 1.047, 1.08) | 0 | 1 | 2 | 0 | 0 | 0 | 0 | 0 | 0 | 0 |
| $P_{max}^C$ | ×0.5 | 0 | -10 | 4 | -2 | -48 | 0 | 28 | 15 | -43 | -19 |
| | ×2 | 0 | 9 | -6 | 3 | 45 | 0 | -23 | -19 | 43 | 18 |
| $\alpha^{CHL}$ | ×0.5 | 0 | -9 | 4 | -2 | -42 | 0 | 24 | 13 | -38 | -17 |
| | ×2 | 1 | 11 | -7 | 4 | 53 | 0 | -27 | -23 | 52 | 21 |
| $k_{NO_{23}}$ | ×0.5 | 0 | 1 | 0 | 0 | 4 | 0 | -2 | -1 | 3 | 1 |
| | ×2 | 0 | -1 | 0 | 0 | -4 | 0 | 2 | 1 | -4 | -1 |
| $k_{NH_4}$ | ×0.5 | 0 | 1 | 0 | 0 | 3 | 0 | -2 | -1 | 2 | 1 |
| | ×2 | 0 | -1 | 0 | 0 | -3 | 0 | 2 | 1 | -3 | -1 |
| $k_{PO_4}$ | ×0.5 | 0 | 0 | 0 | 0 | 1 | 0 | -1 | -1 | 1 | 1 |
| | ×2 | 0 | 0 | 0 | 0 | -2 | 0 | 1 | 1 | -2 | -1 |
| $k_{NO_{23}}$, $k_{NH_4}$, $k_{PO_4}$ | ×0.1 (~lower bound) | 0 | 3 | -2 | 1 | 16 | 0 | -7 | -6 | 13 | 6 |
| | ×4 (~upper bound) | 0 | -4 | 2 | -1 | -22 | 0 | 15 | 7 | -23 | -9 |
| $k_{ew}$ | ×0.5 | 0 | 0 | 0 | 0 | 0 | 0 | 0 | 0 | 0 | 0 |
| | ×2 | 0 | 0 | 0 | 0 | 0 | 0 | 0 | 0 | 0 | 0 |
| $k_e$ | (0.15) | 1 | 13 | -10 | 5 | 66 | 0 | -34 | -29 | 66 | 26 |
| | (0.45) | 1 | 12 | -8 | 5 | 61 | 0 | -31 | -27 | 61 | 24 |
| $k_{SedN,d}$ | ×0.5 | 0 | 0 | 0 | 0 | 0 | 0 | 0 | 0 | -1 | 0 |
| | ×2 | 0 | 0 | 0 | 0 | 0 | 0 | 0 | 0 | 0 | 0 |
| $k_{SedP,d}$ | ×0.5 | 0 | 0 | 0 | 0 | 0 | 0 | 0 | 0 | 0 | 0 |
| | ×2 | 0 | 0 | 0 | 0 | 0 | 0 | 0 | 0 | 0 | 0 |
| $k_{PON,d}$ | ×0.5 | 0 | 1 | 0 | 0 | 3 | 0 | 0 | 1 | -1 | -1 |
| | ×2 | 0 | -1 | 0 | 0 | -6 | 0 | -1 | -1 | 1 | 1 |
| $k_{POP,d}$ | ×0.5 | 0 | 0 | 0 | 0 | -1 | 0 | -1 | -1 | -5 | 3 |
| | ×2 | 0 | 0 | 0 | 0 | 2 | 0 | 2 | 1 | 8 | -5 |
| $k_{DON,d}$ | ×0.5 | 0 | 7 | -8 | 29 | 0 | 0 | 1 | 0 | -1 | 0 |
| | ×2 | 0 | -5 | 9 | -22 | 0 | 0 | 0 | 0 | 0 | 0 |
| $k_{DOP,d}$ | ×0.5 | 0 | -1 | 0 | 0 | -4 | 0 | -5 | -2 | 22 | -2 |
| | ×2 | 0 | 1 | 0 | 0 | 5 | 0 | 4 | 2 | -22 | 2 |
| $k_{nitr}$ | ×0.5 | 0 | 5 | 9 | 0 | 4 | 0 | -2 | -1 | 4 | 2 |
| | ×2 | 0 | -3 | -5 | 0 | -4 | 0 | 2 | 1 | -4 | -1 |
| $k_{denitr}$ | (0.05) | 0 | 25 | 54 | 1 | 10 | 0 | -5 | -4 | 9 | 4 |

| | | | | | | | | | | |
|---|---|---|---|---|---|---|---|---|---|---|
| | (0.075) | 0 | 15 | 31 | 0 | 7 | 0 | -4 | -3 | 6 | 3 |
| | (0.1) | 0 | 8 | 17 | 0 | 4 | 0 | -2 | -1 | 4 | 2 |
| | (0.2) | 0 | -5 | -10 | 0 | -3 | 0 | 2 | 1 | -3 | -1 |
| | (0.25) | 0 | -8 | -17 | 0 | -4 | 0 | 3 | 2 | -5 | -2 |
| | (0.3) | 0 | -11 | -23 | 0 | -6 | 0 | 4 | 2 | -6 | -2 |
| $C_1$ | Half | -50 | -3 | -2 | 1 | -14 | 0 | 22 | -26 | 7 | 3 |
| | Twice | 99 | 8 | 2 | -1 | 38 | 0 | -22 | 23 | -5 | -2 |
| $k_m$ | Half | 1 | 15 | -11 | 4 | 61 | 0 | -35 | -31 | 60 | 24 |
| | Twice | 0 | -10 | 4 | -3 | -48 | 0 | 28 | 15 | -44 | -19 |
| $f_{m,DON}$, $f_{m,PON}$ | (0.15, 0.3) | 0 | 1 | 0 | -1 | 7 | 0 | 1 | 1 | -16 | 3 |
| | (0.6, 0.3) | 0 | -2 | 0 | 1 | -10 | 0 | -2 | -1 | 23 | -4 |
| | (0.3, 0.15) | 0 | -1 | 1 | 1 | -12 | 0 | -1 | -1 | 17 | -5 |
| | (0.3, 0.6) | 0 | 1 | -1 | -1 | 9 | 0 | 1 | 1 | -14 | 3 |
| $f_{PON,DON}$ | (0.4) | 0 | 0 | 0 | 0 | 0 | 0 | 0 | 0 | 0 | 0 |
| | (1.0) | 0 | 0 | 0 | 0 | 0 | 0 | 0 | 0 | 0 | 0 |
| $f_{POP,DOP}$ | (0.4) | 0 | 0 | 0 | 0 | 1 | 0 | 1 | 0 | -4 | 0 |
| | (1.0) | 0 | 0 | 0 | 0 | 0 | 0 | 0 | 0 | 2 | 0 |
| $f_{SedN,DON}$ | (0.4) | 0 | 0 | 0 | 0 | 0 | 0 | 0 | 0 | 0 | 0 |
| | (1.0) | 0 | 0 | 0 | 0 | 0 | 0 | 0 | 0 | 0 | 0 |
| $f_{SedP,DOP}$ | (0.4) | 0 | 0 | 0 | 0 | 0 | 0 | 1 | 0 | -2 | 0 |
| | (1.0) | 0 | 0 | 0 | 0 | 0 | 0 | 0 | 0 | 1 | 0 |
| $f_{PON,DON}$, $f_{POP,DOP}$, $f_{SedN,DON}$, $f_{SedP,DOP}$ | (0.4) | 0 | 0 | 0 | 0 | 1 | 0 | 1 | 0 | -6 | 0 |
| | (1.0) | 0 | 0 | 0 | 0 | 0 | 0 | -1 | 0 | 3 | 0 |
| $f_{SedP,POP}$ | Half | 0 | 0 | 0 | 0 | 0 | 0 | 0 | 0 | 0 | 0 |
| | Twice | 0 | 0 | 0 | 0 | 0 | 0 | 0 | 0 | 0 | 0 |

**Table 8: Model sensitivities to the changes in inputs, components, and parameters examined based on percentage (%) differences in global river loads between the sensitivity and baseline simulations for the year 1990. The changed parameter values other than a decrease by half or an increase by twice are given in parenthesis.**

For nutrient dynamics, the responses of river loads to a removal of each nutrient input source or an increase of it by 15% suggests that terrestrial runoff and erosion are the dominant drivers of river N loads, followed by wastewater (Table 8). For river P loads, terrestrial runoff and erosion are also the dominant drivers, but unlike for N, the second dominent driver is weathering, followed by wastewater. Terrestrial runoff and erosion also play a critical role in explaining model efficiency and spatial distribution of river nutrient loads (Table 9). A removal of them reduces NSE and r values substantially.

Wastewater plays a relatively small role in explaining model efficiency and spatial distribution of river $NH_4^+$ and $PO_4^{3-}$ loads. Weathering plays a modest role in explaining model efficiency and spatial distribution of P loads in all species. A removal of aquaculture or atmospheric deposition has almost no impacts on the amount, model efficiency, and spatial variation of river loads, suggesting that inaccuracies in these inputs have minor impacts on regional and global model estimates, relative to the inaccuracies associated with the other model inputs. However, the importance of each source is likely to vary, depending on

the dominant control on river nutrient loads in a specific region.

| | Model efficiency, NSE, for the year 1990 (range for the years 1990-2000) in the 1st and 2nd lines<br>Model predictive capacity of spatial variation, r, for the year 1990 (range for the years 1990-2000) in the 3rd an 4th lines | | | | | | | |
|---|---|---|---|---|---|---|---|---|
| Treatment | SS | NO$_3^-$ | NH$_4^+$ | DON | TKN | PO$_4^{3-}$ | DOP | TP |
| Baseline | .54<br>(.48,.55)<br><br>.76<br>(.71,.76) | .43<br>(.33,.49)<br><br>.79<br>(.75,.81) | .31<br>(.29,.45)<br><br>.64<br>(.63,.72) | .66<br>(.62,.77)<br><br>.85<br>(.85,.93) | .00<br>(.00,.21)<br><br>.66<br>(.59,.72) | .49<br>(.42,.54)<br><br>.70<br>(.67,.74) | .85<br>(.85,.88)<br><br>.93<br>(.93,.95) | .82<br>(.81,.86)<br><br>.98<br>(.95,.98) |
| No nutrient runoff and erosion fluxes | .54<br>(.47,.55)<br><br>.76<br>(.71,.76) | -1.99<br>(-2.08,-1.95)<br><br>.62<br>(.61,.64) | -3.77<br>(-4.03,-3.77)<br><br>.42<br>(.39,.44) | -15.41<br>(-16.30,-15.41)<br><br>.06<br>(-.01,.07) | -6.62<br>(-6.73,-6.21)<br><br>.21<br>(.15,.21) | -1.37<br>(-1.66,-0.98)<br><br>.56<br>(.47,.62) | .-52.35<br>(.-52.35,-25.79)<br><br>.86<br>(.70,.89) | 0.00<br>(-.10,0.02)<br><br>.91<br>(.91,.94) |
| No wastewater | 55<br>(.48,.56)<br><br>.76<br>(.72,.76) | 44<br>(.35,.50)<br><br>.78<br>(.75,.80) | .26<br>(.26,.40)<br><br>.57<br>(.57,.65) | .69<br>(.66,.81)<br><br>.85<br>(.85,.93) | .21<br>(.21,.41)<br><br>.70<br>(.65,.77) | .44<br>(.39,.50)<br><br>.68<br>(.64,.71) | .86<br>(.86,.89)<br><br>.94<br>(.94,.95) | .82<br>(.79,.85)<br><br>.96<br>(.93,.97) |
| No aquaculture | .54<br>(.48,.55)<br><br>.76<br>(.71,.76) | .43<br>(.33,.49)<br><br>.79<br>(.75,.81) | .31<br>(.29,.46)<br><br>.64<br>(.63,.72) | .66<br>(.62,.77)<br><br>.85<br>(.85,.93) | .00<br>(.00,.21)<br><br>.66<br>(.59,.72) | .49<br>(.42,.54)<br><br>.71<br>(.67,.74) | .85<br>(.85,.88)<br><br>.93<br>(.93,.95) | .82<br>(.81,.86)<br><br>.98<br>(.95,.98) |
| No atmospheric deposition | .54<br>(.48,.55)<br><br>.76<br>(.71,.76) | .43<br>(.33,.49)<br><br>.79<br>(.75,.80) | .31<br>(.29,.45)<br><br>.64<br>(.63,.72) | .66<br>(.62,.77)<br><br>.85<br>(.85,.93) | .00<br>(.00,.21)<br><br>.65<br>(.58,.70) | .49<br>(.42,.53)<br><br>.70<br>(.67,.74) | .85<br>(.85,.88)<br><br>.93<br>(.93,.95) | .82<br>(.81,.86)<br><br>.98<br>(.95,.98) |
| No weathering | .54<br>(.47,.55)<br><br>.76<br>(.71,.76) | .42<br>(.33,.49)<br><br>.79<br>(.75,.81) | .32<br>(.29,.45)<br><br>.65<br>(.63,.72) | .66<br>(.62,.77)<br><br>.84<br>(.84,.93) | .03<br>(.03,.23)<br><br>.67<br>(.59,.72) | .29<br>(.27,.43)<br><br>.67<br>(.64,.72) | .69<br>(.68,.73)<br><br>.85<br>(.85,.88) | .73<br>(.73,.78)<br><br>.94<br>(.93,.97) |
| No dynamic light shading component of algae dynamics Ke=0.15 | .54<br>(.48,.55)<br><br>.76<br>(.71,.76) | .44<br>(.32,.51)<br><br>.76<br>(.70,.79) | .33<br>(.32,.47)<br><br>.64<br>(.63,.71) | .61<br>(.57,.71)<br><br>.84<br>(.84,.91) | .-.16<br>(-.18,.04)<br><br>.71<br>(.61,.73) | .18<br>(.16,.24)<br><br>.58<br>(.55,.59) | .82<br>(.81,.85)<br><br>.92<br>(.92,.94) | .82<br>(.81,.86)<br><br>.98<br>(.95,.98) |
| Ke=0.45 | .54<br>(.47,.55)<br><br>.76<br>(.71,.76) | .44<br>(.33,.51)<br><br>.77<br>(.71,.79) | .33<br>(.32,.47)<br><br>.64<br>(.63,.71) | .61<br>(.58,.72)<br><br>.84<br>(.84,.92) | .-.15<br>(-.17,.05)<br><br>.70<br>(.62,.73) | .26<br>(.23,.31)<br><br>.61<br>(.58,.62) | .82<br>(.82,.85)<br><br>.92<br>(.92,.94) | .82<br>(.81,.86)<br><br>.98<br>(.95,.98) |

**Table 9: Model sensitivities to the changes in inputs and components examined based on Pearson correlation coefficients (r) and Nash–Sutcliffe model efficiency coefficient (NSE) between the log-transformed measurement-based vs. simulated loads across world major rivers for the year 1990 (range for the years 1990-2000 in parenthesis).**


The parameter sensitivity tests show relatively insensitive responses of river nutrient loads to the changes in the rate coefficients of decay processes that break down complex organic compounds into simpler organic or inorganic compounds. The changes in the rate coefficients for hydrolysis, nitrification, and denitrification, however, have relatively large impacts on river nutrient loads. This is especially for the freshwater N cycle, which includes an additional loss pathway to the

atmosphere via denitrification unlike the freshwater P cycle. The role of freshwater denitrification on global river N loads,

however, has not been explicitly investigated by previous global watershed modeling studies. Our sensitivity results imply a prominent role of freshwater denitrification in determining the amount of N loss to the atmosphere vs. N exports to the coastal ocean at both global (Table 8) and regional (Fig. 9) scales.

Algae dynamics play a significant role in determining the relative composition of inorganic vs. organic nutrients in freshwaters. Decreasing the algal mortality rate constant by half enhances algal uptake, decreasing DIN (the sum of $NO_3^-$ and $NH_4^+$) and IP (the sum of $PO_4^{3-}$ and PIP) by -11% and -33% respectively, while it increases ON (the sum of DON and PON) and OP (the sum of DOP and POP) by 24% and 33% respectively. Similarly, increasing the maximum photosynthesis rate or chlorophyll a-specific initial slope of the photosynthesis-light curve by twice enhances algal uptake, decreasing DIN by -6 or
-7% and IP by -21% or -25% respectively, while it increases ON by 18% or 21% and OP by 23% and 28% respectively. The opposite holds for the parameter changes that reduce algal uptake. An analysis of the model sensitivity simulations, wherein the dynamic contributors to light extinction (i.e., ISS, POM, and CHL in Eq. (23)) were removed, further suggests that proper light limitation of algal growth is also important for skillful estimates of freshwater inorganic vs. organic nutrients. Removing the dynamic light shading component leads to a ~26 % and ~35% overestimation of ON and OP loads and an
underestimation of DIN and IP loads by ~10 % and ~32% respectively. Inorganic nutrient levels are suppressed by invigorated algal populations and more nutrients end up in organic forms. Algal controls thus offer an effective means of calibrating the mix of inorganic and organic constituents. We note uncertainty associated with the fractions that partition nutrient fluxes from algae mortality to different nutrient forms, which appears to have a modest effect on organic nutrient loads (Table 8).


Finally, additional uncertainty tests have shown the relatively insensitive responses of river loads to a broad range in 1) the fractions that divide externally prescribed TN and TP inputs into different N and P forms (see Sect. 3.1, Table SI10), 2) the fractions that partition fluxes from complex organic nutrient decomposition to simpler organic vs. inorganic nutrients, and 3) the temperature correction factor values that account for the temperature effect on freshwater biogeochemical reactions.


### 4.5 Discussion on uncertainties and future work

The inputs and transport of solids and nutrients through the terrestrial-freshwater system, and transformations within it are governed by complex and interlinked physical, chemical, and biological processes. The understanding of these processes varies greatly, as does the degree of their inevitable simplifications within LM3-FANSY. We have highlighted the numerous
uncertainties and simplifications in the model description and result presentation. Here, we discuss the prominent uncertainties that will be prioritized in future work.

There are several significant uncertainties in modeling the soil erosion and associated fluxes of solids and PON to rivers and the coastal ocean. Our global river PON load estimate (7, 7-8 TgN yr$^{-1}$) is lower than that of Global NEWS 2 (13.5 TgN yr$^{-1}$,

Table 5), but confidences in both estimates are low, without explicit evaluations due to relatively limited direct measurements of PON. However, both simulated global river SS loads (10, 10-11 Pg yr$^{-1}$) and global litter/soil N storage (86 PgN) in LM3-FANSY are at the lower bounds of previous estimates (11-27 Pg yr$^{-1}$ for SS loads, Table 5 and 95 (70-820) PgN for N storage, Post et al., 1985). While river TKN loads, which include PON, agree fairly well with the measurement-based estimates across various rivers (Fig. 3), it seems probable that our global estimate is on the low end. Several factors may have contributed to this. The current relatively coarse resolution globally implemented herein inevitably "glosses over" areas of peak soil erosion. The single vertical layer formulation of LM3 omits any potential interactions between the vertical distribution of soil erosion and that of litter and soil N storage. An implementation of LM3-FANSY at higher resolution capturing the larger number of rivers will allow an expansive evaluation against the larger number of observations, and facilitate a better assessment of these uncertainties.

The challenges of modeling particulates continue once they have entered the freshwater system. The Rouse number-dependent transport criterion from Pelletier (2012) was adapted to simulate the deposition/resuspension fluxes between the suspended matters (i.e., ISS, PON, POP and PIP) and benthic sediments (i.e., Sed, SedN, and SedP). The criterion was designed to primarily simulate suspended loads, typically accounting for > 80% of total (i.e., suspended and bed) loads from most large (> ~100 km$^2$) river basins (Pelletier, 2012; Turowski et al., 2010), without explicitly modeling benthic sediments. We acknowledge that our simplified benthic sediment component resulted from adapting the Pelletier's approach drives uncertainties in modeling the suspended matters and benthic sediments, including important diagenesis, other biogeochemical transformations, and physical processes (e.g., mineralization, denitrification, and net long-term organic burial) that occur within the benthic sediments. An implementation of more sophisticated benthic sediment dynamics with improved observation-based constraints (Chapra et al., 2008; Di Toro, 2001) and bed load transport processes is thus subject to critical future work.

Uncertainties associated with sediment dynamics and bed load transport are compounded by the relatively simple representation of lakes, and the exclusion of anthropogenic hydraulic controls like damming, irrigation, and diversion that affect many rivers. For model evaluation, if available, we used the natural water discharges of GEMS-GLORI when calculating loads and yields from the GEMS-GLORI's concentrations (see Sect. 3.3). Large dams or reservoirs, however, have been shown to impound solids and nutrients to substantially decrease their loadings to rivers (Vorosmarty et al., 2003). Thus, despite the relatively low global river SS loads in this first implementation of LM3-FANSY, the lack of such sediment trapping may have induced overestimations of solid and nutrient loads from river basins including large dams or reservoirs. As a representative example, the Colorado River Basin is known for nearly complete trapping of solids due to large reservoir construction and flow diversion (Vorosmarty et al., 2003). LM3-FANSY does not capture such an extreme trapping. As a result, the Colorado River SS load simulated by LM3-FANSY (99,232 kt yr$^{-1}$) is more consistent with the corresponding load calculated by using the "natural" water discharge of GEMS-GLORI (120,010 kt yr$^{-1}$) than with the load calculated by

using the "actual" water discharge (649 kt yr$^{-1}$). Although use of the actual water discharges is found to not significantly alter the cross-watershed evaluations (Fig. SI6), such anthropogenic hydraulic controls are expected to further increase in the future (Seitzinger et al., 2010). It will be thus important to consider the effects of such controls in future work.

There is also significant room for further model development and improvement. An improved representation of lakes (e.g., vertical layering) is necessary to better resolve algal processes and associated transformations between inorganic and organic nutrient phases. Modeling large lakes with the ocean component of GFDL's Earth System Model (Adcroft et al., 2019) is one of our priority developments, particularly given the importance of algae as a control on the relative proportions of inorganic vs. organic nutrients in freshwaters. An initial configuration for the U. S. Great Lakes is currently under development.

There is also a need to pursue advances to provide a more comprehensive and consistent approach of modeling the coupled N, C, and P cycles across the terrestrial and freshwater continuum of LM3-FANSY. Expansion of LM3 to include terrestrial P dynamics will be targeted to improve estimates of litter/soil P storage and fluxes to streams and rivers, generating mechanistically consistent estimates of N:P ratios of nutrient loads reaching coastal systems. Priority enhancements will also include integration of freshwater C, alkalinity, and silicon dynamics with the current solid, algae, and nutrient dynamics of FANSY to simulate the impacts of river inputs on coastal C budgets and acidification and better understand a role of silicate limitation on the development and persistence of many HABs worldwide.

The current version of LM3-FANSY simulates denitrification emissions from freshwaters, but does not include processes that explicitly separate $N_2O$ and $N_2$ emissions from the freshwater denitrification. As freshwater $N_2O$ emissions have been recognized as an increasingly important greenhouse gas source (Yao et al., 2020), it will be important to differentiate $N_2$ and $N_2O$ emission processes in future work.

Although LM3-FANSY is capable of producing river solids and nutrients, in various forms and units, some disagreements between the modeled and measurement-based estimates remain. Many observational studies have noted the uncertainties associated with measurement methods, location, and frequency that likely contribute to these disagreements. While, in this new model development study, we have particularly focused on evaluation of LM3-FANSY's capability to capture broad scale differences in yields, loads and concentrations across a globally distributed set of rivers, we have also attempted to include a meaningful analysis of interannual variability and trends to bolster this assessment. It is difficult, however, to comprehensively analyze geographic differences, temporal variability, trends, underlying mechanisms, and model's external (e.g., climate forcings) vs. internal (e.g., "land N memory effects", Lee et al., 2021) forcing effects in the same paper, because data sources are disparate and a substantial fraction of rivers we searched for were too small to be well resolved by our relatively coarse resolution global framework. We thus focused on the largest and most comprehensively observed rivers

supplying climate-scale (multi-decadal) information to provide a meaningful (albeit limited) complement to the spatial patterns. Additional time varying constraints are thus needed to build additional confidence in projected changes. A commitment to fostering long-term, frequent solid and nutrient sampling and further scrutiny of the impacts of using different load estimation methods on time series data are both essential to better constrain the model. Finally, all of these model improvement efforts will be facilitated by implementing LM3-FANSY at higher resolution to allow comparison against extensive measurements from smaller rivers across the world. Resolution alone, however, cannot address model deficiencies, but simultaneous improvements in the other areas described above are needed to improve the model.

## 5 Conclusion

Our comparisons of process-based LM3-FANSY outputs with measurement-based estimates across world major rivers demonstrate skillful simulations for most riverine constituents despite being restricted to a universal parameter set – the same parameters for all the basins (i.e., without tuning of each basin). LM3-FANSY represents a significant step forward in terms of capacity to model coupled algae, SS, N, and P dynamics in freshwaters at a process-based, spatially explicit, global scale. Although this study is focused on model development and descriptions of the coupled freshwater SS, N, and P cycles, the capability of LM3 to simulate changes in vegetation and soil C-N storage in response to many terrestrial dynamics under subannual to centurial historical climate and land use changes (Lee et al., 2016; Lee et al., 2019; Lee et al., 2021) allows applications of LM3-FANSY for studies of temporal (subannual to multiyear) variability and long-term trends in global and regional water pollution. Therefore, LM3-FANSY v1.0 can serve as a baseline for studies aimed at understanding the effects of terrestrial perturbations on coastal eutrophication. The mechanistic modeling framework of LM3-FANSY is also well suited to make future projections by use of a new generation of future socioeconomic and climate scenarios over centuries, though we acknowledge that further work is needed to fully resolve underlying mechanisms.

## Code availability

The LM3-FANSY v1.0 code was written in Fortran. The complete code has been archived on Zenodo (https://doi.org/10.5281/zenodo.10962725, Lee, 2024). The main FANSY equations and descriptions that correspond to those in this paper have been commented within the files, river_physics.F90 in the river directory and land_model.F90 (see the README file in the repository).

## Data availability

All reported data, model inputs and outputs used to produce figures are available in the Supplement. The observation-based, historical climate forcing data are available at https://hydrology.soton.ac.uk. Annual atmospheric $CO_2$ estimates are available

at https://esgf-node.llnl.gov/search/input4mips. The atmospheric N deposition data are available at https://esgf-node.llnl.gov/search/input4mips. The dataset of land-use states and transitions and fertilizer N applications are available at https://luh.umd.edu/data.shtml. The observationally derived, monthly average global vegetation LAI dataset can be obtained freely from the NASA Earth Exchange (NEX) website. The nutrient inputs to rivers from Beusen et al. (2015) are available at https://doi.org/10.17026/dans-zgs-9k9m. The datasets that cannot be obtained from established portals are available at the Zenodo repository (https://doi.org/10.5281/zenodo.10962725, Lee, 2024).

## Author contribution

M. Lee and C. A. Stock developed the FANSY model and wrote major portions of the manuscript with substantial inputs from J. P. Dunne and E. Shevliakova. M. Lee performed the model simulations and analyses. All authors analyzed and discussed the results.

## Competing interests

The authors declare that they have no conflict of interest.

## Acknowledgments

Award NA18OAR4320123 from the National Oceanic and Atmospheric administration, U.S. Department of Commerce (ML). The statements, findings, conclusions, and recommendations are those of the authors and do not necessarily reflect the views of the National Oceanic and Atmospheric Administration, or the U.S. Department of Commerce. We thank Fabien Paulot from NOAA/GFDL and Cristina Schultz from Princeton University for their incisive comments on the manuscript. We thank Nathaniel Chaney from Duke University for providing us the slope data.

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

Contributions of Multiple Environmental Changes, Global Biogeochem. Cycles, 36, e2022GB007347, 2022.

Billen, G., Garnier, J., and Hanset, P.: Modelling phytoplankton development in whole drainage networks: The riverstrahler model applied to the Seine River system, Hydrobiologia, 289, 119–137, 1994.

Bowie, G. L., Mills, W. B., Porcella, D. B., Campbell, C. L., Pagenkopf, J. R., Rupp, G. L., Johnson, K. M., Chan, P. W. H., and Gherini, S. A.: Rates, constants, and kinetics formulations in surface water quality modeling, Athens, Georgia, 1985.

Bouwman, A. F., Beusen, A. H. W., Overbeek, C. C., Bureau, D. P., Pawlowski, M., and Glibert, P. M.: Hindcasts and future projections of global inland and coastal nitrogen and phosphorus loads due to finfish aquaculture, Rev. Fish Sci., 21, 112–156, doi:10.1080/10641262.2013.790340, 2013a.

Bouwman, A. F., Klein Goldewijk, K., Van der Hoek, K. W., Beusen, A. H. W., Van Vuuren, D. P., Willems, W. J., Rufino, M. C., and Stehfest, E.: Exploring global changes in nitrogen and phosphorus cycles in agriculture induced by livestock

production over the 1900–2050 period, P. Natl. Acad. Sci. USA, 110, 20882–20887, doi:10.1073/pnas.1012878108, 2013b.

Bouwman, A. F., Pawłowski, M., Liu, C., Beusen, A. H. W., Shumway, S. E., Glibert, P. M., and Overbeek, C. C.: Global hindcasts and future projections of coastal nitrogen and phosphorus loads due to shellfish and seaweed aquaculture, Rev. Fish Sci., 19, 331–357, doi:10.1080/10641262.2011.603849, 2011.

Bowie, G. L., Mills, W. B., Porcella, D. B., Campbell, C. L., Pagenkopf, J. R., Rupp, G. L., Johnson, K.M., Chan, P. W. H.,

Gherini, S. A., and Chamberlin, C. E.: Rates, Constants, and Kinetic Formulations in Surface Water Quality Modeling, U.S. Envir. Prot. Agency, ORD, Athens, GA, ERL, EPA/600/3-85/040, 1985.

Boyer, E. W., Howarth, R. W., Galloway, J. N., Dentener, F. J., Green, P. A., and Vorosmarty, C. J.: Riverine nitrogen export from the continents to the coasts, Global Biogeochem. Cy., 20, GB1S91, 2006.

Chapra, S. C.: Surface Water-Quality Modeling, Waveland Press, Long Grove, IL, USA, 1997.

Chapra, S. C., Pelletier, G. J., and Tao, H. QUAL2K: A Modeling Framework for Simulating River and Stream Water Quality, Version 2.11: Documentation and Users Manual, Civil and Environmental Engineering Dept., Tufts University, Medford, MA., 2008.

CMIP6 Forcing Datasets Summary: https://docs.google.com/document/d/1pU9IiJvPJwRvIgVaSDdJ4O0Jeorv_2ekEtted34K9cA/edit/, last access: 10 September
1005   2023.

Cerdan, O., Govers, G., Le Bissonnais, Y., Van Oost, K., Poesen, J., Saby, N., Gobin, A., Vacca, A., Quinton, J., Auerswald, K., Klik, A., Kwaad, F. J. P. M., Raclot, D., Ionita, I., Rejman, J., Rousseva, S., Muxart, T., Roxo, M. J., and Dostal, T.: Rates and spatial variations of soil erosion in Europe: A study based on erosion plot data, Geomorphology, 122, 167–177, doi:10.1016/j.geomorph.2010.06.011, 2010.

Cordell, D., Drangert, J., and White, S.: The story of phosphorus: Global food security and food for thought, Glob. Environ. Change, 19, 292-305, 2009.

Danielson, J. J. and Gesch, D. B.: Global multi-resolution terrain elevation data 2010 (GMTED2010): U.S. Geological Survey Open-File Report 2011–1073, https://pubs.usgs.gov/of/2011/1073/pdf/of2011-1073.pdf, 2011.

Dentener, F., Stevenson, D., Ellingsen, K., Noije, T. v., Schultz, M., Amann, M., Atherton, C., Bell, N., Bergmann, D., Bey,
I., Bouwman, L., Butler, T., Cofala, J., Collins, B., Drevet, J., Doherty, R., Eickhout, B., Eskes, H., Fiore, A., Gauss, M., Hauglustaine, D., Horowitz, L., Isaksen, I. S. A., Josse, B., Lawrence, M., Krol, M., Lamarque, J. F., Montanaro, V.,Müller, J. F., Peuch, V. H., Pitari, G., Pyle, J., Rast, S., Rodriguez, J., Sanderson, M., Savage, N. H., Shindell, D., Strahan, S., Szopa, S., Sudo, K., Dingenen, R. V., Wild, O., and Zeng, G.: The global atmospheric environment for the next generation, Environ. Sci. Technol., 40, 3586–3594, 2006.

Di Toro, D. M.: Sediment Flux Modeling. Wiley-Interscience, New York, NY, USA, 2001.

Diaz, R. J. and Rosenberg, R.: Spreading dead zones and consequences for marine ecosystems, Science, 321, 926–929, 2008.

Dio Toro, D. M.: Optics of turbid estuarine waters: approximations and applications, Water Res., 12, 1059–1068, 1978.

Dumont, E., Harrison, J. A., Kroeze, C., Bakker, E. J., and Seitzinger, S. P.: Global distribution and sources of dissolved inorganic nitrogen export to the coastal zone: Results from a spatially explicit, global model, Global Biogeochem. Cy., 19,
GB4S02, 2005.

Dunne, J. P., Armstrong, R. A., Gnanadesikan, A., and Sarmiento, J. L.: Empirical and mechanistic models for the particle export ratio, Global Biogeochem. Cy., 19, GB4026, 2005.

Environment and Natural Resources Department, Wastewater as a resource, European Investment Bank, https://www.eib.org/en/publications/wastewater-as-a-resource, 2022.

Eppley, R. W.: Temperature and phytoplankton growth in the sea, Fish. Bull., 70, 1063–1085, 1972.

Ferguson, R. I., and Church, M.: A simple universal equation for grain settling velocity, J. Sediment. Res., 74, 933–937, 2004.

Fowler, D., Coyle, M., Skiba, U., Sutton, M. A., Cape, J. N., Reis, S., Sheppard, L. J., Jenkins, A., Grizzetti, B., Galloway, J. N., Vitousek, P., Leach, A., Bouwman, A. F., Butterbach-Bahl, K., Dentener, F., Stevenson, D., Amann, M., and Voss, M.: The global nitrogen cycle in the twenty-first century, Phil. Trans. R. Soc. B, 368: 20130164, 2013.

Frost, B. W. and Franzen, N. C.: Grazing and iron limitation in the control of phytoplankton stock and nutrient concentration: a chemostat analogue of the Pacific equatorial upwelling zone, Mar. Ecol. Prog. Ser., 83, 291–303, 1992.

Galloway, J. N., Dentener, F. J., Capone, D. G., Boyer, E. W., Howarth, R. W., Seitzinger, S. P., Asner, G. P., Cleveland, C. C., Green, P. A., Holland, E. A., Karl, D. M., Michaels, A. F., Porter, J. H., Townsend, A. R., and Vöosmarty, C. J.: Nitrogen cycles: past, present, and future, Biogeochemistry, 70, 153–226, 2004.

Garnier, J., Nemery, J., Billen, G., and Thery, S.: Nutrient dynamics and control of eutrophication in the Marne River system: modelling the role of exchangeable phosphorus, J. Hydrol., 304, 397–412, 2005.

Geider, R. J., MacIntyre, H. L., and Kana, T. M.: Dynamic model of phytoplankton growth and acclimation: responses of the balanced growth rate and the chlorophyll a:carbon ratio to light, nutrient-limitation and temperature, Mar. Ecol. Prog. Ser., 148, 187-200, 1997.

Gerber, S., Hedin, L. O., Oppenheimer, M., Pacala, S. W., and Shevliakova, E.: Nitrogen cycling and feedbacks in a global dynamic land model, Glob. Biogeochem. Cy., 24, GB1001, 2010.

Glibert, P. M., Magnien, R., Lomas, M. W., Alexander, J., Fan, C., Haramoto, E., Trice, M., and Kana, T. M.: Harmful algal blooms in the Chesapeake and Coastal Bays of Maryland, USA: Comparison of 1997, 1998, and 1999 events, Estuaries, 24, 875–883, 2001.

Glibert, P. M., Mayorga, E., and Seitzinger, S.: Prorocentrum minimum tracks anthropogenic nitrogen and phosphorus inputs on a global basis: Application of spatially explicit nutrient export models, Harmful Algae, 8, 33-38, 2008.

Glibert, P. M. and Terlizzi, D. E.: Cooccurrence of elevated urea levels and dinoflagellate blooms in temperate estuarine aquaculture ponds, Appl. Environ. Microbiol., 65, 5594–5596, 1999.

Green, P. A., Vörösmarty, C. J., Meybeck, M., Galloway, J. N., Peterson, B. J., and Boyer, E. W.: Pre-industrial and contemporary fluxes of nitrogen through rivers: a global assessment based on typology, Biogeochemistry, 68, 71–105, 2004.

Harrison, J. A., Beusen, A. H. W., Fink, G., Tang, T., Strokal, M., Bouwman, A. F., Metson, G. S., and Vilmin, L.: Modeling phosphorus in rivers at the global scale: recent successes, remaining challenges, and near-term opportunities, Curr. Opin. Environ. Sustain., 36, 68-77, 2019.

Harrison, J. A., Bouwman, A. F., Mayorga, E., and Seitzinger, S.: Magnitudes and sources of dissolved inorganic phosphorus inputs to surface fresh waters and the coastal zone: A new global model, Global Biogeochem. Cy., 24, GB1003, 2010.

Harrison, J. A., Caraco, N., and Seitzinger, S. P.: Global patterns and sources of dissolved organic matter export to the coastal zone: Results from a spatially explicit, global model, Global Biogeochem. Cy., 19, GB4S04, 2005.

Hartmann, J., Moosdorf, N., Lauerwald, R., Hinderer, M., and West, A. J.: Global chemical weathering and associated P-release – The role of lithology, temperature and soil properties, Chem. Geol., 363, 145–163, doi:10.1016/j.chemgeo.2013.10.025, 2014.

Hatono, M. and Yoshimura, K.: Development of a global sediment dynamics model, Prog. Earth Planet. Sci., 7, 59, 2020.

He, B., Kanae, S., Oki, T., Hirabayashi, Y., Yamashiki, Y., and Takara, K.: Assessment of global nitrogen pollution in rivers
using an integrated biogeochemical modeling framework, Water Research, 45, 2573-2586, doi.org/10.1016/j.watres.2011.02.011, 2011.

Heisler, J., Glibert, P. M., Burkholder, J. M., Anderson, D. M., Cochlan, W., Dennison, W. C., Dortch, Q., Gobler, C. J., Heil, C. A., Humphries, E., Lewitus, A., Magnien, R., Marshallm, H. G., Sellner, K., Stockwell, D. A., Stoecker, D. K., and
Suddleson, M.: Eutrophication and harmful algal blooms: A scientific consensus, Harmful Algae, 8, 3–13, 2008.

Howarth, R. W. and Marino, R.: Nitrogen as the limiting nutrient for eutrophication in coastal marine ecosystems: Evolving views over three decades, Limnol. Oceanogr., 51, 364–376, 2006.

Hurtt, G. C., Chini, L., Sahajpal, R., Frolking, S., Bodirsky, B. L., Calvin, K., Doelman, J. C., Fisk, J., Fujimori, S., Klein Goldewijk, K., Hasegawa, T., Havlik, P., Heinimann, A., Humpenöder, F., Jungclaus, J., Kaplan, J. O., Kennedy, J.,
Krisztin, T., Lawrence, D., Lawrence, P., Ma, L., Mertz, O., Pongratz, J., Popp, A., Poulter, B., Riahi, K., Shevliakova, E., Stehfest, E., Thornton, P., Tubiello, F. N., van Vuuren, D. P., and Zhang, X.: Harmonization of global land use change and management for the period 850–2100 (LUH2) for CMIP6, Geosci. Model Dev., 13, 5425–5464, 2020.

Kaushal, S., Groffman, P. M., Band, L. E., Shields, C. A., Morgan, R. P., Palmer, M. A., Belt, K. T., Swan, C. M., Findlay, S. E. G., and Fisher, G. T.: Interaction between urbanization and climate variability amplifies watershed nitrate export in
Maryland, Environ. Sci. Technol., 42, 5872–8, 2008.

Kemp, W. M., Boynton, W. R., Adolf, J. E., Boesch, D. F., Boicourt, W. C., Brush, G., Cornwell, J. C., Fisher, T. R., Glibert, P.M., Hagy, J. D., Harding, L. W., Houde, E. D., Kimmel, D. G., Miller, W. D., Newell, R. I. E., Roman, M. R., Smith, E. M., and Stevenson, J.C.: Eutrophication of Chesapeake Bay: historical trends and ecological interactions, Mar. Ecol. Prog. Ser., 303, 1–29, 2005.

Lacoul, P. and Freedman, B.: Environmental influences on aquatic plants in freshwater ecosystems, Environ. Rev., 14, 89-136, 2006.

Lee, C.: Nutrient and pesticide data collected from the USGS National Water Quality Network and previous networks, 1950-2021, U.S. Geological Survey, https://doi.org/10.5066/P948Z0VZ, 2022.

Lee, M.: minjinl/LM3-FANSY: LM3-FANSY v1.0 (v1.0), Zenodo [code], https://doi.org/10.5281/zenodo.10962725, Lee,
1095 2024.

Lee, C. J., Hirsch, R. M., and Crawford, C. G.: An evaluation of methods for computing annual water-quality loads: U.S. Geological Survey Scientific Investigations Report 2019–5084, 59 p., https://zenodo.org/badge/latestdoi/687709269, 2019.

Lee, M., Malyshev, S., Shevliakova, E., Milly, P. C. D., and Jaffé, P. R.: Capturing interactions between nitrogen and hydrological cycles under historical climate and land use, Susquehanna Watershed analysis with the GFDL Land Model LM3-TAN, Biogeosciences, 11, 5809–5826, 2014.

Lee, C. J., Murphy, J. C., Crawford, C. G., and Deacon, J. R.: Methods for computing water-quality loads at sites in the U.S. Geological Survey National Water Quality Network (ver. 1.3, August 2021): U.S. Geological Survey Open-File Report 2017–1120, 20 p., https://doi.org/10.3133/ofr20171120., 2017.

Lee, M., Shevliakova, E., Malyshev, S., Milly, P. C. D., and Jaffé, P. R.: Climate variability and extremes, interacting with nitrogen storage, amplify eutrophication risk, Geophys. Res. Lett., 43, 7520–8, 2016.

Lee, M., Shevliakova, E., Stock, C. A., Malyshev, S., and Milly, P. C. D.: Prominence of the tropics in the recent rise of global nitrogen pollution, Nat. Commun., 10, 1437, 2019.

Lee, M., Stock, C. A., Shevliakova, E., Malyshev, S., and Milly, P. C. D.: Globally prevalent land nitrogen memory amplifies water pollution following drought years, Environ. Res. Lett., 16, 014049, 2021.

Leong, S. C. Y., Murata, A., Nagashima, Y., and Taguchi, S.: Variability in toxicity of the dinoflagellate Alexandrium tamarense in response to different nitrogen sources and concentrations, Toxicon, 43, 407−415, 2004.

Liu, X., Stock, C. A., Dunne, J. P., Lee, M., Shevliakova, E., Malyshev, S., and Milly, P. C. D.: Simulated global coastal ecosystem responses to a half-century increase in river nitrogen loads, Geophys. Res. Lett., 48, e2021GL094367, 2021.

Maavara, T., Parsons, C. T., Ridenour, C., Stojanovic, S., D€urr, H. H., Powley, H. R., and Van Cappellen, P.: Global phosphorus retention by river damming. Proc. Natl. Acad. Sci. USA, 112, 15603–15608, 2015.

Mayorga, E., Seitzinger, S. P., Harrison, J. A., Dumont, E., Beusen, A. H. W., Bouwman, A. F. Fekete, B. M., Kroeze, C., and Van Drecht, G.: Global Nutrient Export from WaterSheds 2 (NEWS 2): Model development and implementation, Environ. Model. Softw., 25, 837-853, 2010.

McGechan, M. B. and Lewis, D. R.: Sorption of phosphorus by soil, part 1: principles, equations and models, Biosyst. Eng., 82, 1–24, 2002.

McLaughlin, C. J., Smith, C. A., Buddemeier, R. W., Bartley, J. D., and Maxwell, B. A.: Rivers, runoff, and reefs, Glob. Planet. Change, 39, 191-199, 2003.

Meinshausen, M., Vogel, E., Nauels, A., Lorbacher, K., Meinshausen, N., Etheridge, D. M., Fraser, P. J., Montzka, S. A., Rayner, P. J., Trudinger, C. M., Krummel, P. B., Beyerle, U., Canadell, J. G., Daniel, J. S., Enting, I. G., Law, R. M., Lunder, C. R., O'Doherty, S., Prinn, R. G., Reimann, S., Rubino, M., Velders, G. J. M., Vollmer, M. K., Wang, R. H. J., and Weiss, R.: Historical greenhouse gas concentrations for climate modelling (CMIP6), Geosci. Model Dev., 10, 2057–2116, 2017.

Meybeck, M. and Ragu, A.: Presenting the GEMS-GLORI, a compendium of world river discharge to the oceans, Freshwater Contamination, 243, 3-14, 1997.

Meybeck, M. and Ragu, A.: GEMS-GLORI world river discharge database, https://doi.org/10.1594/PANGAEA.804574, 2012.

Milly, P. C. D., Malyshev, S., Shevliakova, E., Dunne, K. A., Findell, K. L., Gleeson, T., Liang, Z., Phillipps, P., Stouffer, R. J., and Swenson, S.: An enhanced model of land water and energy for global hydrologic and earth-system studies, J. Hydrometeorol., 15, 1739–1761, 2014.

Morée, A. L., Beusen, A. H. W., Bouwman, A. F., and Willems, W. J.: Exploring global nitrogen and phosphorus flows in urban wastes during the twentieth century, Global Biogeochem. Cy., 27, 1–11, doi:10.1002/gbc.20072, 2013.

Nemery, J.: Origine et devenir du phosphore dans le continuum aquatique de la Seine, des petits basins à` l'estuaire. Roˆle du phosphore eˊchangeable sur l'eutrophisation, Ph.D. thesis, University of Paris, France, 258 pp., 2003.

Paerl, H. W., Otten, T. G., and Kudela, R.: Mitigating the expansion of harmful algal blooms across the freshwater-to-marine

continuum, Environ. Sci. Technol., 52, 5519–5529, 2018.

Parsons, M. L., Dortch, Q., and Turner, R. E.: Sedimentological evidence of an increase in Pseudo-nitzschia (Bacillariophyceae) abundance in response to coastal eutrophication, Limnol. Oceanogr., 47, 551–558, 2002.

Pelletier, J. D.: A spatially distributed model for the long-term suspended sediment discharge and delivery ratio of drainage basins, J. Geophys. Res., 117, F02028, 2012.

Pelletier, G. J., Chapra, S. C., and Tao, H.: QUAL2Kw—A framework for modeling water quality in streams and rivers using a genetic algorithm for calibration. Environ, Model. Softw., 21, 419–425, 2006.

Poesen, J., Boardman, J., Wilcox, B., and Valentin, C.: Water erosion monitoring and experimentation for global change studies, J. Soil Water Conserv., 51, 386–390, 1996.

Post, W. M., Pastor, J., Zinke, P. J., and Stangenberger, A. G.: Global patterns of soil nitrogen storage, Nature, 317, 613–

1150 616, 1985.

Renschler, C. S. and Harbor, J.: Soil erosion assessment tools from point to regional scales—The role of geomorphologists in land management research and implementation, Geomorphology, 47, 189–209, 2002.

Redfield, A. C., Ketchum, B. H., and Richards, F. A.: The Influence of Organisms on the Composition of Seawater, in: The Sea, edited by: Hill, M. N., Wiley-Interscience, NY, 27-46, 1963.

Restrepo, J. D., Zapata, P., Díaz, J. M., Garzón-Ferreira, J., and García, C. B.: Fluvial fluxes into the Caribbean Sea and their impact on coastal ecosystems: The Magdalena River, Colombia, Glob. Planet. Change, 50, 33-49, 2006.

Riley, G. A.: Oceanography of Long Island sound 1952-1954, II. Physical Oceanography, Bull. Bingham. Oceanog. Collection., 15, 15-16, 1956.

Sayers, M. J., Grimm, A. G., Shuchman, R. A., Deines, A. M., Bunnell, D. B. et al.: A new method to generate a high-

resolution global distribution map of lake chlorophyll, Int. J. Remote Sens., 36, 1942-1964, DOI: 10.1080/01431161.2015.1029099, 2015.

Seitzinger, S., Harrison, J. A., Böhlke, J. K., Bouwman, A. F., Lowrance, R., Peterson, B., Tobias, C., and Van Drecht G.: Denitrification across landscapes and waterscapes: a synthesis, Ecol Appl., 6, 2064-90, 2006.

Seitzinger, S. P., Mayorga, E., Bouwman, A. F., Kroeze, C., Beusen, A. H. W., Billen, G., Van Drecht, G., Dumont, E.,
Fekete, B. M., Garnier, J., and Harrison, J. A.: Global river nutrient export: A scenario analysis of past and future trends,
Global Biogeochem. Cycles, 24, GB0A08, 2010.

Sheffield, J., Goteti, G., and Wood, E. F.: Development of a 50-year high resolution global dataset of meteorological
forcings for land surface modeling, J. Clim., 19, 3088–3111, 2006.

Shevliakova, E., Pacala, S. W., Malyshev, S., Hurtt, G. C., Milly, P. C. D., Caspersen, J. P., Sentman, L. T., Fisk, J. P.,
Wirth, C., and Crevoisier, C.: Carbon cycling under 300 years of land use changes: Importance of the secondary vegetation
sink, Glob. Biogeochem. Cy., 23, GB2022, 2009.

Sipler, R. E. and Bronk, D. A.: Biogeochemistry of Marine Dissolved Organic Matter, in: Biogeochemistry of Marine
Dissolved Organic Matter, edited by: Hansell, D. A. and Carlson, C. A., Academic Press, Kidlington, Oxford, UK, 128-233,
2014.

Smith, V. H.: Eutrophication of freshwater and coastal marine ecosystems a global problem, Environ. Sci. Pollut. Res., 10,
126–139, 2003.

Smith, S. V., Swaney, D. P., Talaue-Mcmanus, L., Bartley, J. D., Sandhei, P. T., McLaughlin, C. J., Dupra, V. C., Crossland,
C. J., Buddemeier, R. W., Maxwell, B. A., and Wulff, F.: Humans, Hydrology, and the Distribution of Inorganic Nutrient
Loading to the Ocean, BioScience, 53, 234-245, 2003.

Steele, J. H. and Henderson, E. W., The significance of interannual variability, in: Towards a Model of Ocean
Biogeochemical Processes, edited by: Evans, G. T. and Fasham, M. J. R., Springer-Verlag, Berlin, Heidelberg, Germany,
227–260, 1992.

Stock, C. A., Dunne, J. P., and John, J. G.: Global-scale carbon and energy flows through the marine planktonic food web:
An analysis with a coupled physical–biological model, Prog. Oceanogr., 120, 1-28, 2014.

Syvitski, J. P. M., Vörösmarty, C. J., Kettner, A. J., and Green, P.: Impact of humans on the flux of terrestrial sediment to the
global coastal ocean, Science, 308, 376–380, 2005.

Tan, Z., Leung, L. R., Li, H., Tesfa, T., Vanmaercke, M., Poesen, J., Zhang, X., Lu, H., and Hartmann, J.: A Global data
analysis for representing sediment and particulate organic carbon yield in Earth System Models, Water Resour. Res., 53,
10674–10700, 2017.

Thomas, S. C. and Martin, A. R.: Carbon Content of Tree Tissues: A Synthesis, Forests, 3, 332-352,
https://doi.org/10.3390/f3020332, 2012.

Tian, H., Yang, Q., Najjar, R. G., Ren, W., Friedrichs, M. A. M., Hopkinson, C. S., and Pan, S.: Anthropogenic and climatic
influences on carbon fluxes from eastern North America to the Atlantic Ocean: A process-based modeling study, J. Geophys.
Res. Biogeosci., 120, 757– 772, doi: 10.1002/2014JG002760, 2015.

Tian, H., Xu, R., Pan, S., Yao, Y., Bian, Z., Cai, W.-J., Hopkinson, C.S., Justic, D., Lohrenz, S., Lu, C., Ren, W., and Yang,
J.: Long-term trajectory of nitrogen loading and delivery from Mississippi River Basin to the Gulf of Mexico, Global
Biogeochem. Cycles 34, e2019GB006475, 2020.

Trainer, V. L., Cochlan, W. P., Erickson, A., Bill, B. D., Cox, F. H., Borchert, J. A., and Lefebvre, K. A.: Recent domoic acid closures of shellfish harvest areas in Washington State inland waterways, Harmful Algae, 6, 449–459, 2007.

Turowski, J. M., Rickenmann, D., and Dadson, S. J.: The partitioning of the total sediment load of a river into suspended load and bedload: A review of empirical data, Sedimentology, 57, 1126–1146, doi:10.1111/j.1365-3091.2009.01140.x, 2010.

Van Meter, K. J. and Van Cappellen, P., and Basu, N. B.: Legacy nitrogen may prevent achievement of water quality goals in the Gulf of Mexico, Science, 360, 427–30, 2018.

Vilmin, L., Bouwman, A. F. , Beusen, A. H. W., van Hoek, W. J., and Mogollón, J. M.: Past anthropogenic activities offset

dissolved inorganic phosphorus retention in the Mississippi River basin, Biogeochemistry, 161, 157-169, 2022.

Vilmin, L., Mogollón, J. M., Beusen, A. H. W., and Bouwman, A. F.: Forms and subannual variability of nitrogen and phosphorus loading to global river networks over the 20th century, Glob. Planet. Change, 163, 67–85, 2018.

Vilmin, L., Mogollón, J. M., Beusen, A. H. W., van Hoek, W. J., Liu, X., Middelburg, J. J., and Bouwman, A. F.: Modeling process-based biogeochemical dynamics in surface fresh waters of large watersheds with the IMAGE-DGNM framework, J.

Adv. Model. Earth Syst., 12, 1-19, 2020.

Vorosmarty, C. J., Meybeck, M., Fekete, B., Sharma, K., Green, P., and Syvitski, J. P. M.: Anthropogenic sediment retention: major global impact from registered river impoundments, Glob. Planet. Change, 39, 169–190, 2003.

Wade, A. J., Durand, P., Beaujouan, V., Wessel, W. W., Raat, K. J., Whitehead, P. G., Butterfield, D., Rankinen, K., and Lepisto, A.: A nitrogen model for European catchments: INCA, new model structure and equations, Hydrol. Earth Syst. Sci.,

1215 6, 559-582, 2002.

Yao, Y., Tian, H., Shi, H., Pan, S., Xu, R., Pan, N., and Canadell, J. G..: Increased global nitrous oxide emissions from streams and rivers in the Anthropocene, Nat. Clim. Chang., 10, 138–142, https://doi.org/10.1038/s41558-019-0665-8, 2020.

Zhang, H., Lauerwald, R., Regnier, P., Ciais, P., Van Oost, K., Naipal, V., Guenet, B., and Yuan, W.: Estimating the lateral transfer of organic carbon through the European river network using a land surface model, Earth Syst. Dynam., 13, 1119–

1144, https://doi.org/10.5194/esd-13-1119-2022, 2022.

Zhu, Z., Bi, J., Pan, Y., Ganguly, S., Anav, A., Xu, L., Samanta, A., Piao, S., Nemani, R. R., and Myneni, R. B.: Global Data Sets of Vegetation Leaf Area Index (LAI)3g and Fraction of Photosynthetically Active Radiation (FPAR)3g Derived from Global Inventory Modeling and Mapping Studies (GIMMS) Normalized Difference Vegetation Index (NDVI3g) for the Period 1981 to 2011, Remote Sens., 5, 927-948, 2013.