# Peer review of "Linking global terrestrial and ocean biogeochemistry with process-based, coupled freshwater algae-nutrient-solid dynamics in LM3-FANSY v1.0"

_Geoscientific Model Development, 2022_

## Author Comment (AC1)

**Response to the reviewer #2**

We greatly appreciate the reviewer's comments and suggestions. We have made a concerted (and hopefully successful) attempt to address the reviewer's concerns in the revised manuscript.

First, we agree that PON exports from land to rivers should be assumed to occur with eroded soil fluxes. We have thus implemented this suggested approach and have made associated modifications throughout the manuscript.

Second, we agree that the discussion of the uncertainties in modeling soil erosion and associated fluxes to rivers and the ocean, as well as the discussion of uncertainties generally, was not sufficient in the previous manuscript. We have thus substantially extended the discussion and highlighted areas for priority future development.

Third, we agree that evaluation of the simulated algae dynamics was not sufficient in the previous manuscript. We have thus showed that our simulated lake chlorophyll a concentrations generally fall within a range of in-situ measurements from globally distributed lakes and added discussion of the uncertainties associated with the algae dynamics.

Finally, we note several relevant additional modifications made in response to the other reviewer. Sensitivity analyses have been expanded substantially by perturbing all calibrated parameters. We have also added sensitivity simulations changing the fractions of prescribed TN and TP inputs assigned to different N and P species to the previous model component and input manipulations. All analyses have been limited to 2000 (instead of 2010), reflecting the limited availability of several components of required forcing data beyond 2000. For cross-watershed and global evaluations, annual results for the year 1990 are analyzed and presented along with ranges for the years 1990-2000. We have also added time series analyses of nutrient and solid loads from large U. S. rivers for the period ~1963-2000.

In this response, the line numbers correspond to those in the original manuscript before making any modifications. Table, figure, and section numbers have been changed due to additions/removals of tables, figures, and sections. Unless noted, the table, figure, and section numbers correspond to the modified ones. The modified tables and figures are placed at the end of this response.

Sincerely,
Dr. Minjin Lee
Associate Research Scholar
Princeton University, NOAA/Geophysical Fluid Dynamics Laboratory
(on behalf of all co-authors)

**Referee: 2**

*In this study, Lee et al. intend to develop a comprehensive freshwater model FANSY which can simulate the dynamics of algae, solid and nutrient dynamics in inland waters, and the fresh water model has been incorporated into the Land Model LM3. The topic is very important and should be attractive to a broad of readers. The paper is well organized and the methods and materials have been described adequately. This study involves tremendous of model development work. I appreciate the effects of the authors on developing such a model.*

*Nonetheless, I still have some concerns on the algorithms used in this study to simulate the transfer & transformation of sediment, N and P through the inland water systems. In specific:*

*1) To calculate the erosion-induced POC loss from soil to inland waters, why not calculate the POC loss rate based on the soil SOC concentration and the soil erosion rate, especially you are going to develop a process-based model? The empirical equation from Busen et al., 2005 might be too coarse to be used here for calculating the POC loss rate, and it cannot represent the effects of SOC dynamics on the riverine POC flux. In addition, many studies have proved the effectiveness of calculating POC loss rate based on the soil erosion rate and the SOC concentration, such as Tian et al., 2015 (Anthropogenic and climatic influences on carbon fluxes from eastern North America to the Atlantic Ocean: A process-based modeling study) and Zhang et al., 2022 (Estimating the lateral transfer of organic carbon through the European river network using a land surface model. Earth System Dynamics)*

We agree with these points and have implemented your suggested change in approach. Please find the details in our combined response to your main comments below.

*2) PON generally flows together with POC, as POC, PON and POP are all contained in the POM (particulate organic matter). Almost all particulate matter loss from land to rivers is caused by soil erosion, rather than the water drainage. Thus, it does not make sense to calculate PON loss based on the soil water drainage. A more reasonable solution is that, loss of all particulate matters (e.g. sediment, POC, POC, POP) from land to rivers should be calculated based on the soil erosion rate and the concentrations of these matters in soil; loss of all dissolved matters (DOC, DON, DOP) should be calculated based on soil water drainage (i.e. the leaching processes). Tian et al., 2015 & Zhang et al., 2022 can be good references.*

Response to 1) & 2)
We greatly appreciate the reviewer's incisive comments and informative references.

We agree that PON exports from land to rivers should be assumed to occur with eroded soil fluxes. We thus now simulate PON fluxes to rivers based on the simulated soil erosion fluxes

and litter/soil N concentrations, adapting an approach of previous studies that simulate POC fluxes based on soil erosion fluxes and soil C concentrations (Tian et al., 2015; Zhang et al., 2022).

We also now use the simulated PON fluxes to calculate inorganic soil fluxes from the total soil erosion fluxes, so that the eroded soil and PON inputs to rivers are self consistent. This now allows us to use inorganic suspended solids (ISS) in freshwaters as our prognostic variable, and derive total suspended solids (SS, i.e., the sum of ISS and POM) from the combination of freshwater ISS and PON. This requires an assumed ratio of PON to POM in eroded soils (details are provided below). Since the content of POM in eroded soils is generally small, our results proved insensitive to uncertainties in this conversion. The same ratio has been used to convert PON to POM in freshwaters for the purpose of calculating SS to compare with observations.

Since we do not model the freshwater C cycle yet, the corresponding POC fluxes are not explicitly simulated (though we note that an estimate is possible by using the simulated litter/soil C concentrations as done for PON fluxes). Similarly, since LM3 does not yet include terrestrial P dynamics, we have retained externally specified POP fluxes from Beusen et al (2015). We have extended the discussion of the need to pursue advances to provide a more comprehensive and consistent approach to modeling the coupled N, C, and P cycles across the terrestrial and freshwater continuum of LM3-FANSY.

In accordance with these changes, we now have removed equations (8)-(10), and (28), as well as the text describing our previous approach (lines 206-214, 289-302 in our initial submission) and modified Fig. 1.

And have replaced them with (line 162):

**In LM3-FANSY, terrestrial soil erosion is controlled by land surface slope, rainfall, and leaf area index (LAI) based on Pelletier (2012), as described in Eq. (3). N fluxes from terrestrial litter and soils, in the form of PON, in Eq. (4) are simulated based on the simulated soil erosion fluxes and litter/soil N concentrations. This approach is consistent with that employed by several previous modeling studies (Tian et al., 2015; Zhang et al., 2022). The litter/soil concentrations for this purpose are estimated by using litter/soil contents and effective soil depths simulated by LM3 (Gerber et al., 2010).**

**Inorganic soil inputs to rivers are derived from the simulated soil erosion fluxes by subtracting the PON contribution, as described in Eq. (5). This requires an assumed ratio of POM:PON in eroded soils. Previous studies have shown a wide range of C content in tree biomass (~42-61%, Thomas and Martin, 2012) and of C:N ratios in litter and soils (~5-500, Gerber et al., 2010 and references in Gerber et al., 2010's Table S1). This implies that the POM:PON ratio in soil erosion fluxes can also vary significantly. We have found,**

however, that predicted SS loads are insensitive to an order of magnitude variation in the ratio (i.e., 1.39 vs. 139, see Sect. 4.4), because organic contents in eroded soils are generally small. We thus used a POM:PON ratio of 13.9. The same ratio has been used to estimate the contribution of PON to SS in freshwaters, again noting that it is generally a small fraction of SS.

In accordance with these changes, we now have added/modified equations as follows:

$$E = C_1 \cdot \frac{\rho_b}{\rho_w} \cdot S^{5/4} \cdot r \cdot e^{-L} , \tag{3}$$

$$E^{PON} = E \cdot \left(\frac{N_{FL}+N_{SL}+N_{SS}}{h_s \cdot \rho_b}\right), \tag{4}$$

$$E^{ISS} = E - r_{DN,Ero} \cdot E^{PON} \tag{5}$$

where E, $E^{PON}$, and $E^{ISS}$ is terrestrial soil erosion flux (dry matter (D)kg m$^{-2}$ s$^{-1}$), terrestrial PON flux (Nkg m$^{-2}$ s$^{-1}$), and terrestrial inorganic soil erosion flux (Dkg m$^{-2}$ s$^{-1}$) respectively, $C_1$ is a free parameter of soil erosion (unitless), $\rho_b$ is soil bulk density (kgD m$^{-3}$), $\rho_w$ is water density(kg m$^{-3}$), S is slope tan$\theta$, with $\theta$ as hillslope angle (unitless), r is rainfall (kg m$^{-2}$ s$^{-1}$), L is LAI (unitless), $N_{FL}$, $N_{SL}$, and $N_{SS}$ is N content in fast litter, slow litter, and slow soil pool respectively (kgN m$^{-2}$), $h_s$ is effective soil depth (m), and $r_{DN,Ero}$ is a POM-to-PON ratio in eroded fluxes (kgD kgN$^{-1}$).

The equation for ISS which, as described above, is now a prognostic variable, is:

For a batch river and lake system, a mass balance for ISS and Sed is written as:

$$\frac{dISS}{dt} = \begin{cases} \frac{Sed}{dt} & R_\# < 1.2 \\ -\frac{w_S}{z}\left(\frac{1}{dt}+\frac{w_S}{z}\right)^{-1}\frac{ISS}{dt} & R_\# \geq 1.2 \end{cases}, \tag{8}$$

$$\frac{dSed}{dt} = \begin{cases} -\frac{Sed}{dt} & R_\# < 1.2 \\ \frac{w_S}{z}\left(\frac{1}{dt}+\frac{w_S}{z}\right)^{-1}\frac{ISS}{dt} & R_\# \geq 1.2 \end{cases}, \tag{9}$$

where ISS is inorganic suspended solid (kgD), Sed is benthic sediment inorganic solid (kgD), and z is river or lake depth (m).

The conversion of PON to POM in freshwaters for the purpose of calculating SS to compare with observations is as:
$$POM = r_{DN} \cdot PON , \tag{10}$$
$$SS = ISS + POM , \tag{11}$$

**where POM, PON, and SS is particulate organic matter (kgD), particulate organic N (kgN), and suspended solid (kgD) respectively, and $r_{DN}$ is a POM-to-PON ratio in freshwaters (kgD kgN$^{-1}$).**

These modifications have changed model results. For solids, we have recalibrated the free parameter of terrestrial soil erosion ($C_1$). The value was 0.015 and is now 0.012. After this relatively modest change, the modified SS results are very similar to the previous results. Correlations between the measurement-based and simulated SS yields, loads, and concentrations across the 64 rivers were 0.66, 0.77, and 0.67 respectively and now are 0.65, 0.76, and 0.67. The associated Fig. 2 and text in lines 448-449 have been modified as follows:

**Measurement-based and simulated annual SS estimates across 64 rivers are significantly correlated, with Pearson correlation coefficient, r values equal to 0.65 (0.57-0.65) for yields, 0.76 (0.71-76) for loads, and 0.67 (0.67-0.69) for concentrations for the year 1990 (range for the years 1990-2000) (Fig. 2, Table 4).**

The total amount of global river SS loads to the coastal ocean was previously estimated as 9-11 Pg yr$^{-1}$ between 1982-2010, and now are 10 Pg yr$^{-1}$ for the year 1990 and 10-11 Pg yr$^{-1}$ for the years 1990-2000. The associated Table 5 and text in lines 456-459 have been modified as follows:

**The total amount of global river SS loads to the coastal ocean estimated as 10 (10-11) Pg yr$^{-1}$ for the year 1990 (range for the years 1990-2000) by LM3-FANSY is at the lower bond of previous estimates (Table 5, Global NEWS estimates of 11-27 Pg yr$^{-1}$, Beusen et al., 2005; Discharge Relief Temperature sediment delivery model (QRT) estimate of 12.6 Pg yr$^{-1}$, Syvitski et al., 2005).**

For N, the changes associated with the modified PON inputs to rivers are most manifest in river PON and TKN (i.e., the sum of PON, DON, and $NH_4^+$) loads.

Correlations between the measurement-based and simulated TKN yields, loads, and concentrations across the 11 rivers were 0.57, 0.77, and 0.59 respectively and now are 0.49, 0.70, and 0.62. Global river PON loads to the coastal ocean have been reduced substantially from 14 (13-16) to 7 (7-8) for the year 1990 (range for the years 1990-2000). This has largely resulted from reduced PON inputs to rivers from 14 (14-16) to 4, which is likely at a lower bound, as both the simulated soil erosion fluxes and litter/soil N concentrations are at the lower bounds of published ranges. This result is further discussed later in this response and within the extended discussion. The associated Fig. 3, Tables 4-5, and text in lines 493-502 have been modified as follows:

**Recent estimates of global river TN loads to the coastal ocean vary widely, ranging from about 36.5 to 48 TgN/yr (Table 5, Beusen et al., 2016; Boyer et al., 2006; Galloway et al., 2004; Green et al., 2004; Mayorga et al. 2010). Our global estimate 35 (34-38) TgN/yr for the year 1990 (range for the years 1990-2000) is at the lower bound of the published range. The simulated global river TN loads contain approximately equal contributions by DIN (the sum of $NO_3^-$ and $NH_4^+$, 14 (14-16) TgN $yr^{-1}$, 41% of TN) and DON (13 (13-14) TgN $yr^{-1}$, 39% of TN), with a lesser contribution by PON (7 (7-8) TgN $yr^{-1}$, 20% of TN). The estimates of global river DIN loads are at the lower end of recent estimates, which range from 14.5 to 18.9 TgN $yr^{-1}$ (Mayorga et al. 2010; Green et al., 2004; Smith et al., 2003). This may be partly due to an overestimate of freshwater denitrification and/or algae-mediated transformations to organic forms. In contrast, our global river DON load estimate is slightly higher than a previous estimate 10 TgN $yr^{-1}$ (Harrison et al., 2005). Our global river PON load estimate is considerably lower than a previous estimate, 13.5 TgN $yr^{-1}$ (Mayorga et al. 2010). See Sect. 4.5 for further discussion.**

The discussion of the need to provide a more comprehensive and consistent approach of modeling the coupled N, C, and P cycles in the terrestrial and freshwater continuum of LM3-FANSY was in lines 670-672 (which has been moved under a newly added subsection Sect. 4.5 in the modified manuscript) and reads as follows:

**In addition, there is a need to pursue advances to provide a more comprehensive and consistent approach of modeling the coupled N, C, and P cycles across the terrestrial and freshwater continuum of LM3-FANSY. Expansion of LM3 to include terrestrial P dynamics will be targeted to improve estimates of litter/soil P storage and fluxes to streams and rivers, generating mechanistically consistent estimates of N:P ratios of nutrient loads reaching coastal systems. Priority enhancements will also include integration of freshwater C and alkalinity dynamics with the current solid, algae, and nutrient dynamics of FANSY to simulate the impacts of river inputs on coastal C budgets and acidification.**

The following references are newly added.

**Tian, H., Yang, Q., Najjar, R. G., Ren, W., Friedrichs, M. A. M., Hopkinson, C. S., and Pan, S.: Anthropogenic and climatic influences on carbon fluxes from eastern North America to the Atlantic Ocean: A process-based modeling study, J. Geophys. Res. Biogeosci., 120, 757– 772, 2015.**

**Zhang, H., Lauerwald, R., Regnier, P., Ciais, P., Van Oost, K., Naipal, V., Guenet, B., and Yuan, W.: Estimating the lateral transfer of organic carbon through the European river network using a land surface model, Earth Syst. Dynam., 13, 1119–1144, https://doi.org/10.5194/esd-13-1119-2022, 2022.**

*3) To my knowledge, riverine DON mostly originated from soil during the leaching processes. Decomposition of PON and algae mortality only contribute a small part of the total DON. However, the FANSY model in this study assumed all DON is originated from the decomposition of PON and algae mortality. This assumption does not make sense. I would suggest the authors to add the fluxes of DOC, DON and DOP from land to river during the leaching processes. Many studies have revealed that the fluxes of dissolved matters caused by leaching is much larger than the loss of particulate matters caused by soil erosion (Regnier et al., 2022; Nature, The land-to-ocean loops of the global carbon cycle)*

We apologize for the misunderstanding: LM3 simulates soil $NO_3^-$, $NH_4^+$, and DON leaching. We have clarified input fluxes from terrestrial system and the atmosphere to rivers in Fig. 1 and on lines 114-116. Consistent with the reviewer's view, river DON loads to the coastal ocean estimated as 13 TgN yr$^{-1}$ are mostly originated from soil DON leaching (49 TgN yr-1) and the sum of soil $NO_3^-$, $NH_4^+$, and DON leaching (74 TgN yr-1) is much greater than the PON flux to rivers (4 TgN yr-1).

As noted in the previous response, since we do not model the freshwater C cycle yet, we did not explicitly simulate soil DOC leaching, and POP leaching is handled through input estimates from Beusen et al (2015). This simplification is largely because the land model, LM3, does not yet include a P component.

*4) I am surprised that the model does not consider the emission of N2O from inland waters. Both observation and simulation indicated that a larger part of the riverine N is loss to air in the form of N2O (Yao et al., Nature Climate Change, Increased global nitrous oxide emissions from streams and rivers in the Anthropocene)*

We have included denitrification emissions from freshwaters, but have not included processes of explicitly separating $N_2O$ and $N_2$ emissions from the freshwater denitrification. For this initial model development, simulating the total amount of N removal to the atmosphere via the freshwater denitrification has been sufficient for the purpose of closing the freshwater N budget and estimating the model's capacity to capture globally distributed N and P loads to the coastal ocean. We agree that freshwater $N_2O$ emission has been recognized as an increasingly important greenhouse gas source, and thus $N_2O$ modeling should be subject to future work.

We have added associated text in line 670 (which has been moved under a newly added subsection Sect. 4.5 in the modified manuscript) as follows:

**There is also significant room for further model development and improvement. The current version of LM3-FANSY simulates denitrification emissions from freshwaters, but does not include processes that explicitly separate $N_2O$ and $N_2$ emissions from the freshwater denitrification. As freshwater $N_2O$ emissions have been recognized as an increasingly important greenhouse gas source (Yao et al., 2020), it will be important to differentiate $N_2$ and $N_2O$ emission processes in future work.**

The following reference is newly added.

**Yao, Y., Tian, H., Shi, H. et al.: Increased global nitrous oxide emissions from streams and rivers in the Anthropocene, Nat. Clim. Chang., 10, 138–142,https://doi.org/10.1038/s41558-019-0665-8, 2020.**

*5) Many of the forcing data used in this study were obtained from Beusen et at., 2015. It is better to provide more information about these data. Foe example, the spatial and temporal resolutions of these data, and the method used by Beusen et al., 2015 to produce these data.*

We have extended the description of data from Beusen et al (2015) in lines 402-403 as follows:

**Solid and nutrient inputs from terrestrial systems and from the atmosphere to rivers are either simulated by LM3-FANSY or provided by Beusen et al (2015) (Table 3). For N, all inputs were simulated by LM3-FANSY except aquaculture, wastewater, and atmospheric deposition, which were provided by Beusen et al (2015). For P, which is not currently included in LM3, all inputs were provided by Beusen et al (2015).**

**Beusen et al (2015) provided five-year interval data for the period 1900-2000 at 0.5 degree resolution. The data were regridded to our 1 degree resolution by summing up the values given in kg yr$^{-1}$ and linearly interpolated across the five-year intervals. Beusen et al (2015)'s wastewater N and P inputs were from Morée et al (2013)'s urban waste N and P discharge estimates to surface waters. Beusen et al (2015) calculated aquaculture N and P inputs using Bouwman et al (2013)'s finfish and Bouwman et al (2011)'s shellfish data. For atmospheric N deposition inputs, Beusen et al (2015)'s input for the year 2000 was from Dentener et al (2006)'s ensemble of reactive-transport models and those for the years before 2000 were made by scaling the deposition with Bouwman et al (2013)'s ammonia emissions. Beusen et al (2015)'s surface runoff P inputs include those leached from soil P budgets (i.e., the sum of fertilizer and animal manure minus crop and grass withdrawal) and those driven by soil erosion estimates based on Cerdan et al (2010). Beusen et al (2015)'s P inputs of litter from floodplains were estimated as 50% of total NPP with a C:P**

ratio of 1200. Beusen et al (2015)'s weathering P inputs were computed based on Hartmann et al (2014)'s chemical weathering P release estimates.

The following references are newly added.

Bouwman, A. F., Beusen, A. H. W., Overbeek, C. C., Bureau, D. P., Pawlowski, M., and Glibert, P. M.: Hindcasts and future projections of global inland and coastal nitrogen and phosphorus loads due to finfish aquaculture, Rev. Fish Sci., 21, 112–156, doi:10.1080/10641262.2013.790340, 2013.

Bouwman, A. F., Klein Goldewijk, K., Van der Hoek, K. W., Beusen, A. H. W., Van Vuuren, D. P., Willems, W. J., Rufino, M. C., and Stehfest, E.: Exploring global changes in nitrogen and phosphorus cycles in agriculture induced by livestock production over the 1900–2050 period, P. Natl. Acad. Sci. USA, 110, 20882–20887, doi:10.1073/pnas.1012878108, 2013.

Bouwman, A. F., Pawłowski, M., Liu, C., Beusen, A. H. W., Shumway, S. E., Glibert, P. M., and Overbeek, C. C.: Global hindcasts and future projections of coastal nitrogen and phosphorus loads due to shellfish and seaweed aquaculture, Rev. Fish Sci., 19, 331–357, doi:10.1080/10641262.2011.603849, 2011.

Cerdan, O., Govers, G., Le Bissonnais, Y., Van Oost, K., Poesen, J., et al.: Rates and spatial variations of soil erosion in Europe: A study based on erosion plot data, Geomorphology, 122, 167–177, doi:10.1016/j.geomorph.2010.06.011, 2010.

Dentener, F., Stevenson, D., Ellingsen, K., Noije, T. v., Schultz, M., et al.: The global atmospheric environment for the next generation, Environ. Sci. Technol., 40, 3586–3594, 2006.

Hartmann, J., Moosdorf, N., Lauerwald, R., Hinderer, M., andWest, A. J.: Global chemical weathering and associated P-release – The role of lithology, temperature and soil properties, Chem. Geol., 363, 145–163, doi:10.1016/j.chemgeo.2013.10.025, 2014.

Morée, A. L., Beusen, A. H. W., Bouwman, A. F., and Willems, W. J.: Exploring global nitrogen and phosphorus flows in urban wastes during the twentieth century, Global Biogeochem. Cy., 27, 1–11, doi:10.1002/gbc.20072, 2013.

*6) Discussion on the uncertainties of this study is weak. Consider the simplification and ignorance of many processes that are closely related to the transfers of sediment, C, N & P. There should be more discussion on the potential uncertainties in the simulated results. For example, the uncertainties in simulated soil erosion rate, POC loss rate and the effects of dams and reservoirs.*

We agree that the discussion and analyses of uncertainties generally, and in particular those of modeling soil erosion and associated fluxes to rivers and the coastal ocean were not sufficient in the previous manuscript.

In the modified manuscript, we have ensured that the uncertainties and simplifications are highlighted through the Methods and Results, and we have extended the discussion of the uncertainties, including the specific issues the reviewer has raised, by adding a new subsection 4.5 as follows:

[revised manuscript text omitted]

*7) Compared to existing models, a highlight point of the FANSY model in this study is that it explicitly simulated the biogeochemical cycles related to the algae. However, there is no evaluation of the simulated algae processes, and it is hard for the readers to determine whether the scheme used in the FANSY model is effective or reliable on simulating algae dynamics. Is it possible for the authors to collect some observation data to validate the simulated algae biomass, growth rate or mortality rate in this study.*

A detailed comparison of LM3-FANSY results against observed chlorophyll a is difficult, because there are no compiled estimates of chlorophyll a in rivers. While Sayers et al (2015) provides a compilation of lake chlorophyll a estimates, many lakes are small and those that are large can have highly heterogenous chlorophyll a distributions. This poses a challenge, given the relatively simple nature of lake biogeochemistry in LM3-FANSY (i.e., vertically unresolved mixed reactors, Lee et al., 2019) and the current coarse global resolution. However, in the Results, we now provide a map of the simulated lake chlorophyll a concentrations (Fig. SI3) and confirm that the range of simulated values are comparable to that in Sayers et al (2015). We have also expanded our sensitivity analysis of algae dynamics to better elucidate the controls that algae exert on the ratios of inorganic to organic nutrients. Finally, we have identified an improvement of the lake representation as a future development priority (please refer to our response to your previous comment 6).

The chlorophyll a result has been added in Sect. 4.1 Model performance analysis, as follows:

**Despite the relatively simple nature of lake biogeochemistry in LM3-FANSY (i.e., vertically unresolved mixed reactors, Lee et al., 2019), the model creates a reasonable range of chlorophyll a concentrations (Fig. SI3) that generally fall within a range of in-situ estimates from globally distributed lakes (Sayers et al., 2015). The in-situ estimates in the compilation of Sayers et al. (2015) range from ~0 to ~100 mg m$^{-3}$, mostly falling between 5 and 50 mg m$^{-3}$ (Fig. 5 of Sayers et al. (2015), available at https://www.tandfonline.com/doi/full/10.1080/01431161.2015.1029099).**

We note that we have slightly decreased algal mortality rate constant from 1.0 10-5 to 0.8 10-5 kgN$^{-1/3}$s$^{-1}$. The text describing the expanded sensitivity tests, highlighting the algae controls on partitioning of inorganic and organic nutrients (line 651) is as follows:

**Algae dynamics play a significant role in determining the relative composition of inorganic vs. organic nutrients in freshwaters. Decreasing the algal mortality rate constant by half enhances algal uptake, decreasing DIN (the sum of $NO_3^-$ and $NH_4^+$) and IP (the sum of $PO_4^{3-}$ and PIP) by -11% and -33% respectively, while it increases ON (the sum of DON and PON) and OP (the sum of DOP and POP) by 24% and 33% respectively. Similarly, increasing the maximum photosynthesis rate or chlorophyll a-specific initial slope of the photosynthesis-light curve by twice enhances algal uptake, decreasing DIN by -6 or -7% and IP by -21% or -25% respectively, while it increases ON by 18% or 21% and OP by 23% and 28% respectively. The opposite holds for the parameter changes that reduce algal uptake. An analysis of the model sensitivity simulations, wherein the dynamic contributors to light extinction (i.e., ISS, POM, and CHL in Eq. (23)) were removed, further suggests that the proper light limitation of algal growth is also important for skillful estimates of freshwater inorganic vs. organic nutrients. Removing the dynamic light shading component leads to a ~26 % and ~35% overestimation of ON and OP loads and an underestimation of DIN and IP loads by ~10 % and ~32% respectively. Inorganic nutrient levels are suppressed by invigorated algal populations and more nutrients end up in organic forms. Algal controls thus offer an effective means of calibrating the mix of inorganic and organic constituents.**

The following reference is newly added.

**Sayers, M. J., Grimm, A. G., Shuchman, R. A., Deines, A. M., Bunnell, D. B. et al.: A new method to generate a high-resolution global distribution map of lake chlorophyll, Int. J. Remote Sens., 36, 1942-1964, DOI: 10.1080/01431161.2015.1029099, 2015.**

*Minor comments:*

*In Eq. 3, how is the slope extracted? Is the slope extracted from DEM at 1-degree resolution?*

The slope was created by using the Global Multi-resolution Terrain Elevation Data 2010 (GMTED 2010) at 1/10-degree resolution, but we had to regrid it to 1-degree resolution by averaging out to match the model resolution.

Danielson, J. J. and Gesch, D. B.: Global multi-resolution terrain elevation data 2010 (GMTED2010): U.S. Geological Survey Open-File Report 2011–1073, https://pubs.usgs.gov/of/2011/1073/pdf/of2011-1073.pdf, 2011.

*In Eq. 4, how the depth of river and lake is calculated? Do you have a map of area and lake as the forcing data of your model? If yes, is the area of river or lake fixed?.*

Shallow lake depths were fixed as 2 m. For large lakes, the depths were calculated by dividing lake volumes by lake surface areas based on data from van der Leeden et al. (1990). Maximum depths calculated in this way were limited to 50 m. Lake areas and lake area fractions of grid cells were prescribed, defined on the basis of the U. S. Geological Survey Global Land Cover Characteristics Database "IGBP Water Bodies" field (Please see the maps attached in this response).

River lengths were prescribed, but river depths, widths, and velocities were simulated at every time step (30 minutes) based on simulated river flows, yielded from hydraulic geometric relations of Leopold and Maddock (1953).

A detailed description of river and lake hydrology and hydrography can be found in Milly et al. (2014).

[revised manuscript text omitted]

**Figure SI3. Chlorophyll a concentrations (mg m$^{-3}$) in lakes for the year 1990.**

[Figure]

**Figure SI6: Pearson correlation coefficients (r) and p values (p) between the measurement-based vs. simulated yields, loads, and concentrations across the world major rivers for the year 1990. Here the actual (instead of natural) water discharges of Meybeck and Ragu (2012) were used when calculating loads and yields from their multi-year average concentrations.**

**Lake areas and lake area fractions of grid cells (in response to the last minor comment).**

**Lake Area, 106 km2**

[Figure]

**Lake Fraction of Grid Cell (in different color scales)**

---

## Author Comment (AC2)

**Responses to the reviewer 1**

We greatly appreciate the reviewer's comments and suggestions. We have made a concerted (and hopefully successful) attempt to address the reviewer's concerns in the revised manuscript.

First, we have added time series evaluations of annual solid and nutrient loads from large U. S. rivers for the period ~1963-2000 against measurement-based estimates from the USGS National Water Quality Network.

Second, we have substantially expanded sensitivity/uncertainty analyses by perturbing all calibrated parameters and perturbing the fractions of dividing externally prescribed TN and TP inputs into different N and P forms, in addition to the previous manipulations of model components and inputs.

Third, we have modified analyzed years. For cross-watershed and global evaluations, annual results for the year 1990 are analyzed and presented. In parenthesis, ranges of using annual results for the years 1990-2000 are also provided throughout the manuscript.

Fourth, as the other reviewer 2 suggested, we have modified the model to simulate PON fluxes from terrestrial litter and soils, based on the simulated soil erosion fluxes and litter/soil N concentrations and have made associated modifications throughout the manuscript.

Finally, we have substantially extended the discussion of the uncertainties inherent in this initial development of the LM3-FANSY modeling framework and the need to pursue advances to provide a more comprehensive and consistent approach of modeling the coupled solid, N, C, and P cycles across the terrestrial and freshwater continuum. In accordance with this forward looking perspective on P, and the reviewer's suggestions, we have de-emphasized and strongly caveated our discussion of river N:P ratios.

In this response, the line numbers correspond to those in the original manuscript before making any modifications. Table, figure, and section numbers have been changed due to additions/removals of tables, figures, and sections. Unless noted, the table, figure, and section numbers correspond to the modified ones. The modified tables and figures are placed at the end of this response.

Sincerely,
Dr. Minjin Lee
Associate Research Scholar
Princeton University, NOAA/Geophysical Fluid Dynamics Laboratory
(on behalf of all co-authors)

**Referee: 1**

*This paper presents the development of a coupled river (and lake) biogeochemical model for N and P on the global scale, based on the coupling of LM3, the freshwater module of FANSY. The P input and some of the N sources are taken from another data source (IMAGE-GNM).*

*The authors claim that the new model FANSY is incorporated in the Land Model (LM3). This coupling results in a more process-based representation of biochemistry model FANSY-LM3 which links the terrestrial and ocean biochemistry. The purpose of the LM3 model: "It captures processes including changes in vegetation functioning, plant-soil nitrogen cycling, and streamflow dynamics. The land model sheds light on the effects of these processes on atmospheric physics and chemistry and, conversely, the effects of changing climate and CO2 concentration on the land." (from https://www.gfdl.noaa.gov/land-model/). It is clearly a goal of this coupled system (FANSY-LM3) to capture changes and streamflow dynamics. The FANSY model has indeed a lot of dynamics incorporated (and are described here), so this model is well suited to couple with the LM3 model. However, the results that are presented here are not dynamic, but are on one single time period. How can the reader discover whether FANSY is capable of producing reasonable results when it is used in combination with the LM3 model? If the purpose of the model only is to simulate the chosen time period, then I would not take this model, but the Global NEWS model. The advantage of the Global NEWS model is that it is published and well cited and the results are very transparent! The current presentation of the result does not convince me to start using a dynamic process-based model.*

First of all, we agree that time series evaluations would be a valuable addition to this model development study and have added time series analysis for large U. S. rivers where long time series of high quality data are available (see our responses below). We point out, however, that evaluation of model outputs from disparate globally-distributed systems under today's climate and land use forms a core component of the development of Earth System Models used for applications of climate and land use changes, with the widely accepted premise that a model's capacity to represent the range of outcomes across disparate systems spanning a range of climates and anthropogenic drivers has a bearing on its capacity to project changes over time.

To outline our thought process, we use a general guideline that building confidence in models used for global climate projections is a progressive process that rests on two pillars: 1) the construction of models from mechanistic principles likely to hold in novel environments expected under climate change, and 2) their ability to recreate features in the current climate and/or past climate changes (Randall et al., 2007). Our inclusion of a relatively comprehensive set of process-based dynamics attempts to advance the first confidence pillar (though we recognize that we are still only scratching the surface of the full complexity of the system and

have strengthened our discussion of the significant simplifications still required in accordance with the reviewer's later suggestions and those of reviewer 2). For the second pillar, we chose to focus on diverse solid and nutrient loads across disparate globally-distributed systems with different climates and other anthropogenic drivers. This focus on a chosen time spatial variation is common in validating new dynamic models added as a prognostic component of Earth System Models (e.g., Zhang et al., 2020), merged land-river model similarly to LM3-FANSY (e.g., He et al., 2011), and Earth System Model evaluation metrics more broadly (e.g., Dunne et al., 2020). It reflects the fact that the strongest current nutrient load constraints come from cross-system comparisons in today's climate.

We have clarified the underlying model evaluation rationale in the Methods section (Sect. 3.3). With the addition of the (admittedly limited) time series analysis and other additions suggested by the reviewers, we feel that our model evaluations are as thorough as any other published global river solid or nutrient modeling studies. We have furthermore noted in the Discussion the need for additional time varying constraints to build additional confidence in projected changes.

Secondly, the goal of FANSY is not to replace the Global NEWS model. Rather, the goal is to develop an initial module to be added as a prognostic component of an Earth System Model to link global terrestrial and ocean biogeochemistry. We anticipate that similar models will be developed for many global Earth System Models used for climate change projections in the coming years. This purpose motivated our efforts to advance the process resolution of the model relative to more empirical approaches, but we recognize that adding process-level constraints can degrade model fit. We thus feel that it is important to present both the overall skill and the skill relative to Global NEWS which, as the reviewer has stated, is a widely cited and pioneering community benchmark. We emphasize, however, that this is in no way an attempt to replace Global NEWS, but a necessary step toward our ultimate goal of integration of FANSY within an Earth System Model framework used for climate projections, and we have taken steps to clarify these points in the Introduction and Methods. We feel that this progressive approach is consistent with the goals of Geoscientific Model Development.

Lastly, as described several places in the manuscript, the merged land-freshwater framework of LM3-FANSY is built on the existing merged land-freshwater N framework of LM3-TAN, by integrating the new freshwater algae, solid, and P dynamics into it. Seasonal to centurial time series results of LM3-TAN have been extensively validated against observational estimates at both global and regional scales (Lee et al., 2014; 2016; 2018; 2019, 2021). The validated results have showed simulated vegetation and litter/soil N storage changes to significantly alter multi-decadal river N trends (Lee et al., 2019) and coastal ecosystem productivity and C exports (Liu et al., 2021), as well as seasonal to multi-year river N extremes (Lee et al., 2016; Lee et al., 2021).

Dunne, J. P., et al.: The GFDL Earth System Model version 4.1 (GFDL-ESM 4.1): Overall coupled model description and simulation characteristics, JAME, 12(11), DOI:10.1029/2019MS002015, 2020.

Randall, D. A., et al: Cilmate Models and Their Evaluation. In: Climate Change 2007: The Physical Science Basis. Contribution of Working Group I to the Fourth Assessment Report of the Intergovernmental Panel on Climate Change [Solomon, S., D. Qin, M. Manning, Z. Chen, M. Marquis, K.B. Averyt, M.Tignor and H.L. Miller (eds.)]. Cambridge University Press, Cambridge, United Kingdom and New York, NY, USA, 2007.

He, B., Kanae, S., Oki, T., Hirabayashi, Y., Yamashiki, Y., and Takara, K.: Assessment of global nitrogen pollution in rivers using an integrated biogeochemical modeling framework, Water Research, 45, 2573-2586, doi.org/10.1016/j.watres.2011.02.011, 2011.

Zhang, H. et al.: Microbial dynamics and soil physicochemical properties explain large-scale variations in soil organic carbon, Glob Change Biol, 26: 2668–2685, https://doi.org/10.1111/gcb.14994, 2020.

*Also the sensitivity analyses are focused on the Pearson correlation coefficient of that same time period on a limited set of rivers. This approach does make it very hard to understand how the model is reacting on different inputs.*

The sensitivity analyses are based on both 1) the Pearson correlation coefficient (Table 4 in the original manuscript), which tests the impact of a parameter, input, or component change on the spatial agreement between the simulated and observed relative river loads, and 2) the amount of river solid and nutrient loads (Table 5 in the original manuscript), which tests the effect of a change on the magnitude of the river loads. We have emphasized this in the discussion of the load differences between the baseline and sensitivity simulations to show the effects of different inputs (e.g. lines 640-643, 651-658).

We have emphasized that the 69 rivers are the world's major rivers influenced by a broad range of climate and land use (lines 435-437). The 69 river basins cover 55% of global land area (excluding the Antarctic) and their broad global distribution and coverage is now illustrated in Fig. SI1.

*It is not clear to me, why these 64 basins are chosen and whether they are a good representation of the global river basins.*

The total number of rivers considered for our analysis is 69 (not 64). As noted above, the 69 river basins cover 55% of global land area (excluding the Antarctic) across a broad range of climate

and land use. We have extended the description of how the 69 rivers were chosen later in this response.

***Also the use of different time periods to do this comparison is creating all kinds of uncertainties.***

We have modified the years for the analyses to consistently present values for the year 1990 and ranges for the years 1990-2000 throughout the manuscript. As suggested by the reviewer, this change was made to be consistent with the time limit of the model forcing.

***I would suggest to do a proper sensitivity analyses (not a one factor analysis but change all parameters at once) on the important model output parameters to show and understand the mechanistic behavior of the model.***

We have substantially expanded sensitivity/uncertainty analyses by changing all calibrated parameters and by changing the fractions of dividing the externally prescribed TN and TP inputs into different N and P species, in addition to the previous model component and input manipulations. This has increased the total number of sensitivity/uncertainty simulations from 22 to 87, requiring 4611 years of total sensitivity/uncertainty simulations. Discussion of the insights gained by these expanded simulations has also been extended considerably.

We do not feel that Monte Carlo experiments changing all parameters at once are appropriate at this initial development stage, because 1) it is challenging, if not impossible, to robustly attribute a response to a parameter or process when one is changing everything at once and 2) many of the simulations arising from such experiments proved implausible when compared with observations. We agree that such approaches are proper when constructing an ensemble of similarly plausible model permutations, but only after proper filtering for model fidelity and once the more basic single factor sensitivities are understood.

We feel that the significant expansion of single factor sensitivities/uncertainties and discussion made in response to the reviewer's comments and those of reviewer 2 have strengthened this model description paper considerably in this latter regard.

***The current validation of the model is not good enough and is not convincing.***

As emphasized in our responses to the reviewer's general comments above and in more detail below, we have included quantitative cross-watershed evaluations of solids, N, and P, in different forms (SS, $NO_3^-$, $NH_4^+$, DIN, DON, TKN, $PO_4^{3-}$, DOP, and TP) and different units (yields, loads, and concentrations), across the 69 globally distributed rivers covering 55% of the land surface (excluding the Antarctic). We have augmented the evaluations further with time

series comparisons suggested by the reviewer and expansive sensitivity analysis. This frankly meets or exceeds the evaluations presented in comparable model description papers in GMD and elsewhere (e.g., Beusen et al., 2015, He et al., 2011, Pelletier, 2012). We also note that this is a model description paper, not just a model evaluation paper.

***Additional validation on long term data sets is needed to show that the dynamic patterns are well simulated.***

We have added time series evaluations of annual solid and nutrient loads from large U. S. rivers for the period ~1963-2000 against reported data from the USGS National Water Quality Network (NWQN, Lee, 2022). We still feel, however, that the global comparison of constituents and quantities across multiple systems, provides the strongest test of the model and the best initial basis for understanding the strengths and limitations of this model for global applications.

A description of the reported time series data has been added in Sect. 3.3 as follows:

[revised manuscript text omitted]

*I would advise to reject the current manuscript or major revisions under the condition that there will be a better sensitivity analyses including a more extensive validation over a longer time period.*

We have substantially extended sensitivity/uncertainty analyses and added time series evaluations of river solid and nutrient loads over long time periods ~1963-2000. We are confident that our documentation and evaluation now meets or exceeds those provided by other model development papers.

*Major comments*

*1.      It is very important to know if the calibration data and validation overlap. However, details about calibration data are absent in this paper .*

We have clarified that all data were included when evaluating and coarsely calibrating the model. The model evaluation thus tests the degree to which the model and forcing can explain the observed patterns across disparate watersheds and nutrient species, allowing for coarse calibration within the constraints of the model structure and parameters. Explicit separation of calibration and validation datasets is often omitted for global watershed model development. The examples include studies that evaluate simulated TN and TP concentrations (IMAGE-GNM, Beusent et al., 2015), nitrate loads and yields (He et al., 2011), and suspended solid yields (Pelletier, 2012).

We further note that LM3-FANSY has been evaluated against three independent datasets: the GEMS-GLORI world river discharge database (Meybeck and Ragu, 2012), Global NEWS (Mayorga et al., 2010), and, in the modified manuscript, time series data from the USGS National Water Quality Network (NWQN, Lee, 2022).

We now clarify these aspects, including the description of how the 69 rivers were chosen, in the Methods section (Sect. 3.3) as follows:

**3.3 Comparisons of measurement-based and modeled estimates**

**For cross-watershed evaluations, we compare LM3-FANSY results of river SS, $NO_3^-$, $NH_4^+$, DIN (the sum of $NO_3^-$ and $NH_4^+$), DON, TKN (the sum of $NH_4^+$, DON, and PON), $PO_4^{3-}$, DOP, and TP (the sum of $PO_4$, DOP, PIP, and POP) yields (kg km$^{-2}$ yr$^{-1}$), loads (kt yr$^{-1}$), and concentrations (mg l$^{-1}$) with measurement-based estimates from 69 world's major rivers (Table SI1-9, the GEMS-GLORI world river discharge database, Meybeck and Ragu, 2012). The 69 river basins cover 55% of global land area (excluding the Antarctic) and are distributed globally across various climates and land use (Fig. SI1). The same comparisons are made by using the Global NEWS database against the measurement-based estimates, yet excluding a few unavailable rivers in Global NEWS (Table SI1, SI4, SI6-9).**

**The 69 rivers were chosen by the following procedure. First, river basins with areas < 100,000 km$^2$, about 10 grid cells in our 1 degree resolution, were excluded from the comparisons. Second, river locations were identified by matching latitudes, longitudes, and basin areas of the modeled and reported rivers, which are located either at river mouths or inlands. Third, rivers that were not properly represented within the LM3-FANSY river network were excluded. For example, the GEMS-GLORI database provides a Sanaga**

River SS concentration at 3.8°N and 10.1°E with its basin area 119,300 km$^2$. LM3-FANSY, however, does not capture the Sanaga River at a comparable location (3.5°N and 10.5°E), where the river has a much smaller basin area (5,607 km$^2$). The Sanaga River was thus excluded. In total, 7 rivers (i.e., Sepik, Don, Huai, Uruguay, Burdekin, Pyasina, and Sanaga) were excluded for SS comparisons. None of the rivers were excluded for nutrient variables. When a river in the LM3-FANSY river network captures two merged small rivers in the GEMS-GLORI database, water discharge weighted mean concentration of the two rivers was used for analysis. When more than one data were given for a river, the data selected for the first line was used, since it was considered as "the most reliable and generally were obtained first hand by local engineers or scientists (Meybeck and Ragu, 1997)". When more than one data were given for a river with different basin areas, the data monitored at the location with the largest basin area (i.e., nearest to the river mouth) was used.

Since hydraulic controls like damming, irrigation, and diversion affect many rivers, Meybeck and Ragu (2012) distinguish natural river discharges from actual, modified ones. LM3-FANSY does not resolve such hydraulic controls and thus, if available, the natural discharges of Meybeck and Ragu (2012) are used, when calculating loads and yields from their multi-year average concentrations. Comparisons of the model results with loads and yields calculated by using the actual discharges are also presented in Supplementary Information. We report the Pearson correlation coefficient (r) and Nash–Sutcliffe model efficiency coefficient (NSE) between the modeled and measurement-based estimates across the 69 rivers. We also report the prediction error computed as the difference between the modeled and measurement-based estimates of loads expressed as a percentage of the measurement-based load. For global evaluations, we compare LM3-FANSY results with reported global estimates from various references. For cross-watershed and global evaluations, annual results for the year 1990 are analyzed and presented. In parenthesis, ranges of using annual results for the years 1990-2000 are also provided throughout the manuscript.

For the description of time series analysis, please refer to our response to the reviewer's previous comment.

The following reference is newly added.

Meybeck, M. and Ragu, A.: Presenting the GEMS-GLORI, a compendium of world river discharge to the oceans, Freshwater Contamination, 243, 3-14, 1997.

*2.    There should be an overview of the loads that flow into the river, the removal loads to the different sectors (sediments, air, …) and the export to the coastal waters for N and P.*

We have added an overview of the loads in lines 493 and 507 respectively as follows:

**Globally, TN inputs to rivers in LM3-FANSY are 85 (85-91) TgN yr$^{-1}$, of which about 59 (56-59)% are lost to the atmosphere and the other 41 (40-44)% are exported to the coastal ocean.**

**Globally, TP inputs to rivers in LM3-FANSY are 8 (8-9) TgN yr-1, of which about 9 (6-10)% are stored within freshwaters and 91 (90-94)% are exported to the coastal ocean.**

*3.       The consistency of the P input compared to the N input is a concern. As long as this is not clear, I would suggest to avoid to calculate and present the N:P ratio.*

We agree that the inconsistency of the P and N inputs is a concern that should be addressed in future work. We have highlighted this as a development priority in the Discussion (shown in our response to the reviewer's earlier comment).

We also agree that, given the inconsistency pointed out by the reviewer, and the generally modest skill achieved, we had placed too much emphasis on the N:P ratio in the initial draft. This conclusion was reinforced by the finding that the modest skill achieved was highly sensitive to outliers and subtle recalibration. We thus have removed Fig. 5, Fig. 9, and Sect. 4.3 in the original manuscript, and have limited ourselves to making the following statements and including one supplementary figure (Fig. SI4) to support it (lines 518-526):

**While the simulations are reasonably successful in capturing cross ecosystem differences in both N and P species, variations in their ratio prove more challenging. The model captures the mean ratio, but little of the variation (Fig. SI4). One factor that likely contributes to this is the inconsistency between the N inputs to rivers, the majority of which (~92%) were simulated within LM3, and the P inputs, all of which were drawn from another model (IMAGE-GNM). Continued refinement is thus needed to reliably capture N:P ratio variations in rivers and their subsequent water quality implications.**

*4.      In line 438-440 it is clearly states that FANSY does not simulate the processes in reservoirs. This should be discussed including what this means when this model is used in "unprecedented scenarios".*

The associated discussion was previously in lines 673-676. We now have extended it within the newly introduced Sect. 4.5, which provides a consolidated discussion of model uncertainties and priority developments. The text is as follows:

**Uncertainties associated with sediment dynamics and bed load transport are compounded by the relatively simple representation of lakes, and the exclusion of anthropogenic hydraulic controls like damming, irrigation, and diversion that affect many rivers. For model evaluation, if available, we used the natural water discharges of Meybeck and Ragu (2012) when calculating loads and yields from their multi-year average concentrations (see Sect. 3.3). Large dams or reservoirs, however, have been shown to impound solids and nutrients to substantially decrease their loadings to rivers (Vorosmarty et al., 2003). Thus, despite the relatively low global river SS loads in this first implementation of LM3-FANSY, the lack of such sediment trapping may have induced overestimations of solid and nutrient loads from river basins including large dams or reservoirs. As a representative example, the Colorado River Basin is known for nearly complete trapping of solids due to large reservoir construction and flow diversion (Vorosmarty et al., 2003). LM3-FANSY does not capture such an extreme trapping. As a result, the Colorado River SS load simulated by LM3-FANSY (99,235 kt yr$^{-1}$) is more consistent with the corresponding load calculated by using the "natural" water discharge of Meybeck and Ragu (2012) (120,010 kt yr$^{-1}$) than with the load calculated by using the "actual" water discharge (649 kt yr$^{-1}$). Although use of the actual water discharges is found to not significantly alter the cross-watershed evaluations (Fig. SI6), such anthropogenic hydraulic controls are expected to further increase in the future (Seitzinger et al., 2010). It will be thus important to consider the effects of such controls in future work.**

*5.      It is not appropriate to use the measurement data unevenly collected during for 1970s-1990s to validate model results for 1982-2010, especially because there are a lot of published global measurement data available for validation for the period 1982-2010. This introduced high temporal uncertainty may be larger than the model uncertainty itself. Substantial work is needed to re-do the model validation to eliminate this uncertainty.*

The GEMS-GLORI database has been used to evaluate global model results for various years including 1990 (IMAGE-GNM, Beusen et al., 2015), 1995 (He et al., 2011), and 2000 (Global NEWS 2, Mayorga et al., 2010). We previously used the LM3-FANSY results for the year 1990

for cross-watershed and global evaluations and additionally provided the ranges by using the annual results for the years 1982-2010.

We now have modified years for the analyses as 1990-2000 for the cross-watershed and global evaluations, the description of which is summarized in line 442 as follows:

**For cross-watershed and global evaluations, annual results for the year 1990 are analyzed and presented. In parenthesis, ranges of using annual results for the years 1990-2000 are also provided throughout the manuscript.**

*Specific comments*

*Line 22-23: How you do evaluate which are "better global estimates of nutrient inputs to rivers"? As far as I know, there is no way to validate nutrient inputs to rivers, and it's hard to say which is a better estimate.*

We have modified the sentence as follow:

**Analyses of the model results and sensitivities to components, parameters, and inputs suggest that fluxes from terrestrial litter and soils, wastewater, and weathering are the most critical inputs to the fidelity of simulated river nutrient loads to the observation-based estimates.**

*Line 32-38: This sentence is too long, and extends for 7 lines. Please re-organize it into several short sentences to increase readability.*

We have modified ", which in turn have been linked to myriad consequences, including"
to
". The consequences include".

*Line 40: "a comprehensive model" is not clear. In terms of what aspect?*

We have modified "a comprehensive freshwater biogeochemistry model"
to

"a freshwater biogeochemistry model that captures intertwined algae, nutrient, and solid dynamics".

*Line 48 and 52: Term use: NO3- instead of "NO3", same for NH4 and PO4, same for everywhere in the text.*

We have made the suggested modifications throughout the manuscript.

*Line 43-45: "In particular, inorganic nutrients, which are characterized by higher bioavailability than organic forms (Sipler and Bronk, 2004), have been recognized as critical drivers of algal blooms (including non-HABs) and hypoxic events (Kemp et al., 2005)." It is "excessive inorganic nutrients" rather than simply "inorganic nutrients" that may drive algal blooms and hypoxic blooms.*

We have modified "inorganic nutrients"
to
"excessive inorganic nutrients".

*Line 45:"Meanwhile" should be "Besides"*

We have modified "Meanwhile"
to
"Besides".

*Line 51-53: "Furthermore, nutrient and algae dynamics are strongly linked with solid dynamics through phosphate (PO4) sorption/desorption interactions with solid particles (McGechan and Lewis, 2002) and algae growth reduction due to light shading by suspended solids (SS) (Dio Toro, 1978) ": There are more processes that link nutrient and algae dynamics with solid dynamics, and what you list here are only two of them.*

We have added "for example" between "dynamics" and "through".

*Line 55-57: "Projecting global freshwater biogeochemistry changes requires process-based models that are robust under unprecedented conditions expected in the next century." This paper does not involve any future projections, but "unprecedented conditions expected in the next century" and similar statements have appeared many times in this paper. This paper does not show that this model can handle "unprecedented" situations. Therefore, I would advise to remove this kind of sentences.*

The term "unprecedented" was used only twice throughout the manuscript: in abstract and line 57.

Lines 57-70 provide background for this study. The paragraph indicates a generally accepted rationale (e.g., Randall et al., 2007) that future projections into novel environments will be more reliable, as they rest on process-level understanding that is likely to hold in these novel environments, as opposed to extrapolation beyond the bounds of empirical relationships based on historical data. This broad goal sets the context for FANSY development, but does not claim that it has been achieved. We thus feel that it is appropriate here, but have added extensive discussion of areas: this first version of LM3-FANSY should be improved in pursuit of this goal (Sect. 4.5).

The statement in abstract lines 24-27 also describes the purpose of FANSY development to be added as a prognostic component of Earth System Models which is commonly used for future projections.

The extensive discussion in Sect. 4.5 includes:

**There is also significant room for further model development and improvement. An improved representation of lakes (e.g., vertical layering) is necessary to better resolve algal processes and associated transformations between inorganic and organic phases. Modeling large lakes with the ocean component of GFDL's Earth System Model (Adcroft et al., 2019) is one of our priority developments, particularly given the importance of algae as a control on the relative proportions of inorganic vs. organic nutrients in freshwaters. An initial configuration for the U. S. Great Lakes is currently under development.**

**There is also a need to pursue advances to provide a more comprehensive and consistent approach of modeling the coupled N, C, and P cycles across the terrestrial and freshwater continuum of LM3-FANSY. Expansion of LM3 to include terrestrial P dynamics will be targeted to improve estimates of litter/soil P storage and fluxes to streams and rivers, generating mechanistically consistent estimates of N:P ratios of nutrient loads reaching coastal systems. Priority enhancements will also include integration of freshwater C and**

**alkalinity dynamics with the current solid, algae, and nutrient dynamics of FANSY to simulate the impacts of river inputs on coastal C budgets and acidification.**

**The current version of LM3-FANSY simulates denitrification emissions from freshwaters, but does not include processes that explicitly separate $N_2O$ and $N_2$ emissions from the freshwater denitrification. As freshwater $N_2O$ emissions have been recognized as an increasingly important greenhouse gas source (Yao et al., 2020), it will be important to differentiate $N_2$ and $N_2O$ emission processes in future work.**

**Although LM3-FANSY is capable of producing river solids and nutrients, in various forms and units, some disagreements between the modeled and measurement-based estimates remain. Many observational studies have noted the uncertainties associated with measurement methods, location, and frequency that likely contribute to these disagreements. Additional time varying constraints are also needed to build additional confidence in projected changes. Finally, all of these model improvement efforts will be greatly facilitated by implementing LM3-FANSY at higher resolution capturing the larger number of rivers and by extensive river measurements across the world, with a better assessment of uncertainties.**

The following references are newly added.

Adcroft, A., Anderson, W., Balaji, V., Blanton, C., Bushuk, M., Dufour, C. O., et al.: The GFDL global ocean and sea ice model OM4.0: Model description and simulation features, J. Adv. Model. Earth Syst., 11, 3167–3211, https://doi.org/10.1029/2019MS001726, 2019.

Yao, Y., Tian, H., Shi, H. et al.: Increased global nitrous oxide emissions from streams and rivers in the Anthropocene, Nat. Clim. Chang., 10, 138–142, https://doi.org/10.1038/s41558-019-0665-8, 2020.

*Line 59: "Prior applications of process-based freshwater biogeochemistry models, such as …, have generally been limited to small watersheds". This is not true. Many process-based models have been developed to simulate freshwater biogeochemistry in inland waters more than "small watersheds", such as DLEM (Tian et al., 2020; Yao et al., 2020; Bian et al., 2022), LOAC (Akbarzadeh et al., 2019), INCA(Whitehead et al., 1998a; Whitehead et al., 1998b; Wade et al., 2002), IMAGE-DGNM (Vilmin et al., 2020; Vilmin et al., 2022), for large river basins and global scale. A better understanding and summary of processed freshwater biogeochemistry models are needed.*

We agree that the statement was too general as written and, as a result, inaccurate. We have rewritten it to acknowledge that process-based models resolving N or P cycle alone have been

widely applied across scales. Applications of models that simulate intertwined algae, multi-nutrient and/or solid dynamics, such as RIVE and QUAL2K, however, have largely been limited to focused studies in smaller watersheds.

We have modified and added statements to clarify our points as follows:

**Process-based freshwater biogeochemistry models of N cycle alone (LM3-TAN, Lee et al., 2014; DLEM, Tian et al., 2020; IMAGE-DGNM, Vilmin et al., 2020; INCA, Wade et al., 2002) or P cycle alone (Bian et al., 2022; Vilmin et al., 2022) have been widely applied across scales. However, applications of models that capture coupled algae, multi-nutrient, and/or solid cycles, such as RIVE (Billen et al., 1994) and QUAL2K (Pelletier et al., 2006), have generally been limited to relatively small watersheds.**

The following references are newly added.

**Bian, Z., Pan, S., Wang, Z., Yao, Y., Xu, R., Shi, H., Kalin, L., Anderson, C., Justic, D., Lohrenz, S., and Tian, H.: A Century-Long Trajectory of Phosphorus Loading and Export From Mississippi River Basin to the Gulf of Mexico: Contributions of Multiple Environmental Changes, Global Biogeochem. Cycles, 36, e2022GB007347, 2022.**

**Tian, H., Xu, R., Pan, S., Yao, Y., Bian, Z., Cai, W.-J., Hopkinson, C.S., Justic, D., Lohrenz, S., Lu, C., Ren, W., and Yang, J.: Long-term trajectory of nitrogen loading and delivery from Mississippi River Basin to the Gulf of Mexico, Global Biogeochem. Cycles 34, e2019GB006475, 2020.**

**Vilmin, L., Bouwman, A. F. , Beusen, A. H. W., van Hoek, W. J., and Mogollón, J. M.: Past anthropogenic activities offset dissolved inorganic phosphorus retention in the Mississippi River basin, Biogeochemistry, 161, 157-169, 2022.**

**Vilmin, L., Mogollón, J. M., Beusen, A. H. W., van Hoek, W. J., Liu, X., Middelburg, J. J., and Bouwman, A. F.: Modeling process-based biogeochemical dynamics in surface fresh waters of large watersheds with the IMAGE-DGNM framework, J. Adv. Model. Earth Syst., 12, 1-19, 2020.**

**Wade, A. J., Durand, P., Beaujouan, V., Wessel, W. W., Raat, K. J., Whitehead, P. G., Butterfield, D., Rankinen, K., and Lepisto, A.: A nitrogen model for European catchments: INCA, new model structure and equations, Hydrol. Earth Syst. Sci., 6, 559-582, 2002.**

*Line 69-70: Indeed IMAGE-GNM does not differentiate the different forms, but therefore the IMAGE-DGNM is developed (Vilmin et al., 2020; Vilmin et al., 2022).*

In lines 62-70, we are discussing global applications. In lines 58-59 where regional applications are discussed, we have added the IMAGE-DGNM references.

*Line 73-75: "IMAGE-GNM takes a mass balance approach to calculate soil nutrient budgets, which at times rests on simple scaling without potential dynamical feedbacks (e.g., an estimation of litter from floodplains to rivers as 50% of total net primary production (Beusen et al., 2015))." Firstly, floodplain vegetation is not calculated as an item in the soil budget in IMAGE-GNM but as a direct source to surface water. Secondly, the use of 50% of net primary production (NPP) for wetlands and floodplains can be uncertain, but the area of floodplain and wetlands due to hydrology, as well as the simulated NPP are both spatially explicit and time-dependent. It is therefore not correct to say that calculation of nutrients from floodplain vegetation to rivers in IMAGE-GNM is just simplified without dynamic feedback. Not to mention that for all the other soil and water budget, IMAGE-GNM simulates them in a mass-balanced and dynamic way.*

We apologize for the confusion. What we meant has been clarified as follow:

**IMAGE-GNM does not explicitly simulate terrestrial dynamics, such as vegetation growth, leaf fall, natural and fire-induced mortality, and soil microbial processes, but takes a mass balance approach to calculate dynamic soil nutrient budgets (Beusen et al., 2015).**

*Line 86-89: A sentence too long to read.*

We have modified "dynamics linking"
to
"dynamics. LM3-FANSY is aimed at linking ".

*Line 90: "performance assessment against measurement-based global and regional estimates across world major rivers". Ambiguous. The global result cannot be compared with "major river (s)" to assess model performance.*

We have clarified it as "performance assessment against measurement-based estimates of nutrient and suspended solid loads across major world rivers".

*Line 119-123: These sentences are unclear: First sentences states that sewage and aquaculture are from prescribed sources. Last line claims all P input is not from LM3? Please make more clear.*

We have clarified the sentences as follow:

**The freshwater component FANSY receives N, P, and solids in multiple forms either from LM3 or from prescribed inputs. It then simulates biogeochemical transformations and transport of each form of the nutrients and solids within streams, rivers, and lakes. The N and P inputs to rivers are specified in Sect. 3.1.**

For clarification, we have extended the input description in Lines 402-403 as follows:

**Solid and nutrient inputs from terrestrial systems and from the atmosphere to rivers are either simulated by LM3-FANSY or provided by Beusen et al. (2015) (Table 3). For N, all inputs were simulated by LM3-FANSY except aquaculture, wastewater, and atmospheric deposition, which were provided by Beusen et al. (2015). For P, which is not currently included in LM3, all inputs were provided by Beusen et al. (2015).**

**Beusen et al. (2015) provided five-year interval data for the period 1900-2000 at 0.5 degree resolution. The data were regridded to our 1 degree resolution by summing up the values given in kg yr$^{-1}$ and linearly interpolated across the five-year intervals. Beusen et al. (2015)'s wastewater N and P inputs were from Morée et al. (2013)'s urban waste N and P discharge estimates to surface waters. Beusen et al. (2015) calculated aquaculture N and P inputs using Bouwman et al. (2013)'s finfish and Bouwman et al. (2011)'s shellfish data. For atmospheric N deposition inputs, Beusen et al. (2015)'s input for the year 2000 was from Dentener et al. (2006)'s ensemble of reactive-transport models and those for the years before 2000 were made by scaling the deposition with Bouwman et al. (2013)'s ammonia emissions. Beusen et al. (2015)'s surface runoff P inputs include those leached from soil P budgets (i.e., the sum of fertilizer and animal manure minus crop and grass withdrawal) and those driven by soil erosion estimates based on Cerdan et al. (2010). Beusen et al. (2015)'s P inputs of litter from floodplains were estimated as 50% of total NPP with a C:P ratio of 1200. Beusen et al. (2015)'s weathering P inputs were computed based on Hartmann et al. (2014)'s chemical weathering P release estimates.**

The following references are newly added.

Bouwman, A. F., Beusen, A. H. W., Overbeek, C. C., Bureau, D. P., Pawlowski, M., and Glibert, P. M.: Hindcasts and future projections of global inland and coastal nitrogen and phosphorus loads due to finfish aquaculture, Rev. Fish Sci., 21, 112–156, doi:10.1080/10641262.2013.790340, 2013.

Bouwman, A. F., Klein Goldewijk, K., Van der Hoek, K. W., Beusen, A. H. W., Van Vuuren, D. P., et al.: Exploring global changes in nitrogen and phosphorus cycles in agriculture induced by livestock production over the 1900–2050 period, P. Natl. Acad. Sci. USA, 110, 20882–20887, doi:10.1073/pnas.1012878108, 2013.

Bouwman, A. F., Pawłowski, M., Liu, C., Beusen, A. H. W., Shumway, S. E., Glibert, P. M., and Overbeek, C. C.: Global hindcasts and future projections of coastal nitrogen and phosphorus loads due to shellfish and seaweed aquaculture, Rev. Fish Sci., 19, 331–357, doi:10.1080/10641262.2011.603849, 2011.

Cerdan, O., Govers, G., Le Bissonnais, Y., Van Oost, K., Poesen, J., et al.: Rates and spatial variations of soil erosion in Europe: A study based on erosion plot data, Geomorphology, 122, 167–177, doi:10.1016/j.geomorph.2010.06.011, 2010.

Dentener, F., Stevenson, D., Ellingsen, K., Noije, T. v., Schultz, M., et al.: The global atmospheric environment for the next generation, Environ. Sci. Technol., 40, 3586–3594, 2006.

Hartmann, J., Moosdorf, N., Lauerwald, R., Hinderer, M., and West, A. J.: Global chemical weathering and associated P-release – The role of lithology, temperature and soil properties, Chem. Geol., 363, 145–163, doi:10.1016/j.chemgeo.2013.10.025, 2014.

Morée, A. L., Beusen, A. H. W., Bouwman, A. F., and Willems, W. J.: Exploring global nitrogen and phosphorus flows in urban wastes during the twentieth century, Global Biogeochem. Cy., 27, 1–11, doi:10.1002/gbc.20072, 2013.

*Line 132-133: "Algae chlorophyll a (CHL), algae C (Ca), algae P (Pa), and algae dry matter (Da) are diagnosed from algal N (Na) and, in the case of CHL, nutrient and light conditions", not clear. It's better to separate it into two sentences and describe the two situations.*

We have modified the sentence as follow:

**Algae C ($C_{al}$) and P ($P_{al}$) are diagnosed from algal N ($N_{al}$) assuming the Redfield C:N:P ratio (Chapra, 1997; Redfield et al., 1963). Chlorophyll a (CHL) is derived using the photoacclimation model of Geider et al. (1997) to predict a CHL-to-C ratio ($r_{CHLC}$) and calculate CHL from $r_{CHLC}$ and $C_{al}$.**

*Line 135-138: "The 5 prognostic P variables include the same organic forms as for N (dissolved organic P (DOP), particulate organic P (POP), and sedimentary organic P (SedP)), but includes dissolved and particulate inorganic forms (PO4 and PIP, Sec. 2.2.4) rather than the oxidation state distinction as done for N. " This sentence is too long and not so readable. Consider improving it. Perhaps add "variables" at the end of this sentence, after "N".*

We have modified the sentence as follow:

**The 5 prognostic P variables include the same organic forms as for N variables (dissolved organic P (DOP), particulate organic P (POP), and sedimentary organic P (SedP), Sect. 2.2.4). The other two variables are dissolved and particulate inorganic forms ($PO_4^{3-}$ and PIP), rather than distinguished by the oxidation state as done for N variables.**

*Line 140: Ambiguous expression of "variable processes": do you mean processes which are variable, or variable's processes, or variables and processes?*

We clarified it as "a detailed description of each variable and associated processes".

*Line 4 in Table 1: From Line 128 in the text, I think you mean Sed is the abbreviation for "bottom sediment". But in Table 1, it is not clear and differs from the text. Modification is needed to ensure the term use is consistent and clear.*

We have replaced the "sediment solids" with "benthic sediment inorganic solids" in Table 1 and throughout the manuscript.

*Table 1 and text: This is a biogeochemistry model which includes quite some biogeochemical processes and different chemical forms. Therefore, it is not appropriate to use the common*

*symbols of Na, Ca and Pa, as new abbreviations of the terms in this study. Please consider revising these symbols to ensure clarity and avoid confusion.*

We have replaced the symbols to $N_{al}$, $C_{al}$, and $P_{al}$.

*Figure 1: (1) The pathway of Algae dry matter to SS is quite strange. Do you mean that this part of organic matter will go to the pool of SS in water or finally Sed in sediment? Then this results in permanent additive pools and the pathway, which will only increase. This means that you may have partitioned the algae organic matter directly into one part which will be degraded, and the other refractory part which will not participate in biogeochemical reactions. The refractory part after partitioning will always increase the SS (water) and Sed pools (bottom sediment) pools, but will not add to the total PON and POP pools, which is quite strange. Or another possible assumption for this is that the dead algae are counted as SS which only participates in the physical processes, while in fact the dead algae already become NH4+ (see your pathway of another algae mortality), which should not add to the SS pool.*

*Besides, you have SedN and SedP specially for N- and P-contained sediment. The interactions seem somewhat unclear and not self-explanatory.*

We were able to remove "the pathway of algae dry matter to SS" after adopting the advice of reviewer 2 to link SS and PON fluxes from land to rivers. Please see the modified Fig. 1.

We now simulate PON fluxes from terrestrial litter and soils to rivers based on the simulated soil erosion fluxes and litter/soil N concentrations, adapting an approach of previous studies that simulate POC fluxes based on soil erosion fluxes and soil C concentrations (Tian et al., 2015; Zhang et al., 2022). We also now use the simulated PON fluxes to calculate inorganic soil fluxes from the total soil erosion fluxes, so that the eroded soil and PON inputs to rivers are self consistent. This now allows us to use inorganic suspended solids (ISS) in freshwaters as our prognostic variable, and derive total suspended solids (SS, i.e., the sum of ISS and POM) from the combination of freshwater ISS and PON. Accordingly, Sed is now benthic sediment "inorganic" solids, differently from "organic" nutrients of SedN and SedP.

For details, please see our response to the reviewer 2.

The following references are newly added.

**Tian, H., Yang, Q., Najjar, R. G., Ren, W., Friedrichs, M. A. M., Hopkinson, C. S., and Pan, S.: Anthropogenic and climatic influences on carbon fluxes from eastern North**

America to the Atlantic Ocean: A process-based modeling study, J. Geophys. Res. Biogeosci., 120, 757– 772, 2015.

Zhang, H., Lauerwald, R., Regnier, P., Ciais, P., Van Oost, K., Naipal, V., Guenet, B., and Yuan, W.: Estimating the lateral transfer of organic carbon through the European river network using a land surface model, Earth Syst. Dynam., 13, 1119–1144, https://doi.org/10.5194/esd-13-1119-2022, 2022.

*(2) algae growth and mortality are not the common definitions themselves (see Line 216-217), which calls for earlier explanations or notes to inform the readers from the very beginning. Otherwise this is too confusing.*

We have clarified the statement as follows:

**Algae dynamics are governed by the balance of net growth (i.e., gross growth – respiration) and generalized mortality (i.e., non-predatory mortality + grazing + settling + excretion). The net growth is the difference between gross photosynthesis and respiration. The mortality may include contributions from grazing, excretion, viruses, cell death, and excretion, though these diverse contributions are ultimately parameterized as a simple density-dependent loss term (Dunne et al., 2005). For a batch river and lake system, a mass balance for algae is written as:**

$$\frac{dSB_{N_{al}}}{dt} = \mu(I_{av}, T, N, P) \cdot N_{al} - m(T) \,, \tag{12}$$

**where $N_{al}$ is algae N (kgN) and $\mu(I_{av}, T, N, P)$ is algae net growth rate (s$^{-1}$) as a function of euphotic zone averaged irradiance $I_{av}$, T, N, and P.**

*Line 156 – 157: Equation 1 and 2. The parameter Ri is used in both equations with the same sign. Is that correct? Is Ri not part of Fin or Fout? Can you explain this better?*

As described in lines 152-154, Ri is settling/resuspension dynamics and/or biogeochemical reactions for variable i, which are detailed throughout the sections 2.2.1-2.2.4. For example, if i is PON, Ri is Eq. (29) in the original manuscript (now the Eq numbers have been changed). If i is SedN, Ri is Eq. (30).

Inputs and outputs via river flows (Fin and Fout) are only applicable for water column species in Eq. (1). Because Eq. (2) is only for benthic sediment species, it does not have Fin and Fout terms.

We have clarified this as follows:

**In each river reach or lake, for each prognostic variable i, Settling/resuspension dynamics and/or Biogeochemical reactions (SB$_i$) are calculated according to the process-based formulations described in the following subsections 2.2.1-2.2.4. For example, if i is PON, SB$_i$ is Eq. (28). If i is SedN, SB$_i$ is Eq. (29). A general mass balance for a variable i in a river reach or lake at each computation time step (30 minutes in this study) is written as:**

$$\frac{dX_i}{dt} = F_i^{in} - F_i^{out} + I_i + SB_i ,$$ **if i is a river/lake water column variable,**

**i.e., i = all variables except Sed, SedN, and SedP    (1)**

$$\frac{dX_i}{dt} = SB_i ,$$ **if i is a river/lake benthic sediment variable,**

**i.e., i = Sed, SedN, or SedP                    (2)**

**where i is a prognostic variable listed in Table 1, X$_i$ is the amount of species i (kg), F$_i^{in}$ and F$_i^{out}$ are inflow and outflow of the variable i (kg s$^{-1}$), I$_i$ is inputs of the variable i from terrestrial systems and the atmosphere (kg s$^{-1}$), and SB$_i$ is settling/resuspension dynamics and/or biogeochemical reactions of the variable i (kg s$^{-1}$).**

We have also consistently modified the notation of the equations in the sections 2.2.1-2.2.4 (Please notice that we have changed $N_{al}$ to $SB_{N_{al}}$ in Eq. (12) in the previous response.

*Line 167: Which rainfall is meant here? Annual, monthly, weekly, daily? Clarify.*

This is the simulated 3 hourly rainfall. The details can be found in the description of model forcing (lines 394-396).

*Line 170: C1 is calibrated. How and on what? Please elaborate on this.*

Yes. We have modified the statement in lines 170-172 as follows:

**C$_1$ is a single global value scale parameter (Pelletier, 2012) and coarsely calibrated to reduce prediction errors of SS loads across world major rivers (see Sect. 2.2.5 for details in model calibration).**

We have also introduced a new subsection "2.2.5 Model calibration" to describe details about the LM3-FANSY calibration.

**2.2.5 Model calibration**

Because many of the reported parameters required to simulate the coupled algae, nutrient, and solid dynamics within LM3-FANSY vary widely, it is difficult to assign a single global value for each parameter. Informed by the parameter sensitivity analysis herein (Sect. 4.4), our approach was to coarsely calibrate a limited set of uncertain yet highly influential parameters within their broad observed ranges to reduce prediction errors of SS, N, and P loads in different forms across world major rivers.

First, the free parameter of terrestrial soil erosion, $C_1$, was adapted from Pelletier (2012) to account for the input and model resolution difference. It is a scale parameter, as demonstrated in Eq. (3), that affects the overall amount of SS and was calibrated to reduce prediction errors of SS loads.

Second, the generalized algal mortality constant, $k_m$, was adapted from Dunne et al. (2005) to reduce the complexity of various algal mortality pathways. This parameter has been tuned to produce reasonable chlorophyll a concentrations in globally distributed lakes, acknowledging the limitation of the present lake biogeochemistry in LM3-FANSY (see Sect. 4.5 further discussion). We adapted an approach of Chapra (2008) to partition nutrient fluxes from the algae mortality to different pools of nutrients in different forms (i.e., particulate organic, dissolved organic, and inorganic), based on fixed fractions. Uncertainties in these fixed factions due to the lack of theoretical and empirical evidence have been investigated in the sensitivity analysis (Sect. 4.4).

Third, we find that the rate coefficients for hydrolysis, nitrification, and denitrification are highly influential parameters for determining river nutrient loads in different forms, relative to those for decay processes that breakdown complex organic compounds into simpler organic and inorganic compounds. These highly influential parameters have been calibrated to reduce prediction errors of nutrient loads in different forms. We adapted an approach of Bowie et al (1985) to partition fluxes from complex organic nutrient decomposition to more simpler organic vs. inorganic nutrient pools based on a uniform fraction. Uncertainties in the faction have been investigated in the sensitivity analysis (Sect. 4.4).

*Line 1 in Table 2: "Many reactions are reported at 20°C" is not appropriate for the use of the ideal temperature (Tref) of all reactions. Detailed citations of mechanism studies are needed. Also, the dynamic feedback, which has been advertised so many times, cannot be well examined with the same Tref for every biogeochemical process related to every element form*

*driven by every organism group. Many mechanism studies have shown the obvious difference in Tref of nitrification, denitrification, mineralization, primary production, etc. Using the same Tref constraint for all biogeochemical processes related to all element forms driven by organism groups will yield a wrong estimation of the process rates, element and algae processing and the consequent nutrient concentrations.*

The choice of a reference temperature is generally a decision made by the investigator fitting the relationships that we draw from in constructing the model. These relationships can generally be recast to mathematically identical forms using different reference temperatures, but 20 °C, is a common and thus intuitive reference, e.g.: "many reactions are reported at 20 °C" as stated in a book "Surface Water-Quality Modeling" by Chapra (1997).

Reported temperature correction factor, $\theta$, values for different reactions vary widely (Bowie et al., 1985 and references there). The value of 1.066, which we used for all the reactions, is a typical value for phytoplankton growth (Chapra, 1997; Eppley, 1972), but is also within the reported ranges for the other reactions. The $\theta$ values for organic nutrient decompositions that break down complex organic nutrients into simpler ones (e.g., PON to DON, POP and SedP to DOP) range from 1.02 to 1.08. The $\theta$ values for organic nutrient decompositions to inorganic ones (e.g, DON, PON, and SedN to $NH_4^+$; DOP, POP, and SedP to $PO_4^{3-}$) range from 1.02 to 1.14. The $\theta$ values range from 1.02 to 1.08 for nitrification and range from 1.02 to 1.09 for denitrification.

Given the wide reported ranges of the $\theta$ values, it is difficult to assign a specific value for each reaction, reflecting the limitation of any given global models including LM3-FANSY. We now have assigned the $\theta$ value of 1.066 for algal dynamics and of 1.08 (a typical value for sediment oxygen demand) for the other reactions.

We have also conducted 4 sensitivity simulations to show model responses to different $\theta$ values: (1) 1.066 for algal dynamics and 1.024 (a typical value for oxygen reaeration) for the other reactions, (2) 1.066 for algal dynamics and 1.047 (a typical value for biological oxygen demand) for the other reactions, (3) 1.066 for algal dynamics and 1.066 for the other reactions, and (4) 1.066 for algal dynamics, 1.08 for sediment decompositions, and 1.047 for the other reactions.

We find insensitiveness of the model to these variations, which can be found in the sensitivity analysis (Sect. 4.4, Table 7).

*Line 6 in Table 2 and Line 192: "R" symbol is used before, as term reaction. It is confusing to use the same letter for two different terms. Change one of them.*

For settling/resuspension dynamics and/or biogeochemical reactions, we have modified Ri to SBi.

***Equation 8: Please refer to the original reference (Ludwig and Probst). This equation is very old and only based on 19 observations. Are there not more observations to improve this?***

We were able to remove this equation after adopting the advice of reviewer 2 to link SS and PON fluxes from land to rivers. For details, please see our earlier response to the reviewer's comment on Figure 1: (1).

***Line 216-217: the definitions of algae growth and mortality differ from other studies! This should be mentioned to inform readers, or at least tell them these processes are "generalized" or "net". Otherwise, it is too confusing.***

We have attempted to clarify this in our earlier response to the reviewer's comment on Figure 1: (2):

We hope that addresses the reviewer's concern. If we have missed the mark, please let us know which other studies have adopted different definitions and we will clarify further.

***Line 273: it is hard to understand how equation #23 calculates the combined the net flux of a generalized mortality (non-predatory mortality + grazing + settling + excretion) as mentioned in Line 216-217. Contradictorily, according to Line 271-277, this equation does not reflect so-called "generalized mortality", but grazing (death because of predation).***

As discussed in Dunne et al. (2005), phytoplankton mortality processes are generally divided according to the degree of their density dependence. Grazing, for example, is often modeled as density-dependent loss with a quadratic relationship. Such a relationship implicitly assumes that the biomass of grazers scales in proportion to the biomass of their prey without lag. Similar assumptions have been used for losses to marine viruses (e.g., Stock et al., 2014) and aggregation (e.g., Doney et al., 1996). In reality, however, phytoplankton and their grazers are not perfectly coupled. Zooplankton responses lag phytoplankton fluctuations. There are also loss mechanisms such as excretion, basal respiration, and cell death that may not be density dependent. Dunne et

al. (2005) found that a density-depedent but weaker 4/3 scaling rather than a quadratic scaling did a better job of simulating dynamic shifts in phytoplankton biomass (e.g., blooms) while also setting an upper biomass bound in a given nutrient and light environment.

We have attempted to clarify these points while leaning on previously published works for the details:

"where km is temperature-dependent algae mortality rate (kgN-1/3 s-1) that reflecting the combined impacts of zooplankton grazing and other phytoplankton loss terms (e.g., viral-induced losses, cell death). Exponents between 4/3 and 2 have been commonly applied in this relationship, with higher values corresponding to more tightly coupled top-down control (Dunne et al., 2005). We have adopted a value of 4/3 to enable high biomass in nutrient rich environments."

Doney, S.C., Glover, D.M., and Najjar, R.J.: A new coupled, one-dimensional biological–physical model for the upper ocean: applications to the JGOFS Bermuda Atlantic Time-series Study (BATS) site. Deep-Sea Research Part II 43, 591–624, 1996.

Stock, C. A., Dunne, J. P., and John, J. G.: Global-scale carbon and energy flows through the marine planktonic food web: An analysis with a coupled physical–biological model, Progress in Oceanography, 120, 1-28, 2014.

***Line 291: "assuming C weight content …. 1500 kg m3". Does this mean that you use a global number for bulk density and for the C content of the soil? I hope not…***

Sorry for the misunderstanding. Dynamically simulated soil N and, by extension, C contents vary spatially and temporally. The simulated effective soil depth, as defined by Gerber et al., 2010, was used to calculate spatially and temporally varying N concentrations per unit volume from N contents per unit area. We have now clarified this in the text.

***Line 297-296: I wonder why PON cannot be calculated from TKN-NH4-DON for this calibration purpose.***

As shown in Table SI3-5, there is only one river data available for the all three species TKN, $NH_4^+$, and DON.

*Line 299-302: I wonder whether or not fPON can or should be used for compensating hydraulic controls like dams and reservoirs. I'm afraid this is not very appropriate for a mechanistic model. This factor is constant (Table 2) and from the calibration of measurements. But it is unclear how this calibration was performed. Line 295 claims that fPON slows down the N movement, but that that not make any sense…. Also the "limited measurement-based PON estimates" makes it a bit a mystery.*

We were able to remove Eq. (28) including fPON after adopting the advice of reviewer 2 to mechanistically link SS and PON fluxes from land to rivers. For details, please see our earlier response to the reviewer's comment on Figure 1: (1).

*Line 328: see comment for Line 273. I don't find any rationality for using a part of the mortality calculation of predation-caused algae death as DON excretion to NH4+. Also, I don't know how fDON used here is calibrated, which is very strange to be done because generally it is unknown how much DON excretion will account for in algal mortality due to predation.*

We apologize for the misunderstanding: there is no path of "algae death as DON excretion to $NH_4^+$" in FANSY. We have clarified the model calibration by adding a new section (Sect. 2.2.5 Model calibration). Please see for the full model calibration description in our earlier response to the reviewer's comment on Line 170, which includes the description of fDON calibration:

**We adapted an approach of Chapra (2008) to partition nutrient fluxes from the algae mortality to different pools of nutrients in different forms (i.e., particulate organic, dissolved organic, and inorganic), based on fixed fractions. Uncertainties in these fixed factions due to the lack of theoretical and empirical evidence have been investigated in the sensitivity analysis (Sect. 4.4).**

*Equations 35 – 38: What is CPN?*

We corrected the typo cPN to rPN. It is the algae P to N ratio used to diagnose algae P from algae N.

***Line 367: Calculation for 1900 – 2010. But Beusen et al. (2015) has provided only data up to 2000. How did you calculate the last ten years?***

As shown in our earlier response to the reviewer's major comment 5, we now have limited our analyses up to 2000.

***Line 368: Confusing: "reported point and nonpoint N and P inputs to rivers" Can you better describe what you used?***

The input description has been clarified in our earlier response to the reviewer's comment on Line 119-123.

***Line 370: 1990 calibrated (constant) from partitioning factors for instream processes may be not suitable for the period 1982-2010 considering the possible temporal changes. Calibration process is not clear here.***

We have removed the statements. As shown in our earlier response to the reviewer's major comment 5, we now have modified the years for the analyses to focus on 1990-2000. Please see our earlier response to the reviewer's comment on Line 170 for the full description of model calibration (Sect. 2.2.5 Model calibration).

***Line 373-381: This atmospheric N deposition is calculated here, and can be different from Beusen et al. 2015. However, In Table 3 it is clearly written that atmospheric N deposition is from Beusen et al. 2015. So what are you using? Please clarify.***

As described in lines 380-381, the atmospheric N deposition is only applied to soils, not to rivers. The description of inputs to rivers has been clarified in our earlier response to the reviewer's comment on Line 119-123.

***Table 3: So LM3-FANSY simulates soil N budget and soil budget and their delivery to rivers and lakes, but no P budget. This means that P is not coupled to the LM3. How consistent are***

*the assumptions of P of Beusen et al. (2015) with the assumptions of LM3? Do they both have agricultural land on the same spots? This is a major concern of me, because one of the results is the N:P ratio. If the assumptions do not match, then the N:P ratio does not mean anything. Perhaps the N:P ratio is a bridge to far for this model at the moment…..*

We agree and have addressed this concern in our earlier response to the reviewer's major comment 3.

*Table 3: The numbers in Table 3 are not the same as in in Vilmin et al. (2018). The fractions for wastewater are dependent on the treatment level. Here it is assumed that whole the world has primary treatment, which is not true. The fractions of agricultural surface runoff and aquaculture are different than Vilmin et al. (2018). How is this determined?*

We agree that the description of using the fractions from Vilmin et al. (2018) was not clear in the previous manuscript. Since only aggregated inputs were available from Buesen et al. (2015), we had to adopt an admittedly simplified assumption for the wastewater additions to each of the three species. We have extended the description, and also conducted sensitivity tests to show that uncertainties associated with the fractions had only a small impact on our results (Table SI10). The clarified descriptions have been added in Table 3 and line 404 as follows:

**Vilmin et al (2018) suggest fractions to divide TN and TP inputs to rivers from different sources into three N and P species respectively. For sewage, three groups of fractions are given for three types of sewage (i.e., untreated, primary treated, and secondary/tertiary treated). The sewage inputs from Beusen et al. (2015) that we used are aggregated for all three types of sewage. To divide Beusen et al. (2015)'s sewage TN and TP inputs into three N and P species respectively, we have taken a middle ground by using the fractions for primary treated sewage. Although we acknowledge that use of the fractions is unrealistic, we find that our results are relatively insensitive to a wide range of alternatives, as assessed by 4 parameter perturbation experiments (see Table 3 and the sensitivity analysis in Sect. 4.4). The first 2 uncertainty simulations used 1) the fractions for untreated sewage and 2) the fractions for secondary/tertiary treated sewage. The next 2 uncertainty simulations used assumed fractions based on an estimate that globally, over 80% of wastewater is discharged without "adequate treatment" (Environment and Natural Resources Department, 2022). Specifically, the fractions are driven by assuming that 1) 80%, 10%, and 10% of Beusen et al. (2015)'s sewage are untreated, primary treated, and secondary/tertiary treated respectively and 2) 40%, 40%, and 20% of Beusen et al. (2015)'s sewage are untreated, primary treated, and secondary/tertiary treated respectively.  In all**

**cases, the simulated river loads of each species changed by < 9%, and the simulated total loads did not change.**

**For TP fluxes from agricultural lands to rivers, two groups of fractions are given for two sources (i.e., surficial runoff and soil loss). Agricultural surface runoff TP inputs from Beusen et al. (2015) are aggregated for both sources. To divide Beusen et al. (2015)'s agricultural surface runoff TP inputs into three P species, we have used nearly equal fractions. Although we acknowledge that use of the fractions produces uncertainties, we used perturbation experiments to show that our results are relatively insensitive to a wide range of the fractions (see Table 3 and the sensitivity analysis in Sect. 4.4). The 6 uncertainty simulations used 1) the fractions for surficial runoff for all agricultural fluxes, 2) the fractions for soil loss for all agricultural fluxes, fractions driven by assuming that 3) 40% and 60% of Beusen et al. (2015)'s agricultural surface runoff TP inputs are from surficial runoff and soil loss respectively, 4) 60% and 40% of those are from surficial runoff and soil loss respectively, 5) 20% and 80% of those are from surficial runoff and soil loss respectively, and 6) 80% and 20% of those are from surficial runoff and soil loss respectively.  In all cases, the simulated riverine loads of each species changed by < 13%, and the simulated total load changed by < 1%.**

**For aquaculture, two groups of fractions are given for particulate and dissolved sources. The fractions in Table 3 are driven by assuming that about 12% and 31% of aquaculture TN and TP inputs from Beusen et al. (2015) are particulate, based on Figure 3 of Bouwman et al (2013). We did not conduct sensitivity simulations for aquaculture, because aquaculture nutrient inputs are very small compared to the other sources (< 1% of total inputs for both N and P), and thus uncertainties associated with the fractions should be negligible for the global application herein.**

The following reference is newly added.

Environment and Natural Resources Department, Wastewater as a resource, European Investment Bank, https://www.eib.org/en/publications/wastewater-as-a-resource, 2022

*Line 434-445: The use of observation data for this model is very strange. There are a lot of published global measurement data available for validation for the period 1982-2010, but this paper intentionally chooses the measurement data during the 1970s-1990s which Global NEWS used. This introduced high temporal uncertainty may be larger than the model uncertainty itself. It is not clear how the comparison is made. The timestep is 30 minutes. So how is the value that is used for calibration constructed? Discharge weighed? Mouth of the river or the location of the observation? Only the year 1990? Or average over yearly*

*concentrations of 1970 – 1990 or 1982 – 2010? This is very unclear. The time period difference is very strange and is also a concern. A lot has happened with the N and P loads/concentrations in the period 1982 – 2010.*

GEMS-GLORI is a widely cited database (Meybeck and Ragu, 2012) that has been used by many global watershed modeling studies (e.g., He et al., 2011; Beusen et al., 2015), in addition to Global NEWS. We have also used the database, because it provides perhaps the most comprehensive river chemistry. The GEMS-GLORI database was compiled in a consistent manner and provides a detailed data description.

As noted in our earlier response, we further evaluated our results against Global NEWS considering it as a "gold standard" and widely cited and pioneering community benchmark.

All the dynamics in FANSY described in Sect. 2.2 are updated every 30 minutes. This means that the model outputs (including water, algae, solid, and nutrients throughout the river network) are given at the 30 minute time step. The annual results for yields, loads, and concentrations are the averages of 30 minute results with applicable unit conversions. This high temporal resolution does not require discharge-weighted calculations.

We have clarified the years used for the analyses in our earlier response to the reviewer's major comment 5.

Also, we have clarified the description of the comparisons of measurement-based and modeled estimates in our earlier response to the reviewer's major comment 1.

*Line 444-445: "Cross-watershed …. and trends." Here the purpose of the model is defined. I do not agree with this. If this is true, then you don't need to spinup 11000 years, and do all the dynamic calculations. Just use the Global NEWS model. This model is far to complicated for this purpose and this purpose does not fit in the LM3 framework.*

We have extensively clarified the purpose of this model and this manuscript at the beginning of this response.

A land model spin-up is to create steady state conditions of terrestrial systems before forcing the model with contemporary conditions. It is a necessary procedure even for evaluating a single time period results of mostly unperturbed systems (e.g., Zhang et al., 2020).

*Line 450-452: It is unclear whether it is because of the calibrated parameter of instream solid dynamics that improves the model performance, because the results without calibrated parameter are not shown.*

As demonstrated in Eq. (3) and lines 167-172, $C_1$ is a single global value scale parameter to account for the difference in input data and model resolution. Eq. (3) demonstrates that the model predictive capacity for the spatial variation has little to do with this scale parameter, but is more likely explained by spatially varying inputs (e.g., slope, rainfall).

*Table SI1 – SI9: Nice to have all the observations together, but I miss the results from FANSY in these tables. Then the reader can also see, which rivers are doing fine or not.*

The LM3-FANSY results corresponding to the observational estimates in Table SI1-S9, along with all the other LM3-FANSY results, are reported in the supplementary excel file. Thus, the readers can see the LM3-FANSY performance relative to the observational estimates and perform additional analysis if they desire.

*Table 4: Again, "R" is used as a new abbreviation for another term in the same paper.*

The symbol "R" in Table 4 has been removed.

*Table 4 and 5: Not clear how this 15% is applied. Is the change applied only for 1990 and the calculation is only done for 1990? Constant over the year or not? Or total N or P and then some forms get more change than others? Please elaborate on this in the methods section.*

We have extended and clarified the description of sensitivity simulations in Sect. 3.2 as follows:

**3.2 Sensitivity simulations**

**Model sensitivities to the inputs, components, and parameters are examined by analyzing the responses of river solid and nutrient loads to their changes for the period 1948-2000. Model sensitivities to the nutrient inputs to rivers are examined by increasing each input source by 15% and removing it. When the inputs are from LM3, the 15% increase was**

**applied to each input variable (i.e., $NO_3^-$, $NH_4^+$, DON, and PON). When the inputs are externally prescribed, the 15% increase was applied to each input source listed in Table 3. In both cases, the increases were applied uniformly over the grid and time.**

**One of the distinct features of LM3-FANSY is the capability of modeling interactions of algae and nutrient dynamics with solid dynamics. Light shading by SS and algae themselves modulates the strength of algal productivity and, in turn, river solids and nutrients. In LM3-FANSY, the light extinction coefficient is dynamically simulated as a function of ISS, POM, and CHL (Eq. 23), instead of using a prescribed parameter. To evaluate how critical the dynamic light extinction component is for model predictive capacity, the component was replaced with a prescribed parameter value ($k_e = 0.15$ or $0.45$) and the river load responses to more active algal populations are examined.**

**Model sensitivities to the parameters are examined by decreasing each of all calibrated parameters listed in Table 2 by half and increasing it by twice. For the parameters which have much smaller or much larger impacts compared to the other parameters, additional sensitivity tests have been performed to show responses of river loads to the parameter changes in their broad observed ranges.**

*Tables 4-5: hard to interpret. To me, this sensitivity does not make any sense. First of all because the output parameter is the Pearson correlation coefficient. I can not understand this, how this helps me to understand the model mechanics. Better to do this on output parameters of the model. Then also an explanation could be added on why this difference is big or not. Secondly, I think that a one factor analysis is very limited for this kind of models.*

As shown in our response to the reviewer's opening comments, the sensitivity analyses are based on both 1) the Pearson correlation coefficient (Table 4 in the original manuscript), which tests the impact of a parameter, input, or component change on the spatial agreement between the simulated and observed relative river loads, and 2) the amount of river solid and nutrient loads (Table 5 in the original manuscript), which tests the effect of a change on the magnitude of the river loads. In the sensitivity analysis, we show the load differences between the baseline and sensitivity simulations and the model responses to different inputs/components (e.g. lines 640-643, 651-658).

In the modified manuscript, as done in a previous watershed modeling study (Harrison et al., 2010), we have further extended analysis by adding the Nash–Sutcliffe model efficiency coefficient which reflects "Model Efficiency". This addition is to evaluate the effects of each input source or model component removal on model predictive capacity. According to Harrison et al. (2010),

"Components with little impact on the model had little impact on Nash-Sutcliffe efficiency, whereas removal of critical model components had a large impact on Nash-Sutcliffe efficiency."

As shown in our response to the reviewer's earlier comment, we have substantially expanded sensitivity/uncertainty analyses by changing all calibrated parameters and by changing the fractions of dividing the externally prescribed TN and TP inputs into different N and P species, in addition to the previous model component and input manipulations. This has increased the total number of sensitivity/uncertainty simulations from 22 to 87, requiring 4611 years of total sensitivity/uncertainty simulations. Discussion of the insights gained by these expanded simulations has also been extended considerably.

We do not feel that Monte Carlo experiments changing all parameters at once are appropriate at this initial development stage because 1) it is challenging, if not impossible, to robustly attribute a response to a parameter or process when one is changing everything at once and 2) many of the simulations arising from such experiments prove implausible when compared with observations. We agree that such approaches are proper when constructing an ensemble of similarly plausible model permutations, but only after proper filtering for model fidelity and once the more basic single factor sensitivities are understood.

We feel that the significant expansion of single factor sensitivities/uncertainties and discussion made in response to the reviewer's comments and those of reviewer 2 have strengthened this model description paper considerably. The extended sensitivity analysis is as follows:

[revised manuscript text omitted]

**fluxes from algae mortality to different forms, which appears to have a modest effect on organic nutrient loads (Table 7).**

**Finally, additional uncertainty tests have shown the relatively insensitive responses of river loads to a broad range in 1) the fractions that divide externally prescribed TN and TP inputs into different N and P forms (Table SI10), 2) the fractions that partition fluxes from complex organic nutrient decomposition to simpler organic vs. inorganic nutrients, and 3) the temperature correction factor, θ, values that account for the temperature effect on freshwater biogeochemical reactions.**

*Table 6: Needs to mention the estimates from IMAGE-GNM are for 2000 instead of the 1990s as in other studies.*

We have added years for the estimates listed in Table 6 (which is Table 5 in the modified manuscript).

*Table 6: LM3-FANSY results needed to be added in this table for straightforward comparison.*

We have added the LM3-FANSY results in Table 6 (which is Table 5 in the modified manuscript).

*Line 485-486: This is not true. global river chemical measurement data are not limited, but they are simply not used in this paper.*

The statement in lines 485-486 means that measurement-based estimates of particulate nutrient compounds are limited. The relative lack of particulate nutrient measurements is also noted in the description of the GEMS-GLORI database (Meybeck and Ragu, 1997). To date, we have not seen any global watershed model evaluated against observational PON or POP across globally distributed rivers. We have clarified the statement as follows:

**Measurement-based estimates of particulate nutrient compounds to evaluate our model results at 1 degree resolution are limited.**

*Line 495-498: Explanations are needed for the estimate ranges like 36.4-41.3. Not clear whether this is because of temporal change or uncertainty or something different.*

The definition of the ranges (which are due to interannual variability) was previously defined in the caption of Table 5 wherein the estimates were reported. As shown in our earlier response to the reviewer's major comment 5, we now have modified the years for the analyses.

*Line 507: This model estimates a global export of 6.5-7.8 TgP. But if I correctly understand this article, you used the same riverine input as Beusen et al (2015) who estimated a global export of 4 TgP. So there is problem here. The global input of P is around 9 TgP (stated in Beusen et al. (2015). So the P loss in the river is around 30% whereas the P loss in Beusen et al is above 50%. This is a huge difference and should be explained here. Also the N loss should be checked!*

We note that the overall freshwater N and P budgets have been described in our earlier response to the reviewer's major comment 2 as follows:

**Globally, TN inputs to rivers in LM3-FANSY are 85 (85-91) TgN yr$^{-1}$, of which about 59 (56-59)% are lost to the atmosphere and the other 41 (40-44)% are exported to the coastal ocean.**

**Globally, TP inputs to rivers in LM3-FANSY are 8 (8-9) TgN yr-1, of which about 9 (6-10)% are stored within freshwaters and 91 (90-94)% are exported to the coastal ocean.**

In line 507 with the P budget, we now have added the following statements:

**IMAGE-GNM estimates that ~56% (5 of 9 Tg yr-1) of global TP inputs to rivers are stored within freshwaters (Beusen et al., 2016). This is a large difference from our estimate of 9 (6-10)% storage, but the difference is around a very uncertain number as it has not been directly measured. Our estimate of global river TP loads to the coastal ocean (7, 7-8 Tg yr$^{-1}$) falls within the range of other estimates (9 Tg yr$^{-1}$ from Global NEWS, Mayorga e al., 2010 and 4 Tg yr$^{-1}$ from IMAGE-GNM, Beusen et al., 2016, Table 5). The relatively low freshwater P retention in LM3-FANSY may arise, in part, from the lack of dams and reservoirs (see Sect. 4.5 for further discussion). The overall consistency of our SS and P estimates with the observed cross-watershed constraints (Fig. 2, 4), however, suggests that the bias introduced by the simplification may not be large.**

*Line 521: should be "Arctic".*

The word has been removed with the reduction of the N:P analysis following the reviewer's earlier comments.

*Line 551: grammar error: "are the among the …"*

We have modified "are the among"
to
"are among".

*Figures 6-8: The maps are for 1990 but the comparisons of concentrations, loads and yields are performed for 1982-2010. Why is there a difference in time between the two result analyses?*

As shown in our earlier response to the reviewer's major comment 5, we now have modified the years for the analyses.

[Figure]

**Figure 1: FANCY structure with arrows depicting fluxes of constituents of algae, nutrients, and solids in rivers and lakes. The constituents are listed in Table 1.**

[Figure]

**Figure 8: Pearson correlation coefficients (r) and p values (p) between the measurement-based vs. simulated annual loads from large U. S. rivers for the period ~1968-2000. The grey color represents load responses to the changes in the freshwater denitrification rate coefficient from 0.15 day⁻¹ to 0.05 day⁻¹ for N dynamics and the free parameter of terrestrial soil erosion by half for solid dynamics (Table 7).**

[Figure]

**Figure SI1: The 69 rivers considered for analyses are distributed broadly covering various climate and land use. The basin area of the 69 rivers covers 55% of global land area (excluding the Antarctic).**

[Figure]

**Figure SI3: Chlorophyll a concentrations (mg m$^{-3}$) in lakes for the year 1990.**

[Figure]

**Figure SI4: Pearson correlation coefficient (r) and p value (p) between the measurement-based vs. simulated DIN:DIP molar ratios across 35 rivers for the year 1990.**

[Figure]

**Figure SI5: Pearson correlation coefficients (r) and p values (p) between the measurement-based vs. simulated annual loads across 5 stations in the Mississippi River Basin for the periods ~1963-2000.**

[Figure]

**Figure SI6: Pearson correlation coefficients (r) and p values (p) between the measurement-based vs. simulated yields, loads, and concentrations across the world major rivers for the year 1990. Here the actual (instead of natural) water discharges of Meybeck and Ragu (2012) were used when calculating loads and yields from their multi-year average concentrations.**

[revised manuscript text omitted]

**Table SI10: Model sensitivities to the fractions of dividing TN and TP inputs from Beusen et al (2015) into different N and P species (Table 3), examined based on the percentage (%) difference in global river loads between the sensitivity and baseline simulations for the year 1990.**

---

## Author Response (AR3)

*This is the third time that I looked at this paper. The authors did not improve the model. So this new revision does not have new results from an improved model. I think that is a pity. The two major problems that still exists are:*

We objectively responded to all of the reviewer's comments in good faith during both of the previous rounds. In our first revision, we did substantially modify and improve the model and manuscript to address both reviewers' concerns. Thanks in part to these reviews and thousands of years of additional simulations, the model now recreates observed river solid and nutrient loads and concentrations as well or better than other published global models and has been more thoroughly validated than any of them. This validation is based on comparisons against observations (Figs. 2-4, 8, and SI5-6, Tables 4 and 7), other models (e.g., Fig. SI2, Tables 5-6), and published uncertainty ranges (Tables 5-6). It includes the cross-watershed validation of temporal mean conditions across globally distributed rivers, the time series validation of temporal variability and trends, and the validation of global means. We relied on and, when requested, extended above and beyond community-wide practices for assessing global river models over the last several decades.

In our second revision, we demonstrated using a simple uncertainty analysis that, contrary to the reviewer's concern, our P losses due to retention and N losses due to the sum of retention and denitrification fall safely within the bounds of current published uncertainties (Table 6, pages 27-28, lines 605-613 and page 29, lines 636-647). We also directly and transparently addressed the reviewer's concern that "the load may be correct for $t = 0$, but the trend is not" by adding a comparison of time trends against available time series from large rivers. While the availability of climate-scale time series for such a comparison is limited, the model trends are generally consistent with those observed (Table 7, page 36, lines 756-758). Further model improvements were thus not required to address the reviewer's second round of comments. We are thus surprised to see more general elements of model skill raised as a major issue in our third revision.

In this third revision, we have again endeavored to directly and thoroughly address the concerns raised by the reviewer. We have clarified that P retention in freshwaters is indeed larger than N retention, and we have addressed the new questions of model skill the reviewer has now raised. Our detailed responses are provided below. We objectively show that our model meets or exceeds current published global river modeling standards set by widely used and successful models, including models published in GMD. This is discussed explicitly in this response and, if any doubt remains, can easily be further verified by the editor and reviewer with the references we provide. The reviewer's contention that our model is not up to GMD and community skill standards is thus verifiably false. Further reviews will not change this fact. Nor will they change the corollary that the most widely used and successful global river nutrient models to date would have never been published if were the reviewer's criteria applied.

We share the reviewer's desire for improvement. We are cognizant of global model skill challenges we must meet as a community in the years to come and now highlight these limitations more in the Abstract. In particular, we emphasize that while the model shows considerable skill in matching contrasts of nutrient and SS loads and concentrations across globally distributed rivers, misfits for any single river can be substantial. This is a ubiquitous limitation of models designed for global robustness over regional optimality and we have no qualms against calling the communities attention to it. We hope that this additional transparency helps address the reviewer's concern.

We are proud of the results herein, which reflect the hard won fruits of years of development. We hope to have the opportunity to present what is now a uniquely thorough documentation of a skillful model that includes frank discussion of the model strengths and weaknesses to the Geoscientific Model Development community in a manner consistent with Geoscientific Model Development's mission.

Our detailed responses to each of the reviewer's specific comments are provided below.

**1. the RELATIVE difference between N and P retention is huge and mostly there is more P than N retention (in percentage). The new text does not answer this aspect.**

Our response to the reviewer's previous comment focused on contrasting the relative freshwater N and P loss mechanisms that control how much N and P made it to the river mouths. This breakdown is more commonly discussed in the literature, allowing for comparisons with additional published estimates. We demonstrated, in our previous response, that the high total loss of N relative to P when expressed as a fraction of their respective inputs falls within current uncertainty bounds, as acknowledged by Maranger et al. (2018). A key to this result, however, is that N losses include both denitrification and retention/burial, while P losses are just retention/burial. Upon re-reading the reviewer's previous comment, we see that the language shifted from "loss" at the outset of the comment to "retention" at the end. We focus more specifically on retention here.

Our retention results are consistent with the reviewer's expectation: P retention is higher than N retention when expressed as a fraction of the inputs. The 6-10% of P inputs lost due to freshwater processes are all from retention/burial, mainly reflecting the sorption of inorganic P onto solids and its deposition to bottom sediments. While organic N and P can accumulate within and be removed from the bottom sediments in response to variations in supply and remineralization in FANSY, the model does not include net long-term organic burial. Thus, effectively all of the 56-60% of N inputs lost due to freshwater processes are from denitrification. The model thus

estimates that P retention is indeed larger than N retention as a fraction of the inputs. This is now clearly stated on pages 27-28, lines 605-613 and page 29, lines 636-647.

The degree of net long-term organic N and P burial on a global scale is not well constrained. Maranger et al. (2018) estimated that around 15% of N inputs are lost due to burial. This estimate, however, is subject to uncertainties of similarly large scale to all other aspects of Maranger's first order budget. As the authors themselves state:

"For N, freshwaters appear to bury around 15% of terrestrial inputs (16 Tg N yr-1 , Table 1), however as outlined above, this number is based on the difference of a relatively imprecise loading estimate."

Net long-term organic burial could be added to FANSY as a simple fractional efficiency but, in the absence of better constraints and mechanistic contrasts between regions, we would be blindly shifting a secondary loss process. Instead, we now discuss the lack of net long-term organic burial in the model and the need to address this in the future through improved observation-based constraints and more realistic models of sediment diagenesis (page 43, lines 847-857).

The simplification of sediment dynamics is indeed shared by all published global river nutrient models. While such advances are needed, they would require a multi-year effort beyond the scope of this already comprehensive paper, and ultimately still suffer from a paucity of constraints. This is supported by the fact that we find that a model with no net long-term organic N and P burial can reconcile basic observed contrasts between N and P inputs and those observed at the river mouths within their admittedly large uncertainty ranges.

The model estimates of N and P retention and their comparison is now clearly stated as:

On pages 27-28, lines 605-613 as:

Globally, TN inputs to rivers (N inputs hereafter) in LM3-FANSY are 85 (85-91) TgN yr$^{-1}$, of which about 59 (56-60)% are lost to the atmosphere or retained within freshwaters (N loss/retention hereafter) and the other 41 (40-43)% are exported to the coastal ocean (N exports hereafter) (Table 6). **LM3-FANSY does not include net long-term N burial to bottom sediments, as all organic N delivered to the sediments is ultimately remineralized. While year to year sediment N inventories may vary, effectively all long-term N losses are to the atmosphere via freshwater denitrification.** LM3-FANSY estimates that N loss/retention is 144 (130-148)% of the N exports, consistent with the 147% and 143% of Galloway et al. (2004) and Seitzinger et al. (2006), yet larger than the 73% estimated by IMAGE-GNM (Beusen et al., 2016). LM3-FANSY estimates of 59 (56-60)% of the N inputs are lost or retained in freshwaters,

consistent with the 60% of Galloway et al. (2004), yet larger than the 42% of IMAGE-GNM (Beusen et al., 2015).

On page 29, lines 636-647 as:

Globally, TP inputs to rivers in LM3-FANSY are 8 (8-9) TgP yr$^{-1}$, of which about 9 (6-10)% are stored within freshwaters and 91 (90-94)% are exported to the coastal ocean (Table 6). IMAGE-GNM estimates that ~56% (5 of 9 TgP yr$^{-1}$) of the P inputs are stored within freshwaters (Beusen et al., 2016). This is a large difference from our estimate of 9 (6-10)% retention, but the difference is around a very uncertain number as the storage has not been directly measured. LM3-FANSY, which does not account for dams and reservoirs, likely underestimates global freshwater P retention by at least ~12% (Table 6, Maavara et al., 2015, see Sect. 4.5 for further discussion). The overall consistency of our SS, N, and P estimates with the observed cross-watershed constraints (Figs. 2-4, SI6, Table 4), however, suggests that the bias introduced by the lack of dams and reservoirs may not be large. In contrast, underestimates of P exports to the coastal ocean from high-exporting basins such as the Amazon, Ganges and Yangtze Rivers shown in Figure 2 of Harrison et al. (2019) imply that IMAGE-GNM likely overestimates global freshwater P retention. **Even though our freshwater P retention estimates are near the lower bound, our P retention estimates are far higher than those for N, mainly reflecting the sorption of PO$_4^{3-}$ onto solids and its deposition to bottom sediments.**

Discussion of the lack of net long-term organic burial in the model and the need to address this in the future is stated on page 43, lines 847-857.

The Rouse number-dependent transport criterion from Pelletier (2012) was adapted to simulate the deposition/resuspension fluxes between the suspended matters (i.e., ISS, PON, POP and PIP) and benthic sediments (i.e., Sed, SedN, and SedP). The criterion was designed to primarily simulate suspended loads, typically accounting for > 80% of total (i.e., suspended and bed) loads from most large (> ~100 km$^2$) river basins (Pelletier, 2012; Turowski et al., 2010), without explicitly modeling benthic sediments. **We acknowledge that our simplified benthic sediment component resulted from adapting the Pelletier's approach drives uncertainties in modeling the suspended matters and benthic sediments, including important diagenesis, other biogeochemical transformations, and physical processes (e.g., mineralization, denitrification, and net long-term organic burial) that occur within the benthic sediments. An implementation of more sophisticated benthic sediment dynamics (Chapra et al., 2008; Di Toro, 2001) with improved observation-based constraints and bed load transport processes is thus subject to critical future work.**

**2. The validation method.**

We have broken up this comment to ensure that each part is addressed.

***The one point validation is okay when you do a regression for a single time step. This model is dynamic and the authors first calibrate on this time point and then perform a validation on this same point.***

The model was calibrated based on the 1990 year results, yet validated based on 11 years (1990-2000) of annual results for the cross-watershed analysis (Table 4) and ~38 years (~1963-2000) of annual results for the 37 time series analysis (Table 7, Fig. 8). These additional efforts were made in our first revision in response to the reviewer's request to evaluate dynamic results, and in our second revision to add a time trend analysis. Thus, the model was not validated based on the single time point.

We also emphasize that FANSY is not a regression model. We are not fitting a simple set of predictor variables to river loads using a series of additive linear and/or non-linear relationships. Rather, our model, like others of its kind, is formulated as a set of interacting processes constrained as much as possible by mechanistic relationships and parameter values drawn from the literature. This overall mechanistic structure and a universal parameter set – the same parameters for all the basins (i.e., without tuning of each basin) – is imposed. We then perform a limited calibration of highly uncertain yet influential processes (e.g., denitrification) to remove first order biases (pages 17-18, lines 375-397). While the regression analogy can be useful, it is imperfect. The foundational validation comes from determining whether the model formulation, composed by both its mechanistic structure and mechanistic parameter values, can explain global contrasts in solid and nutrient concentrations and loads. This is a foundational comparison for almost all global models throughout the published literature (e.g., Mayorga et al., 2010; Tian et al., 2023; Pelletier et al., 2012). We have no objection to raising the validation bar and we have now added extensive additional evaluations at the reviewer's request. We strongly argue, however, that constructing a mechanistic process-based model that can skillfully capture many aspects of the observed global distribution of river solid and nutrient concentrations and loads is not a trivial result as the reviewer seems to imply. The literature is firmly on our side in this regard.

In addition, the cross-watershed validation method (based on the Pearson correlation coefficient (r) or the Nash–Sutcliffe model efficiency coefficient (NSE) of log-transformed data) that we have used in this study is not something that we have created, but it is the method that has been shared by almost all global river modeling communities for the last several decades, no matter whether a model is dynamic (e.g., DLEM-TAC, Tian et al., 2023) or not (e.g., Global NEWS, Mayorga et al., 2010). For sound reasons which we describe below, it is the standard approach

used when one is attempting to determine if a model can capture observed patterns that vary over orders of magnitude. This will be discussed further below.

***My hypothetical example of straight lines is an example of a set of models which will have a perfect fit. But this example could also be for parabolic or hyperbolic functions. However the authors react with another statistical method to prove that the model is different than a straight line (I think, because I could not fully understand what they were now adding to the manuscript).***

As we articulate above, we are not performing a regression analysis and the model will never have a perfect fit. The fit achieved reflects the degree to which the imposed process-based model structure, parameters, and forcing can represent the global patterns in river nutrient concentrations and loads.

The reviewer's concern about the calibration was:

"This is exactly why this calibration method is not good enough. The load could be correct for t=0, but the trend is not."

We responded by adding a comparison of time trends against observed ones from select rivers where climate-scale time series were available. We found that the trends in the model, which was not calibrated but emerged naturally from the imposed forcing and process-based model dynamics, are largely consistent with those observed (Table 7, page 36, lines 756-758). This was the most direct way that we could possibly respond to the reviewer's concern and we are confused by this reviewer's response. Perhaps it is linked to miscommunication of the differences between our model data comparison and a standard regression? We have attempted to clarify this above. We cannot simply "fit the slope" in the process-based model FANSY and every change we make requires thousands of years of integration. Even if we could, the limited climate scale time trend data would provide little in the way of new constraints. We do, however, view the fact that the simulated model trends are largely consistent with those observed as a positive, as it builds support for the model's capacity to simulate future changes.

***Conclusion: I don't think that the new version of the manuscript has improved.***

We feel that we responded directly and convincingly to the two primary requests that the reviewer made in the reviewer's second round of comments. We reconciled our results with those of Maranger et al. (2018) by showing that our results are fully within the bounds of current published uncertainties (and have now further clarified this in our response to the reviewer's above comment). We added the additional analysis of time trends in response to the reviewer's

specific request to do so. In this revision, we again have done our best to respond to the reviewer's new comments below.

*A very simple test to look at the results/validation is to take the average over all observations and the model results (coupled to the observations). I made an overview of Figure SI6. The expectation of this exercise is that the average of both is almost equal. I divided the observation by the model result to get a fraction. When the fraction is below 1, then the model gives an underestimation. When the fraction is above one, the model gives an underestimation.*

*My conclusions from this table:*
*SS: Model gives an underestimation of more than a factor of 2. Not good.*
*NO3: Yield and Loads reasonable, but overestimation of concentration.*
*NH4: Good.*
*DON: Yield fine, Loads to low, concentration to low.*
*TKN: Overestimation of results.*
*PO4: Yield and Loads underestimation, concentration small difference.*
*DOP: Yield fine, Loads underestimation, concentration overestimation.*
*TP: underestimation of results.*

*So suspended solids needs some attention. The loads of all phosphorus forms are all underestimated. Which is a huge problem, because there is almost no loss in the rivers, so it is not*
*easy to increase the phosphorus loads! The concentration of all nitrogen forms are overestimated and for the loads nitrate is compensating DON.*
*Based on the analyses above, I really think the model should be improved! The model description is OK now, but the results and implementation is still too poor to meet the standards of GMD.*

We disagree with these comments and will clearly and objectively rebut them in 3 steps:

1. We will demonstrate that the arithmetic mean which the reviewer suggests as a way to compare the model results and observations results in a misfit that largely reflects the model's fidelity to the largest few rivers, rather than its capacity to represent globally distributed rivers of different magnitudes and characteristics. The latter is our goal and the log-transform is a community standard tool for quantifying the extent to which this has been achieved.
2. We will clearly show that the widely-used IMAGE-GNM and Global NEWS models, the former of which is published in GMD, would have been rejected if the reviewer's criteria were applied. This, combined with the facts that our model has achieved a skill

comparable to these models using standard approaches while including far more comparisons objectively rebuts the reviewer's contention that the skill does not meet GMD and broader community standards.

3. We have prominently recognized (in the Abstract) the potential of global models to have substantial misfits for individual rivers. Hopefully this will raise awareness of this shared limitation of all global river solid and nutrient loading products. Addressing this, however, is a community-wide challenge requiring improvements in observations, model forcing and dynamics that will take years. We have enhanced discussion of pathways to model improvement.

Each of these steps is described in more detail below.

**Step 1: The problem with using the cross-site arithmetic mean as a global skill metric**

The reviewer's contention is that the model should reproduce the sum/average of loads across the rivers that happen to have been sampled. When river solid and nutrient concentrations and loads vary over many orders of magnitude, however, this choice emphasizes only one or a few large rivers in the sample. Our goal, as is the case with other global models, is to capture global contrasts between rivers of vastly different magnitudes and nutrient concentrations. The log-transformed data is the standard tool for this (e.g., Mayorga et al., 2010; Tian et al., 2023; Pelletier et al., 2012) because it weights fits equally across different magnitudes (a factor of 2 misfit is penalized the same whether the baseline value is 1 or 1000). The "Bad" classifications the reviewer has given to some quantities using the reviewer's arithmetic mean approach does not reflect a systematic bias across rivers, but is primarily because the largest river happened to be either over- or under-estimated.

To illustrate the implications of the reviewer's alternative evaluation method, we consider the fit to suspended solids (SS). As the reviewer points out, the ratio of the arithmetic mean of the observed to modeled SS loads is 2.35. The observed loads, however, vary over 4 orders of magnitude (e.g., 118 to 7,846,601 kt/yr for SS loads) across globally distributed rivers. Removing the single largest river (i.e., Huang He/Haiho) of the 65 sampled rivers, reduces the ratio from 2.35 to 1.03 (which becomes "Good" according to the reviewer's judgment criteria). *This example shows how the arithmetic mean is more a measure of the model fit to the largest river, rather than an assessment of the model fit across globally distributed rivers varying in size and other climate/land use conditions. Like other global modeling studies, our primary goal is the latter.*

**Step 2: Is the model fit sufficient for publication?**

To further investigate the issue of whether the reviewer's ratio provides a valid measure of whether our model's skill is sufficient to merit publication, we repeated the suggested evaluation for the most widely known models Global NEWS and IMAGE-GNM (the latter is also published in GMD). As shown in the below Table R1, our model achieves a similar level of skill with the reviewer's metric as these two alternative models despite being less directly fit to the data. ***That is, the arithmetic mean ratio-based evaluation and criteria the reviewer suggests would have led to the rejection of these prominent and highly successful models. It thus clearly does not reflect GMD or broader community standards***.

| | LM3-FANSY | | | Global NEWS | | | IMAGE-GNM |
|---|---|---|---|---|---|---|---|
| | Yields | Loads | Concentrations | Yields | Loads | Concentrations | Concentrations |
| SS | 2.40 | 2.35 | 2.23 | 1.05 | 0.83 | 2.40 | |
| $NO_3^-$ | 0.79 | 0.87 | 0.60 | | | | |
| $NH_4^+$ | 1.03 | 1.15 | 0.92 | | | | |
| DIN | 0.76 | 0.96 | 0.68 | 0.98 | 0.76 | 1.13 | |
| DON | 1.01 | 1.66 | 0.51 | 0.68 | 0.57 | 0.77 | |
| TKN | 0.53 | 0.56 | 0.41 | | | | |
| $PO_4^{3-}$ | 1.54 | 1.82 | 1.22 | 1.31 | 1.21 | 1.30 | |
| DOP | 0.86 | 1.42 | 0.37 | 0.93 | 1.02 | 0.88 | |
| TP | 1.71 | 2.45 | 1.58 | 2.00 | 1.92 | 3.39 | |
| TN | | | | | | | 0.68 |

Table R1: Ratio of the arithmetic mean of the observed to modeled yields, loads, and concentrations. The ratio for Global NEWS was calculated by using the data reported in Fig. SI2 and the supplementary excel file. The ratio for IMAGE-GNM was calculated by using the data reported in Figure 10 and the supplementary excel file of Beusen et al. (2015).

The log transform that we and others (e.g., Table R2, Mayorga et al., 2010; Tian et al., 2023; Pelletier et al., 2012) have used for model validation addresses the issue demonstrated above by ensuring that the model fit is not disproportionately influenced by a few very large rivers. That is, following the log transform, a factor of 2 misfit for a base SS load of 118 kt/yr is given the same weight as a factor of 2 misfit around a value of 7,846,601 kt/yr. Without the log-transform, higher values are given much greater weight than lower ones. For this reason, no matter whether a model is dynamic/process-based (e.g., Tian et al., 2023) or statistical (e.g., Global NEWS), log transformed data analysis based on the Pearson correlation coefficients (r), Determination coefficient ($R^2$), and/or Nash–Sutcliffe model efficiency coefficients (NSE) has been a standard means of cross-watershed validation of global river models over the last several decades. Throughout the manuscript, we have shown that our overall model skill based on this analysis metric is at least as good as the skill of previous models (e.g., Fig. SI2 and refer to the explicit discussion of relative skill throughout our Results section). We feel this is particularly notable

given that our model is simulating coupled algae, solid, and nutrient cycles at the global scale for the first time.

Furthermore, thanks in part to the reviews, the extent of our evaluation now far surpasses that of any other global river models in the published literature. We have provided a Table R2 below that contrasts the extensive comparisons in our paper with those included in Global NEWS, IMAGE-GNM, DLEM-TAC, and the model of Pelletier et al. (2012). ***The FANSY model development description and validation thus objectively meet and exceed those of the past model documentation papers of GMD and others.***

| | LM3-FANSY | DLEM-TAC | IMAGE-GNM | Global NEWS | Pelletier et al. (2012) |
|---|---|---|---|---|---|
| Instream biogeochemical, physical, and/or hydrological processes | | | | | |
|    Process-based | √ | √ | | | |
|    Dynamic | √ | √ | √ | | |
| Cross-watershed validation using one time point results (i.e., temporal mean condition validation) | | | | | |
|    Log transformed | √ | √ | | √ | √ |
|    Statistics | r, NSE, prediction error | $R^2$ | No statistics | $R^2$, NSE, prediction error | r |
| Time series validation using yearly and/or monthly results (i.e., temporal variability and/or trends validation) | | | | | |
|    Statistics | r, prediction error, t-statistic of linear trend | $R^2$, NSE | RMSE | No comparison | No comparison |
| Analyzed unit | | | | | |
|    Yield | √ | | | √ | √ |
|    Load | √ | √ | | √ | |
|    Concentration | √ | | √ | | |
| Validated chemical species | | | | | |
|    SS | √ | | | √ | √ |
|    $NO_3^-$ | √ | | | | |
|    $NH_4^+$ | √ | | | | |
|    DIN | √ | | | √ | |
|    DON | √ | | | √ | |
|    TKN | √ | | | | |
|    $PO_4^{3-}$ | √ | | | √ | |
|    DOP | √ | | | √ | |
|    TP | √ | | √ | | |
|    TN | √ | | √ | | |
|    DIC | | √ | | | |
|    DOC | | √ | | | |
|    POC | | √ | | | |
|    TOC | | √ | | | |

Table R2: Synthesis of the model validation and approaches used in model documentation papers for widely enlisted models. Note the prevalence of the log-transformed approach, and the much more extensive evaluation included in this study relative to the others. In the table, "process-based" refers to a model that resolves freshwater biogeochemical and physical processes in mechanistic ways. "Dynamic" refers to a model that provides time series results.
Pearson correlation coefficients (r), Determination coefficient ($R^2$), Nash–Sutcliffe model efficiency coefficients (NSE), and Root mean squared error (RMSE).

**Step 3: Communicating current skill limitations and a pathway for further improvement**

Finally, we agree with the reviewer that the misfits of global models like ours can be quite large for individual rivers and that this impacts the uncertainty in global total/mean loads. This is, however, a community-wide challenge common across global river models that must value

global robustness over regional optimality. This fundamental challenge still exists for even the simplest of variables (e.g., surface air temperatures in global climate models still have prominent regional biases). Regional fidelity must be addressed via concerted improvements in observations, models, and forcings (e.g., precipitation, fertilizer estimates) and thousands of years of simulations over the course of many years. Addressing these fundamental limitations is a "years to decades" problem. It is not something we can do within the timescale of a response to a reviewer. We have extensively noted the prominent uncertainties that will be prioritized in future work towards achieving this goal in the Discussion (pages 42-44, lines 829-900). In addition to this, we now call attention to the challenge of regional fidelity in the Abstract, explicitly noting that *"While the simulations are able to capture significant cross-watershed contrasts at a global scale, disagreement for individual rivers can be substantial. This limitation is shared by other global river models and could be ameliorated through further refinements in nutrient sources, freshwater model dynamics, and observations."*

References

**Tian, H., Yao, Y., Li, Y., Shi, H., Pan, S., Najjar, R. G., et al.: Increased terrestrial carbon export and CO2 evasion from global inland waters since the preindustrial era. Global Biogeochemical Cycles, 37, e2023GB007776. https://doi.org/10.1029/2023GB007776, 2023.**

**Di Toro, D. M.: Sediment Flux Modeling. Wiley-Interscience, New York, NY, USA, 2001.**